# Simulating age of air and the distribution of $SF_6$ in the stratosphere with the SILAM model

Rostislav Kouznetsov[1,2], Mikhail Sofiev[1], Julius Vira[1,3], and Gabriele Stiller[4]

[1]Finnish Meteorological Institute, Helsinki, Finland
[2]Obukhov Institute for Atmospheric Physics, Moscow, Russia
[3]Cornell University, Ithaca, NY, USA
[4]Karlsruhe Institute of Technology, Karlsruhe, Germany

**Correspondence:** Rostislav Kouznetsov (Rostislav.Kouznetsov@fmi.fi)

**Abstract.** The paper presents a comparative study of age of air (AoA) derived with several approaches: a widely used passive tracer accumulation method, the $SF_6$ accumulation, and a direct calculation of an "ideal age" tracer. The simulations were performed with the Eulerian chemistry transport model SILAM driven with the ERA-Interim reanalysis for 1980-2018.

The Eulerian environment allowed for simultaneous application of several approaches within the same simulation, and interpretation of the obtained differences. A series of sensitivity simulations revealed the role of the vertical profile of turbulent diffusion in the stratosphere, destruction of $SF_6$ in the mesosphere, as well as the effect of gravitational separation of gases with strongly different molar masses.

The simulations reproduced well the main features of the $SF_6$ distribution in the atmosphere retrieved from the MIPAS satellite instrument. It was shown that the apparent very old air in the upper stratosphere derived from the $SF_6$ profile observations is a result of destruction and gravitational separation of this gas in the upper stratosphere and mesosphere. These processes make the apparent $SF_6$ AoA in the stratosphere several years older than "ideal-age" AoA, which, according to our calculations, does not exceed 6-6.5 years. The destruction of $SF_6$ and varying rate of emission make $SF_6$ unsuitable for reliably deriving AoA or its trends. However, observations of $SF_6$ provide a very useful means for validation of stratospheric circulation in a model with properly implemented $SF_6$ loss.

## 1 Introduction

AoA is defined as the time spent by an air parcel in the stratosphere since its entry across the tropopause (Li and Waugh, 1999; Waugh and Hall, 2002). The distribution of the age of air (AoA) is controlled by the global atmospheric circulations, first of all, the Brewer-Dobson and the polar circulation. In particular, the temporal variation of AoA has been used as an indicator of long-term changes in the stratospheric circulation (Engel et al., 2009; Waugh, 2009). AoA has been extensively used to evaluate and compare general circulation and chemical transport models in the stratosphere (Waugh and Hall, 2002; Engel et al., 2009).

Simulations of the AoA according to the definition above have been performed with Lagrangian transport models. The trajectories are initiated with positions distributed in the stratosphere and integrated backwards until they cross the tropopause. The time elapsed since the initialization is attributed as age of air at the point of initialization. Moreover, the distribution of

the ages of particles originating from some location can be used to get the age spectrum there. Until recently Lagrangian

simulations of AoA did not explicitly account for turbulent mixing in the stratosphere (Eluszkiewicz et al., 2000; Waugh and Hall, 2002; Diallo et al., 2012; Monge-Sanz et al., 2012). The account for mixing adds up to two years to the mean AoA in tropical upper stratosphere (Garny et al., 2014). In Lagrangian formulation the mixing can be simulated with random-walk of particles (Garny et al., 2014), or by inter-parcel mixing (Plöger et al., 2015; Brinkop and Jöckel, 2019).

The Eulerian simulations of AoA can be formulated in several different ways. The approaches with an accumulating tracer,

whose mixing ratio increases linearly in the troposphere, were used in a comprehensive study by Krol et al. (2018) and several studies before (e.g. Eluszkiewicz et al., 2000; Monge-Sanz et al., 2012). Another approach is to simulate a steady distribution of a decaying tracer, such as $^{221}$Rn, emitted at the surface at a constant rate (Krol et al., 2018). Besides that, a special tracer that is analogous to a Lagrangian clock has been used. The tracer appears in literature under names "clock-type tracer"(Monge-Sanz et al., 2012) or "ideal age" (Waugh and Hall, 2002). The ideal age has constant rate of increasing of mixing ratio everywhere,

except for the surface where it is continuously forced to zero. Similar tracers have been long used to simulate transport times of oceanic water (e.g. England, 1995; Thiele and Sarmiento, 1990).

Direct observations of the age of air, as it is defined above, are not possible; therefore AoA is usually derived from observed mixing ratios of various tracers. The tracers belong to one of two types: various tracers with known tropospheric mixing ratios and lifetimes (Bhandari et al., 1966; Koch and Rind, 1998; Jacob et al., 1997; Patra et al., 2011), and long-living tracers with

known variations in tropospheric mixing ratios. The studies published to-date used carbon dioxide $CO_2$ (Andrews et al., 2001; Engel et al., 2009), nitrous oxide $N_2O$ (Boering et al., 1996; Andrews et al., 2001), sulphur hexafluoride $SF_6$ (Waugh, 2009; Stiller et al., 2012), methane $CH_4$ (Andrews et al., 2001; Remsberg, 2015), and various fluorocarbons (Leedham Elvidge et al., 2018).

For accumulating tracers, the mean AoA at some point in the stratosphere is calculated as a lag between the times when a

certain mixing ratio is observed near the surface and at that point. The lag time is equivalent to the mean AoA defined above only in the case of a strictly linear growth and uniform distribution of the tracer in the troposphere (Hall and Plumb, 1994).

In reality, there is no tracer whose mixing ratio in the troposphere grows strictly linearly. The violation of assumption of the linear growth leads to biases in the resulting AoA distribution and its trends. It has been pointed out that increasing growth rates of $CO_2$ and $SF_6$ lead to a low-bias of AoA and its trends, and make these tracers ambiguous proxies for AoA (Garcia et al.,

2011). Various corrections have been applied in several studies (Hall and Plumb, 1994; Waugh and Hall, 2002; Engel et al., 2009; Stiller et al., 2012; Leedham Elvidge et al., 2018) to deduce the "true" AoA from observations of tracers with increasing growth rates. The effect of the correction method on AoA estimates has not been investigated and must be considered as source of uncertainty in resulting estimates (Garcia et al., 2011). Garcia et al. (2011) further conclude that accounting for the biases in trend estimates due to varying growth rates would likely require uniform and continuous knowledge of the evolution of trace

species, which is not available from any existing observational dataset. Recently Leedham Elvidge et al. (2018) have shown a minor sensitivity of AoA to the choice of particular correction method, however without detailed analysis of assumptions behind these methods. For a similar problem with ages of oceanic water it has been shown (Waugh et al., 2003) that in case of

a non-linearly varying tracer the tracer age is strongly influenced by the shape of transient time distribution (TTD, also known as an "age spectrum") at particular location and time.

Another major source of uncertainty in observational AoA is violation of conservation of a tracer due to sources and sinks, such as oxidation of carbon monoxide and methane for $CO_2$, or mesospheric destruction for $SF_6$. The mesospheric sink of $SF_6$ leads to an "over-aging", especially pronounced in the area of polar vortices. The magnitude of the over-aging was estimated as 2 or more years Waugh and Hall (2002). Besides being visible in many evaluations, e.g. Stiller et al. (2012, Fig. 4), Kovács et al. (2017, Fig. 8), a dedicated study on the over-aging of polar winter stratospheric air was performed by Ray et al. (2017,

Fig. 4).

The simulations of $SF_6$ and AoA in the atmosphere with WACCM model (Kovács et al., 2017) have reproduced the effect of over-aging, but of much smaller magnitude than if inferred from $SF_6$ retrievals from the limb-viewing **MIPAS** instrument operated on-board of the Envisat satellite in 2002-2012 (Stiller et al., 2012), and *in-situ* observations from the ER-2 aircraft (Hall et al., 1999). Kovács et al. (2017) offered two possible scenarios for the discrepancy: either $SF_6$ loss is still underestimated

in WACCM, or MIPAS $SF_6$ is low biased above $\sim 20\,\mathrm{km}$. Neither of the scenarios have been analysed in depth so far, which leaves the status of MIPAS, the richest to date observational dataset for the stratospheric $SF_6$, unclear.

The aim of the present study is to provide self-consistent simulations of spatio-temporal distribution of AoA and of the $SF_6$ mixing ratio in the troposphere and stratosphere during last 39 years. The main modelling tool is the Eulerian chemistry transport model SILAM (backronym for System for Integrated modeLling of Atmospheric coMposition). The stratospheric

balloon observations and retrievals from the limb-viewing **MIPAS** instrument operated on-board of the Envisat satellite in 2002-2012 are used as a validation for the simulated distribution.

With the results of these simulations we

   – compare different methods of estimating the AoA and quantify inconsistencies in AoA and its trends arising from violations of the underlying assumptions behind each method

– analyze the causes of the discrepancies in the upper stratosphere between different methods of deriving the AoA

   – provide a solid basis for further studies of stratospheric circulation with observations of various trace gases and for studies of climate effects of $SF_6$

The paper is organized as follows. Sec. 2 gives an overview of the modelling tools, and the modelling and observational data used for the study. Sec. 3 describes the developments made for SILAM in order to perform the simulations: vertical eddy

diffusivity parametrisation in the stratosphere and the lower mesosphere, and $SF_6$ destruction parametrization, as well as the modelling setup. The sensitivity tests and evaluation of the simulations against MIPAS satellite retrievals, and stratospheric-balloon measurements of $SF_6$ mixing ratios are given in Sec. 4. Sensitivity of AoA and its trends to the choice of the simulation setup and AoA proxy is studied in Sec. 5. The findings of the whole study are summarised in Sec. 7.

## 2    Methods and input data

{sec:ma

### 2.1    SILAM model

SILAM (System for Integrated modeLling of Atmospheric coMposition) is an off-line chemical-transport model. SILAM features a mass-conservative and positive-definite advection scheme that makes the model suitable for long-term runs (Sofiev et al., 2015). The model can be run at a range of resolutions starting from a kilometer scale in limited-area or in global mode. The vertical structure of modelling domain consist of stacked layers starting from the surface. The layers can be defined either in z- or hybrid sigma-pressure coordinates. The model can be driven with a variety of NWP- (numerical weather prediction) or climate models.

In order to accurately model the AoA and needed tracers, the vertical diffusion part of the transport scheme of SILAM has been refined to account for gravitational separation. In addition, several tracers with corresponding transformation and transport routines have been implemented into the model. The model setup used for the present study is described in Sec. 3.5.

### 2.2    ECMWF ERA-interim reanalysis

The ERA-Interim reanalysis from the European Centre for Medium-range Weather Forecasts (ECMWF) had been used as a meteorological driver for our simulations. The data set has T255 spectral resolution and covers the whole atmosphere with 60 hybrid sigma-pressure levels (Dee et al., 2011), having the uppermost layer from 0.2 to 0 hPa with nominal pressure of 0.2 hPa. The reanalysis uses a 12-h data assimilation cycle, and the forecasts are stored with a 3-hour time step. We used the fields retrieved form the ECMWF's MARS archive on a lat-lon grid 500x250 points with step of 0.72 degrees. The four forecast times (+3h, +6h, +9 h and +12h) were used from every assimilation cycle to obtain a continuous dataset with 3-hour time step. To drive the dispersion model, the data on horizontal winds, temperature and humidity for 1980-2018 were used. The procedure for diagnosing the vertical transport is desctribed in Sec. 3.5.

The ERA-interim reanalysis has been used earlier for Lagrangian simulations of AoA (Diallo et al., 2012) and found to provide ages that agree with those inferred from in-situ observations in the lower stratosphere.

### 2.3    MIPAS observations of $SF_6$

{sec:MI

To evaluate the results of $SF_6$ modelling we used the data from the MIPAS instrument operated on-board of Envisat satellite in 2002-2012. MIPAS was a limb-sounding Fourier transform spectrometer with a high spectral resolution measuring in the infrared. Due to its limb geometry, a good vertical resolution of the derived trace gas profiles and a high sensitivity to low-abundant species around the tangent point has been achieved. Along the orbit path, MIPAS measured a profile of atmospheric radiances about every $400\,km$ with an altitude coverage, in its nominal mode, of about $6-70\,km$. The vertical sampling was $1.5\,km$ in the lower part of the stratosphere (up to $32\,km$) and $3\,km$ above, with a vertical field of view covering $3\,km$ at the tangent point. Over a day, about 1300 profiles along 14.4 orbits were measured, covering all latitudes up to the poles at sunlit

and dark conditions. The vertical distributions of trace gases were derived from the radiance profiles by an inversion procedure,
fitting simulated spectra to the measured ones while varying the atmospheric state parameters.

The retrieval of $SF_6$ is based on the spectral signature of this species in the vicinity of $10.55\,\mu m$ wavelength and is in principle described in Stiller et al. (2008, 2012); Haenel et al. (2015). In this study here, we use an updated version of $SF_6$ data (compared to the one described in Haenel et al. (2015)) called V5H/R_SF6_21/224/225; the absorption cross-section data on $SF_6$ and a new CFC-11 band in the vicinity of the $SF_6$ signature by Harrison (2018) has been used instead of the older cross
section data by Varanasi et al. (1994). The updated version provides considerably higher $SF_6$ mixing ratios in the upper part of the stratosphere (above $30\,km$) than the versions before and is closer to independent reference data.

The retrieved profiles are sampled on an altitude grid spaced at $1\,km$, where as the actual resolution of the profiles is between $4$ and $10\,km$ for altitudes below $30\,km$. The retrievals are supplemented with averaging kernels and error covariance matrices describing uncertainties due to random noise in the radiance measurements, called measurement noise error or target noise
error or retrieval noise error in the following. This error component, which is normally in the order of 10% of the retrieved value, is fully uncorrelated from profile to profile, and therefore virtually cancels out when averaging over a large number of profiles. In contrast, there exist systematic error components that are fully correlated between the profiles. Their assessment is difficult and depends on the knowledge about sources of systematic errors. Stiller et al. (2008) has assessed them to be in the order of 10% at 60 km, and 4% at 30 km. These error components have to be considered when comparisons of larger datasets
(monthly or seasonal means) with other data are performed.

## 3    SILAM developments

The destruction of atmospheric $SF_6$ occurs at altitudes above 60 km (Totterdill et al., 2015) that fall within the topmost layer of the ERA-Interim. The exchange processes in the upper stratosphere and lower mesosphere have to be adequately parameterized together with the destruction process. In our simulations we have suppressed the transport with of $SF_6$ mean wind above the
modelling domain top (0.1 hPa, 65 km) and parameterized the $SF_6$ loss due to the eddy and molecular diffusion towards the altitudes where destruction occurs. In this section we introduce the set of parametrizations that were implemented in SILAM for this study.

### 3.1    $SF_6$ destruction

Estimates of AoA from the $SF_6$ tracer rely on the assumption of it being a passive tracer. $SF_6$ is indeed essentially stable in
the troposphere and stratosphere. IPCC (2013, Sec 8.2.3.5) mentions that photolysis in the stratosphere as the main mechanism of $SF_6$ loss, however without any reference to original studies. The statement is probably taken from Ravishankara et al. (1993). Reddmann et al. (2001) pointed associative electron attachment in the upper stratosphere and mesosphere as the main destruction mechanism for $SF_6$ below 80 km. The recent study of Totterdill et al. (2015) gives some 1-2 order of magnitude slower rates of electron attachment, however keeping it the dominant mechanism of the $SF_6$ destruction in the altitude range
up to 100 km. The highest destruction rate of $1 \times 10^{-5}\,s^{-1}$ occurs at the altitude of 80 km (Fig. 1). An important feature

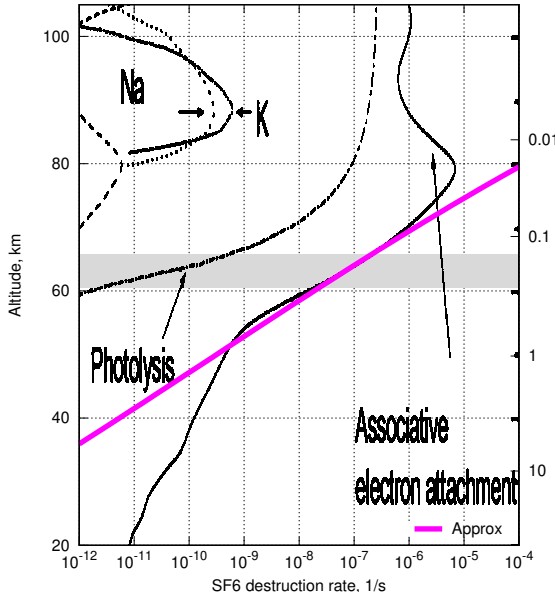

**Figure 1.** The vertical profiles of $SF_6$ destruction rate (after Totterdill et al., 2015) and its approximation in range of 55-75 km, given by Eq. (1).                                                                                                                                {fig:sf

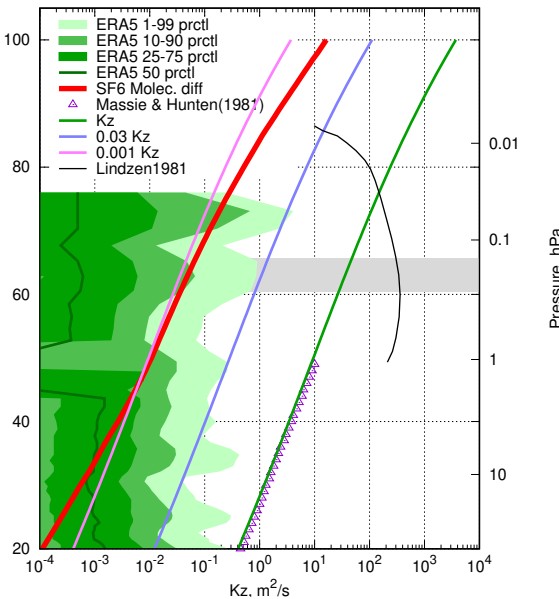

**Figure 2.** Vertical profiles of diffusion coefficients. The distribution of ERA5 profiles of the "mean turbulent diffusion coefficient for heat" parameter, molecular diffusivity for $SF_6$ in standard atmosphere, and three prescribed Kz profiles. The eddy diffusion profile due to breaking gravity waves (after Lindzen, 1981) is given for a reference.                                                                        {fig:di

of this profile is that the destruction rate becomes significant above the top of our modelling domain (0.1 hPa, 65 km). The ERA-Interim meteorological fields have the uppermost level at $0.1\,\mathrm{hPa}$ and do not resolve a vertical structure of the atmosphere above that level. In order to assess the loss of $SF_6$ due to destruction we have to parameterise the combined effect of transport of $SF_6$ transport through the $0.1\,\mathrm{hPa}$ and its destruction. Then the resulting fluxes can be applied as the upper boundary condition

for our simulations.

As an approximation to the vertical profile of the destruction rate in an altitude range of 50–80 km we have fitted a corresponding part of the curve in Fig. 9a of Totterdill et al. (2015) with a power function of pressure (magenta line in Fig. 1):

$$\frac{1}{\tau} = 3 \times 10^{-8}\,\mathrm{s}^{-1} \left( \frac{0.2\,\mathrm{hPa}}{p} \right)^3 , \qquad\qquad (1) \qquad \texttt{\{eq:Des}$$

where $\tau$ is the lifetime of $SF_6$ at the altitude corresponding to pressure $p$.

## 3.2  Eddy diffusivity

`{sec:kz`

Large variety of vertical profiles for eddy diffusivity in the stratosphere and lower mesosphere can be found in literature. In many studies in 1970-s – 1980-s the vertical profiles were derived from observed tracer concentrations neglecting the mean transport. Most studies suggested that the vertical eddy diffusion has a minimum of 0.2-0.5 m$^2$/s (Pisso and Legras, 2008) at

15-20 km agreeing quite well to the ones derived from radar measurements in the range of 15-20 km Wilson (2004). Above that altitude $K_z$ was suggested to gradually increases by about 1.5 orders of magnitude at 50 km due to breaking gravity waves (Lindzen, 1981).

The theoretical estimates of the effective exchange coefficients considering the layered and patchy structure of stratospheric turbulence suggest 0.5–2.5 m$^2$/s for the upper troposphere and 0.015–0.02 m$^2$/s for the lower stratosphere (Osman et al., 2016),

which is about an order of magnitude lower than the estimates above.

The values of eddy exchange coefficient at heights of 10-20km estimated from high-resolution balloon temperature measurements (Gavrilov et al., 2005) are $\sim 0.01$ m$^2$/s with no noticeable vertical variation. It is not clear, however, how representative the derived values are for UTLS in general. We could not find any reliable observations of vertical diffusion in a range of 30-50 km.

The parameterisation for vertical eddy diffusivity above the boundary layer used in SILAM has been adapted from the IFS model of the European Centre for Medium-range Weather Forecasts (ECMWF, 2015). However, in the upper troposphere the predicted eddy diffusivity is nearly zero. For numerical reasons a lower limit of 0.01 m$^2$/s is set for $K_z$ in SILAM. Our sensitivity tests have shown that long-term simulations are insensitive to this limit as long as it is low enough (see results and discussion). The $K_z$ in the stratosphere is routinely set to the limiting value with relatively rare peaks, mostly in UTLS.

Such scheme essentially turns off turbulent diffusion in the stratosphere. Same is true for recent ERA5 reanalysis dataset (Copernicus Climate Change Service (C3S), 2017) that provides the values of $K_z$ among other model-level fields: the eddy diffusion routinely falls below the molecular diffusivity above 40 km (Fig. 2).

As a reference for this study, we took a tabulated profile of Hunten (1975), as it was quoted by Massie and Hunten (1981). The original profile covers the range up to 50 km, and the extrapolation up to 80 km matches the theoretical estimates by

(Lindzen, 1981) and by Allen et al. (1981). We approximate the profile as a function of pressure in the range of $100 - 0.01\,\mathrm{hPa}$ ($15 - 60\,\mathrm{km}$):

$$K_z(p) = 8\,\mathrm{m}^2/\mathrm{s} \left(\frac{1\,\mathrm{hPa}}{p}\right)^{0.75}. \tag{2} \quad \{\mathrm{eq:Emp}$$

The approximated profile was stitched with the default SILAM profile with a gradual transition within an altitude range of $10 - 15\,\mathrm{km}$ to keep the tropospheric dispersion intact. This profile gives values of $K_z$ is $3 - 6$ orders of magnitude higher

than ones provided by ERA5 reanalysis (Fig. 2), and 1-2 orders of magnitude higher than more recent estimates (Legras et al., 2005).

In order to cover the range of vertical profiles of $K_z$ between ERA5 profiles and the reference one (2) we used two intermediate profiles obtained by scaling the reference one with factors 0.03 and 0.001. The three prescribed eddy-diffusivity profiles are hereinafter referred as "1Kz", "0.03Kz", and "0.001Kz" respectively. The dynamic eddy-diffusivity profile adopted from

the ECMWF IFS model is referred to as "ECMWF Kz". In all simulations the parameterization of $K_z$ in the troposphere is the same, and linear transition from the SILAM $K_z$ to the prescribed one occurs in the altitude range of $10 - 15\,\mathrm{km}$.

### 3.3 Molecular diffusivity and gravitational separation

In tropospheric and stratospheric CTMs gaseous admixtures are transported as tracers, i.e. advection and turbulent mixing do not depend on a species properties, whereas the molecular diffusion is negligible. Models that cover the mesosphere, such as

WACCM (Smith et al., 2011), account for molecular diffusion explicitly. Since some of the $K_z$ parametrizations above often result in values below the molecular diffusivity, the parametrization of molecular diffusion has been implemented in SILAM.

The molecular diffusivity of $SF_6$ in the air at temperature $T_0 = 300$ K and pressure $p_0 = 1000\,\mathrm{hPa}$, is $D_0 = 1 \times 10^{-5}\,\mathrm{m\,s}^{-2}$ (Marrero and Mason, 1972, Table 22). The diffusivity at a different temperature $T$ and pressure $p$ is given by:

$$D = D_0 \frac{p_0}{p} \left(\frac{T}{T_0}\right)^{3/2}, \tag{3} \quad \{\mathrm{Dpt}\}$$

see e.g. Cussler (1997). The vertical profile of molecular diffusivity in the US standard atmosphere (NOAA et al., 1976) is shown in (Fig. 2). Note that the value for the reference diffusivity of $SF_6$ used in this paper is about a half of the one used in simulations with WACCM by Kovács et al. (2017). The reason is that WACCM uses a universal parametrization (Smith et al., 2011, Eq. 7 there) for all compounds. That parametrization relies solely on molecular mass of a tracer and does not account for e.g. the molecule collision radius. The latter is about twice larger for $SF_6$ molecule than for most of stratospheric tracers.

Thus, for this study we use the value from Marrero and Mason (1972), which results from fitting laboratory data for diffusion of $SF_6$ in the air.

The vertical diffusion transport velocity of admixture with number concentration $\tilde{n}$ and molecular mass $\tilde{\mu}$ in neutrally-stratified media is given by (Mange, 1957):

$$w = -D\left[\frac{1}{\tilde{\mu}}\frac{\partial\tilde{\mu}}{\partial z} + \left(\frac{\tilde{\mu}}{\mu} - 1\right)\frac{\mu g}{kT}\right], \tag{4} \quad \{\mathrm{eq:dif}$$

where $\mu$ is molecular masses of air, $g$ – acceleration due to gravity, $k$ is the Boltzmann constant, and $T$ is temperature. With ideal gas law $p = nkT$, in which $p$ is pressure, and $n$ is number concentration, and static law $dp/dz = -g\rho$, where $\rho = \mu n$ is the air density, the equation (4) can be reformulated in terms of the admixture mixing ratio $\xi = \tilde{n}/n$ and pressure. Then the vertical gradient of the equilibrium mixing ratio will be:

$$\frac{\partial \xi}{\partial p} = \left( \frac{\tilde{\mu}}{\mu} - 1 \right) \frac{\xi}{p}. \tag{5}$$ {eq:mol

It is non-zero for an admixture of a molecular mass different from one of the air. Integrating the gradient (5) over vertical, one can obtain that equilibrium mixing ratios $\xi_1$ and $\xi_2$ at two levels with corresponding pressures $p_1$ and $p_2$ are related as:

$$\frac{\xi_1}{\xi_2} = \left( \frac{p_1}{p_2} \right)^{\tilde{\mu}/\mu - 1}. \tag{6}$$ {xi1xi2

For heavy admixtures, such as $SF_6$ ($\tilde{\mu} = 0.146$ kg/mole) the equilibrium gradient of a mixing ratio is substantial. For example, the difference of equilibrium mixing ratio of $SF_6$ in the atmosphere between $0.1$ and $0.2$ hPa is a factor of 16.

In most of the atmosphere, the effect of gravitational separation is insignificant due to the overwhelming effect of other mixing mechanisms, whereas in the upper stratosphere the molecular diffusivity may become significant. Therefore, in the upper stratosphere heavy gases can no longer be considered as tracers and the molecular diffusion should be treated explicitly. The effect of gravitational separation of nitrogen and oxygen isotopes in the stratosphere has been observed (Ishidoya et al., 2008, 2013; Sugawara et al., 2018), however for isotopes the ratio of masses is relatively small, so the observed differences
were also small (up to $10^{-5}$). For $SF_6$ the molecular mass difference is much larger.

In order to enable the gravitational separation in SILAM we have introduced a molecular diffusion mechanism, which can be enabled along with the turbulent diffusion scheme. The exchange coefficients due to molecular diffusion between the model layers are pre-calculated according to Eq. (4) discretized for the given layer structure for each species according to its diffusivity and molar mass. The US standard atmosphere (NOAA et al., 1976) was assumed for vertical profiles of temperature and air
density during pre-calculation of the exchange coefficients. The exchange has been applied throughout the domain at every model time step with a simple explicit scheme.

### 3.4  Parametrization for destruction of $SF_6$ in the mesosphere

{sec:SF

As it has been mentioned above, the topmost level of the ERA-Interim meteorological data set is located at $0.1$ hPa, which is below the layer where the destruction of $SF_6$ occurs. Therefore we have to put a boundary condition to our simulations
to account for the upward flux of $SF_6$ through the upper boundary of the simulation domain. For that we assume that $SF_6$ distribution above the computational domain is in equilibrium with destruction and vertical flux.

Assuming the profiles for $K_z(p)$ and the $SF_6$ lifetime $\tau(p)$ are given by (2) and (1), one can obtain a steady-state distribution of mass-mixing ratio $\xi$ of $SF_6$ due to destruction in the mesosphere at any point where both (2) and (1) are valid and vertical advection is negligible. The latter assumption implies that the diffusive vertical flux overwhelms the advective one. The validity
and implications of neglecting the regular vertical transport are discussed below. The steady-state profile of $\xi$ can be obtained

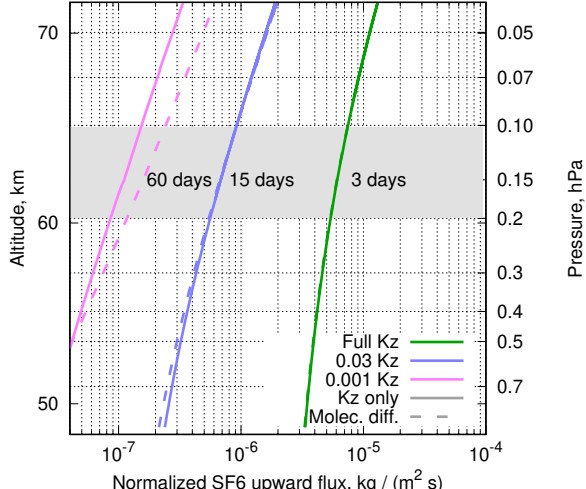

**Figure 3.** Vertical profiles of steady-state upward flux of $SF_6$ normalized with mass mixing ratio $F(p)/\xi(p)$, for eddy diffusivity and lifetime profiles given by (2) and (1). The upper model layer of SILAM and effective lifetimes of $SF_6$ there due to the destruction in the mesosphere for different Kz profiles are given.                                                                                    {fig:in

from a solution of a steady-state diffusion equation with a sink:

$$\frac{\partial \xi}{\partial t} = g\frac{\partial}{\partial p}(F) - \frac{\xi}{\tau(p)} = 0, \tag{7} \quad \{eq:dif$$

where $\rho(p)$ is air density, and $g$ is acceleration due to gravity, and the upward flux of $SF_6$ is given by

$$F(p) = g\rho^2 K_z(p)\frac{\partial \xi}{\partial p} \tag{8} \quad \{eq:dif$$

The above equation was solved numerically as a boundary value problem with unit mixing ratio at a height of $1\,\mathrm{hPa}$ and vanishing flux $F(p)$ at $p = 0$ for the set of Kz profiles. The shooting method was used together with bisection to get the steady-state profiles of $\xi(p)$ and $F(p)$, corresponding to $\xi(1\,\mathrm{hPa}) = 1$. For all considered cases the flux $F(p)$ decreased by several orders of magnitude already at the level of a few Pa, i.e. below the maximum of the depletion profile of Totterdill et al. (2015), indicating that particular shape of $\tau(p)$ above that level does not influence the fluxes at the domain top ($0.1\,\mathrm{hPa}$). The

steady-state upward flux of $SF_6$ $F(p)$ normalized with corresponding mixing ratio at each pressure $F(p)/\xi(p)$, for the three test profiles of $K_z$ is shown in Fig. 3 with solid lines.

    The gravitational separation can be accounted for by introducing into the vertical flux Eq. (8) a term responsible for molecular diffusion and its equilibrium state (5):

$$F(p) = g\rho^2 K_z(p)\frac{\partial \xi}{\partial p} + g\rho^2 D(p)\left(\frac{\partial \xi}{\partial p} - \frac{\tilde{\mu} - \mu}{\mu}\frac{\xi}{p}\right) \tag{9} \quad \{eq:dif$$

The profiles of $F(p)/\xi(p)$ resulting from this term $F(p)$ in the equation (7 are given in Fig. 3 with dashed lines. The magnitude of $F(p)/\xi(p)$ gives an equivalent regular vertical air-mass flux that would result in the same vertical flux of $SF_6$ if it were

passive and non-diffusive. The equivalent regular vertical velocity $\omega_{eq}$ (in units of the Lagrangian tendency of a parcel pressure due to vertical advection) can be expressed as:

$$\omega_{eq} = -gF(p)/\xi(p). \tag{10}$$ {eq:ome

Accounting for molecular diffusion may either enhance or reduce the upward flux of $SF_6$ in the model. Along with setting the equilibrium state with the bulk of a heavy admixture being in the lower layers, molecular diffusion provides additional means for transport to the upper layers where the destruction occurs. For very low eddy diffusivities, the molecular diffusion is a sole mechanism of upward transport of $SF_6$ towards depletion layers. For higher eddy diffusivity the effect of molecular diffusion and gravitational separation becomes negligible.

For a model consisting of stacked well-mixed finite layers, the loss of $SF_6$ from the topmost layer due to the steady upward flux would be proportional to the $SF_6$ mixing ratio in the layer. This loss of mass is equivalent to a linear decay of $SF_6$ in the layer at a rate

$$\tau^{-1} = g\frac{F(p)}{\xi(p)\Delta p}, \tag{11}$$ {eq:inv

where $\Delta p$ is a pressure drop in the layer.

For the upper layer of our simulations (between 0.1 hPa and 0.2 hPa, grey rectangle in Fig. 3), and $K_z(p)$ given with Eq. (2), the $SF_6$ lifetime $\tau$ due to turbulent diffusion is about 3 days. After scaling the $K_z(p)$ profile with factors of 0.03, and 0.001 one gets the lifetimes of 15 and 60 days respectively. Noteworthy, the molecular diffusion sets the upper limit to the $SF_6$ lifetime in the topmost model layer: it can not be longer than 60 days for the 0.1 - 0.2 hPa layer. Close to this regime, the system becomes insensitive to the actual profile and values of the turbulent diffusion coefficient. The loss of $SF_6$ through the domain top was
implemented as a linear decay of $SF_6$ in the topmost model layer, at a rate corresponding to the $K_z(p)$ profile used in each simulation.

### 3.5  Simulation setup

{sec:se

The simulations of atmospheric transport were performed with the SILAM model for 1980-2018 years on a 1.44x1.44 degree global grid with 60 hybrid sigma-pressure layers starting from surface, with the uppermost layer between pressures of 0.1 and
0.2 hPa. The model time step of 15 minutes was used and the output of daily mean concentrations of tracers together with air density was arranged.

The simulations were driven with ERA-interim meteorology at 0.72-degree resolution, so the meteorological input for both cell-interface for winds, and cell mid-points for other parameters (surface pressure, temperature and humidity) was available without further interpolation. The gridded ERA-interim fields are, however, a result of reprojection of the original meteoro-
logical fields from spherical harmonics. Moreover, differences in the representation of model vertical structure between IFS and SILAM make a vertical reprojection necessary. These reprojections together with a limited precision of the gridded fields and inevitable small differences in physical parametrizations between IFS and SILAM result in inconsistency between surface-pressure tendencies and vertically-integrated air-mass fluxes calculated from the meteorological fields in SILAM. Such inconsistencies cause spurious variations in wind-field divergence that on long-term run result in accumulation of errors in tracer

mixing ratios, and consequently, in the simulated AoA. Therefore, horizontal wind fields were adjusted by distributing the residuals of pressure tendency and vertically-integrated horizontal air-mass fluxes as a correction to the horizontal winds following the procedure suggested by Heimann and Keeling (1989). The correction is of the order of centimeters per second, which is comparable to the precision of the input wind fields. The vertical wind component was then re-diagnosed from a divergence of horizontal air-mass fluxes for individual SILAM layers as described in Sofiev et al. (2015).

SILAM performs 3D transport by means of a dimension split: transport along each dimension is performed separately as 1D transport. To minimize the inconsistency between the tracer transport and air-mass fluxes, caused by the dimension split at finite time step, the splitting sequence has been inverted at each time step to reduce the accumulation of errors. The residual inconsistency was resolved by using a separate unity tracer, which was initialized to the constant mass mixing ratio of 1 at the beginning of a simulation. If advection was perfect, the concentration of unity would be equivalent to air density (mixing ratio would stay equal to 1). The mixing ratios of simulated tracers were then evaluated as a ratio of a tracer mass in a cell to the mass of unity.

In order to assess the effects of gravitational separation and destruction on the atmospheric distribution of $SF_6$, we have used four tracers: $SF_6$ as a passive tracer "sf6pass", $SF_6$ with gravitational separation but no destruction "sf6nochem" (no chemistry), $SF_6$ with destruction but no gravitational separation "sf6nograv", and $SF_6$ with both gravitational separation and destruction in the upper model level "sf6".

All $SF_6$ tracers had the same emission according to the $SF_6$ emission inventory (Rigby et al., 2010). The inventory covers 1970-2008, and was extrapolated with a linearly growing trend of $0.294\,\mathrm{Gg/y/y}$ until July 2016. The last 2.5 years were run without $SF_6$ emissions to evaluate its destruction rate. Note, that the emission extrapolation gives for 2016 $9.4\,\mathrm{Gg/y}$, which is somewhat higher than later estimate $8.8\,\mathrm{Gg/y}$ (Engel et al., 2018).

Besides the four $SF_6$ tracers we have used a "passive" tracer emitted uniformly at the surface at constant rate during the whole simulation time and an "ideal age" tracer. The "ideal age" tracer is defined as a tracer whose mixing ratio $\xi_{ia}$ obeys continuity equation (Waugh and Hall, 2002):

$$\frac{\partial \xi_{ia}}{\partial t} + \mathcal{L}(\xi_{ia}) = 1, \tag{12}$$

(where $\mathcal{L}$ is an advection-diffusion operator), and boundary condition $\xi_{ia} = 0$ at the surface. The "ideal age" tracer is transported as a regular gaseous tracer, and to maintain consistency with other tracer mixing ratios, the ideal age is updated at every model time step $\Delta t$ using the unity tracer:

$$M_{ia} \mapsto \begin{cases} 0, & \text{at lowest layer,} \\ M_{ia} + M_{\text{unity}}\Delta t, & \text{otherwise,} \end{cases} \tag{13}$$

where $M_{ia}$ and $M_{\text{unity}}$ are masses of the "ideal age" tracer and of the unity tracer in a grid cell. The mixing ratio of the "ideal age" tracer is a direct measure of the mean age of air in a cell, so the tracer is a direct Eulerian analog of the time-tagged Lagrangian particles with clock reset at the surface. Note that the AoA derived from the "ideal age" tracer and AoA from a passive tracer with a linearly-growing near-surface mixing ratio are equivalent (Waugh and Hall, 2002), and implementation of both provides a redundancy needed to ensure self-consistency of our results.

A set of the simulations was performed with four settings for the eddy diffusivity profile within the model domain, described in Sec. 3.2 and corresponding destruction rates of "sf6" and "sf6nograv" tracers in the uppermost model layer. All runs were
initialized with the mixing ratios from the final state of a special initialization run. The initialization simulation with "0.1Kz" eddy diffusivity was started from 1970 with zero fields for all tracers, except for unity tracer that was set to unity mixing ratio. The simulation was run with 1970-1989 emissions for $SF_6$ species from the same inventory as for the main runs (Rigby et al., 2010), and driven with twice repeated ERA-Interim meteorological fields for 1980-1989. The mixing ratios of all $SF_6$ tracers at the end of the initialization run were scaled to match the total $SF_6$ burden of $20.17\,Gg$ in 1980 (Levin et al., 2010).

**4   Sensitivity and validation of $SF_6$ simulations**

{sec:va

**4.1   Gravitational separation and mesospheric depletion**

{sec:gr

To evaluate the relative importance of gravitational separation and mesospheric depletion and their effect on the $SF_6$ concentrations we have compared the simulations for various $SF_6$ tracers and evaluated the relative reduction of $SF_6$ content in the stratosphere due to these processes. As a conservative estimate of the reduction, we evaluated the relative differences between
the tracers in the latitude belt of 70-85S, since both processes have the most pronounced effect in southern polar vortex, where the downwelling of Brewer-Dobson circulation is the strongest.

Hereafter we quantify the effect of a relative difference between atmospheric contents of two $SF_6$ tracers "X" and "Y" defined as:

$$\Delta(\text{"X"},\text{"Y"}) = 2\frac{\xi_X - \xi_Y}{\xi_X + \xi_Y} \cdot 100\% \tag{14}$$   {eq:rel

The relative differences for the $SF_6$ tracers in the Southern polar region (70-85S) simulated with two extreme models for $K_z$ is given in Fig. 4 as a function of time and altitude. Noteworthy, every 5% of decrease of $SF_6$ with respect to its passive counterpart correspond to about one year of a positive bias in AoA derived from $SF_6$ mixing ratios.

The reduction of $SF_6$ content due to gravitational separation if the mesospheric depletion is disabled is given by the relative difference of "sf6nochem" and "sf6pass" (Fig. 4ab). Expectedly, the effect of gravitational separation is most pronounced for
the case of low eddy diffusivity ("0.001 Kz"), and the reduction of $SF_6$ in the altitude range of 30–50 km reaches $2 - 5$ %. In the case of strong mixing, the effect of separation is about 1 %.

The reduction of $SF_6$ content due to gravitational separation in presence of stratospheric depletion given by the relative difference of "sf6nograv" and "sf6" tracers. The effect of the separation for low $K_z$ is very similar between depletion and no-depletion case (Fig. 4c vs. Fig. 4a). Depletion reduces the effect of the gravitational separation for high $K_z$ (Fig. 4b vs Fig. 4d).
Regardless depletion, stronger $K_z$ reduces the effect of the gravitational separation, however the latter is still non-negligible if precisions of order of a month for AoA are required.

MAS: Replace with "sf6pass" vs "sf6nograv"  Roux: Then the picture will be identical to $\Delta(\text{"sf6pass"},\text{"sf6"})$.

The combined effect of depletion and gravitational separation is seen in the relative difference of "sf6pass" and "sf6" tracers (Fig. 4e and 4f). For both $K_z$ cases the effect of depletion is stronger than diffusive separation by more than one order of

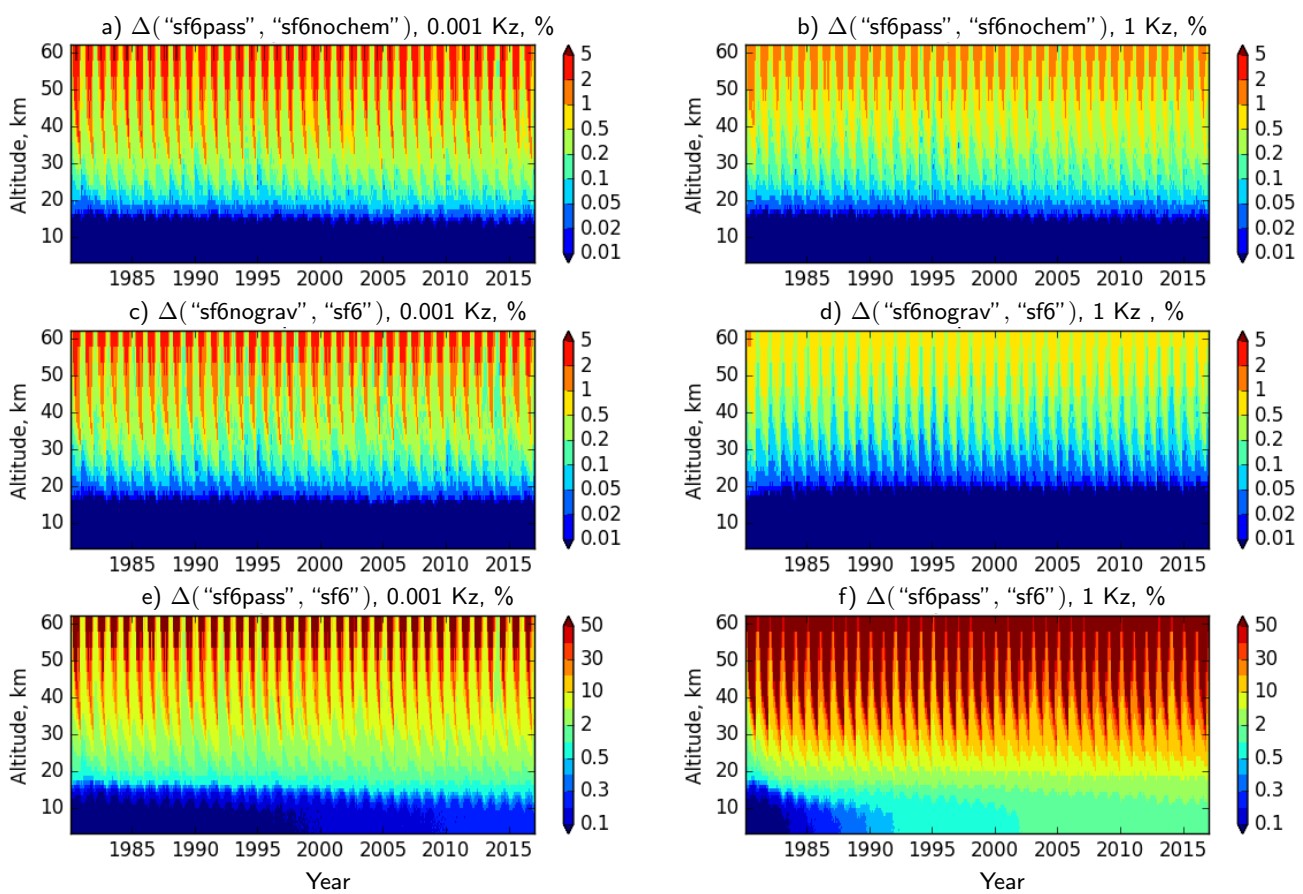

**Figure 4.** The relative reduction of $SF_6$ content (in %) at 70-85S due to gravitational separation with (a, b) and without (c, d) depletion, and due to combined effect of depletion and separation (e, f) at two extreme Kz cases. Note different color scales for e) and f).                {fig:de

magnitude. Regardless of the $K_z$ profiles used, the reduction exceeds 50 %, which roughly corresponds to 10 years of an offset in the apparent AoA.

In all cases the reduction of the $SF_6$ content has strong annual cycle associated with the cycle of downwelling in winter and upwelling in summer. Besides that reduction has a noticeable inter-annual variability that poses substantial difficulties on applying a consistent correction to the apparent AoA. Contrary to the former two comparisons, strong eddy mixing leads to

strong reduction of $SF_6$ since it intensifies the transport to the depletion layers, and thus enhances the depletion rate.

The simulations for different $K_z$ have been initialized with the same state obtained from a separate spin-up simulation with "0.01 Kz", which was scaled to match total burden of $SF_6$ in 1980. Thus a relaxation of the $SF_6$ vertical distribution during the first few years of the simulations is clearly seen in Fig. 4. For "1 Kz" case (4f) the gradual increase of the difference between $SF_6$ and its passive version in the troposphere can be seen. The rate of this increase is about 0.5% per 39 years of simulations.

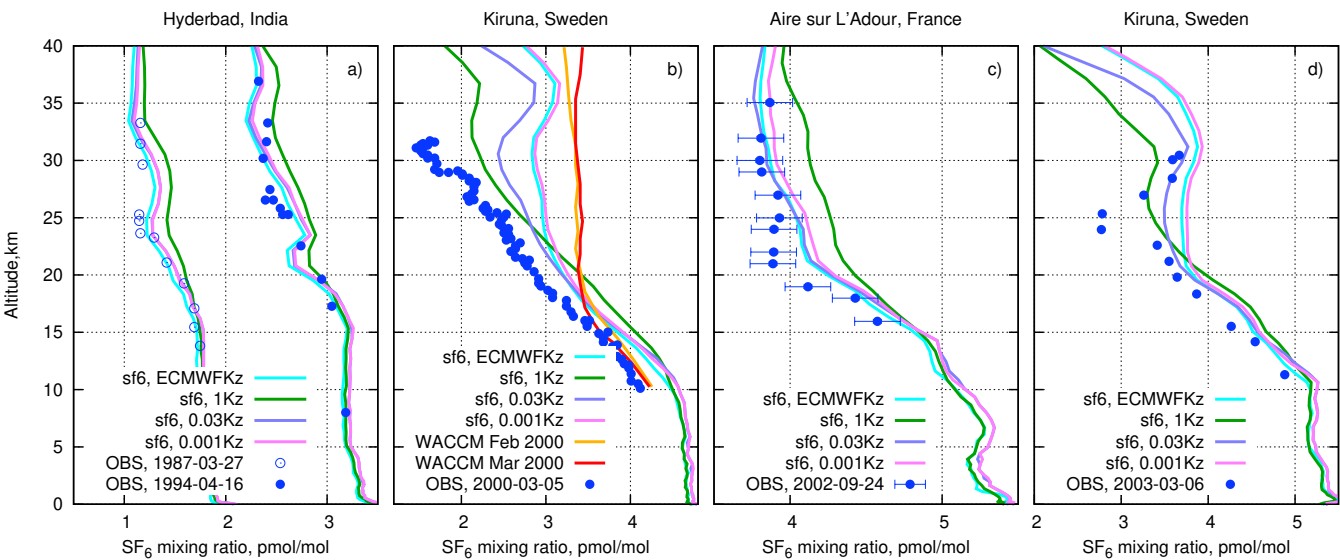

**Figure 5.** Observed $SF_6$ balloon profiles and corresponding daily-mean SILAM profiles for the date of observations. The observational data obtained from Patra et al. (1997), Ray et al. (2017), Ray et al. (2014), and Engel et al. (2006) for panels a–d correspondingly. The model profiles from WACCM model are from Ray et al. (2017).

{fig:ba

This rate should not be confused with the depletion rate of $SF_6$ in the atmosphere since the difference is a combined effect of depletion and growth of emission rate, despite the latter is exactly the same for both tracers.

The above comparison indicates that the depletion has the stronger effect on the distribution of the $SF_6$ mixing ratio in the upper stratosphere than gravitational separation and molecular diffusion. However, the important role of the molecular diffusion in the model is that it maintains the upward flux towards the mesosphere in the simulations even if the eddy diffusivity ceases.

Further in this paper only the "sf6pass" and "sf6' tracers will be used.

### 4.2  Evaluation against balloon profiles

{sec:ba

The tropospheric concentrations of $SF_6$ in our simulations have been insensitive to $SF_6$ destruction or to the choice of the eddy diffusivity profile in the stratosphere. The difference in the modelled profiles can however be seen above the tropopause. For comparison we took the simulations with prescribed eddy diffusivity in stratosphere (1Kz, 0.03Kz, and 0.001Kz, see Sec. 3.2),

and with dynamic eddy diffusivity "ECMWF Kz". The simulations were matched with stratospheric balloon observations ( Fig. 5) published by Patra et al. (1997); Engel et al. (2006); Ray et al. (2014, 2017).

Two balloon profiles observed at Hyderbad (17.5N,78.6E) in 1987 and 1994 by Patra et al. (1997) indicate an increase of $SF_6$ content during the time between the soundings (Fig. 5a). Both profiles have a clear transition layer from tropopause at $\sim 17$ km to undisturbed upper stratosphere above $\sim 25$ km. The simulated profiles agree quite well to the observed profiles,

except for the most diffusive case that gave notably smoother profiles and somewhat overstated $SF_6$ mixing ratios due to too strong upward transport by the diffusion through the tropopause and in the lower stratosphere.

The profile in Fig. 5b has been obtained from Kiruna (68N, 21E) in early spring 2000 during the SAGE III Ozone Loss and Validation Experiment, SOLVE, (Ray et al., 2002) with the Lightweight Airborne Chromatograph (Moore et al., 2003). The profile is affected by the polar vortex and clearly indicates a strong reduction of $SF_6$ with height with a pronounced local

minimum at 32 km. The corresponding SILAM profiles tend to overestimate the $SF_6$ vmr. The $SF_6$ profiles for "ECMWF Kz" and "0.001Kz" match each other, since vertical mixing is negligible in both cases. The most diffusive profile "1Kz" has the strongest depletion in the upper part, but the largest deviation from the observations below 20 km. The intermediate-diffusion profile ("0.03Kz") is almost as close to observations as the non-diffusive profile. Moreover, the "0.03Kz" profile has a minimum at the same altitude as the observed one, albeit the modelled minimum is substantially less deep.

For comparison, Fig. 5b also contains monthly-mean profiles from the WACCM simulations of Ray et al. (2017) along with the observation data. The WACCM profiles match very well the observations below 17 km, but turn nearly constant above, thus under-representing the depletion of $SF_6$ inside the polar vortex. Monthly-mean SILAM profiles (not shown) were much closer to plotted daily profiles than to monthly WACCM ones. Note, that the version of WACCM, used for the simulations did not include the electron attachment mechanism.

For the mid-latitude profile in Fig. 5c from Aire-sur-l'Adour, France (43.7N,0.3W), all SILAM profiles except for "1Kz" fall within the observational error bars provided together with the data by Ray et al. (2017). Similar to the Kiruna case in Fig. 5b, the SILAM profiles are much smoother than the observed ones and are unable to reproduce the sharp transition at 20 km.

Another profile from within the polar vortex (Fig. 5d) was observed at the same Kiruna site as the one in Fig. 5b, but three years later. The observed profile also has a minimum that is much deeper than in the modelled profiles. Similar to the case in

Fig. 5b, the "0.03Kz" profile is the only one that has a pronounced minimum at the same altitude as the observed one. The minimum is a result of the spring breakdown of the polar vortex, when a regular down draught ceases, and atmospheric layers decouple from each other. The reduced depth of the modelled minimum is probably caused by insufficient decoupling of the layers in the driving meteorology.

In all above cases, the "1Kz" profile is clearly far too diffusive: in non-polar cases and is an outlier that is furthest from

the observations, whereas for Kiruna cases it overstates the lower part of the profiles and smears the vertical structure of the profiles further away form the tropopause. The $SF_6$ profiles simulated with "ECMWF Kz" and "0.001Kz" match each other in all simulations, since vertical mixing is negligible in both cases. The $SF_6$ resulting from "0.03Kz" case appear to be most realistic out of the four considered simulations: they are close to observed ones and have local minima at right altitudes for both Kiruna profiles.

### 4.3  Evaluation of $SF_6$ against MIPAS data

The MIPAS observations provide the richest observational dataset for stratospheric $SF_6$. However, each individual observation has a substantial retrieval noise error, which is noticeably larger than the difference between observation and any of the SILAM simulations. The largest diversity of the modelled $SF_6$ profiles was observed in polar regions, therefore below we show the

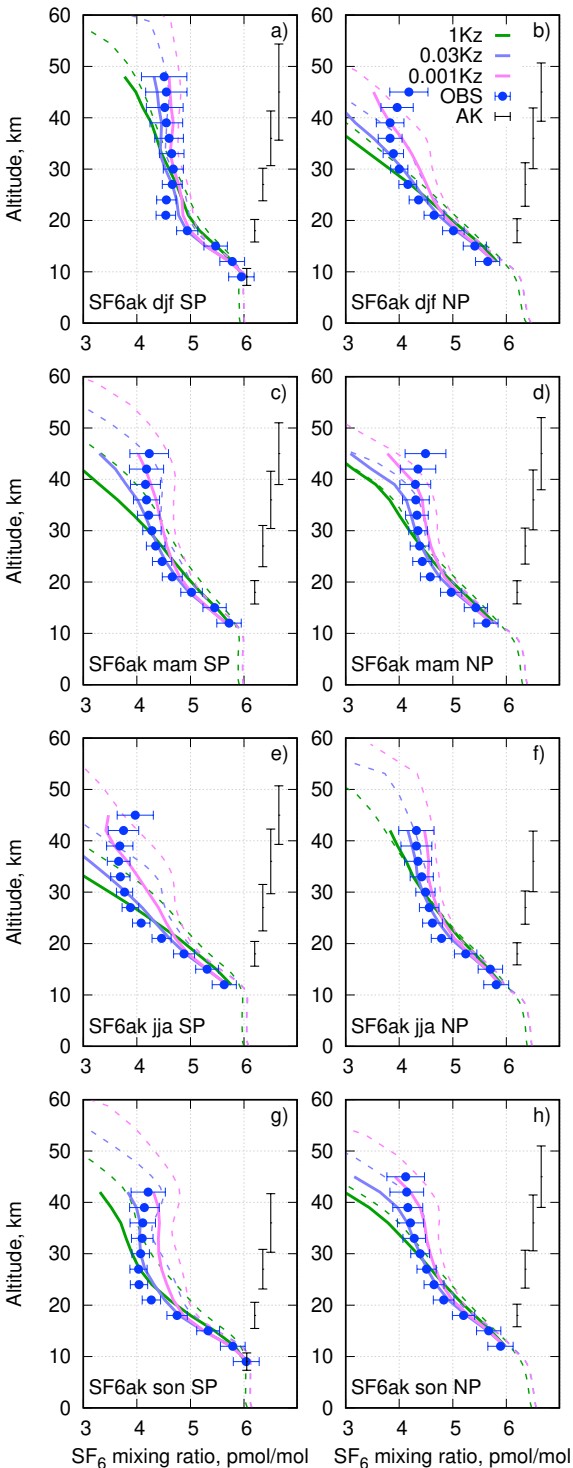

**Figure 6.** Seasonal mean collocated SILAM $SF_6$ and MIPAS profiles for 2007, for southern and northern polar regions. Typical ranges covering 75% of averaging kernel are given with error bars at the right-hand side of each panel. The horizontal error bars indicate systematic uncertainties of the observations that are fully correlated among profiles and do not cancel out when averaging over a large number of measurements.

{fig:MI

mean profiles for each season in southern and northern polar areas. Besides that, we consider statistics of the model performance

against MIPAS measurements in lower and upper stratosphere separately. For simplicity, we do not show the statistics for the
"ECMWF Kz" runs, since it is very similar to one for "0.001Kz".

For the comparison, the daily mean model profiles were collocated to the observed ones in space and time, and then, an averaging kernel of the corresponding MIPAS profile was applied to the SILAM profile. For the comparison we took only the data points with all the following criteria met:

– MIPAS visibility flag equals 1

– MIPAS Averaging kernel diagonal elements exceed 0.03

– MIPAS retrieval vertical resolution, i.e. full-width at half-maximum of the row of the averaging kernel, is better than 20 km

– MIPAS volume mixing ratio noise error of $SF_6$ is less than $3 \, \mathrm{pmol/mol}$

The mean seasonal profiles of the $SF_6$ mixing ratio for southern and northern polar regions derived from the MIPAS observations and SILAM simulations for 2007 are given in Fig. 6. In order to facilitate the comparison of our evaluation with earlier study by Kovács et al. (2017) we have chosen the same year and same layout of the panels as Fig. 3 there. The main differences between Kovács et al. (2017) and current evaluation are:

– We used averages of collocated model profiles (bold lines). The non-collocated seasonal- and area-mean model profiles
are given with thin dashed lines for comparison.

– we use a newer version of MIPAS $SF_6$ data with considerably larger values (up to 0.6 pptv) in the upper stratosphere, compared to the version that was used by Kovács et al. (2017).

– The horizontal error bars for the observed data indicate the systematic error component that is fully correlated among the profiles and does not cancel out by averaging, or, in other words, the estimate of a possible bias, as analysed by Stiller
et al. (2008). These errors are in the order of 4% (below 30 km) up to 10% (at 60 km). The contribution of retrieval noise error is essentially negligible due to averaging. The error bars shown by Kovács et al. (2017) are noticeably larger, probably indicating that they are for individual observed values, rather than uncertainties of the mean.

– We use 3-km vertical bins for the profiles to make the points in MIPAS profiles distinguishable

– We also plot the vertical extent of the averaging kernels corresponding to their half-width.

First of all, note a substantial difference between collocated and non-collocated model profiles. The difference is caused by uneven sampling of the atmosphere by the satellite both in space and time. In particular, MIPAS, being a polar-orbiting instrument, makes more profiles per unit area closer to the pole than further away. The difference gets somewhat reduced if one uses equal weights for all model grid cells instead of area-weighted averaging, especially for wide latitude belts. The major

difference comes probably from the inability of MIPAS to retrieve $SF_6$ profiles in presence of polar stratospheric clouds that

clutter lower layers of the stratosphere and make the sampling of polar regions quite uneven both in time and in vertical. This

hypothesis agrees with fact that the difference is most pronounced for the winter pole, especially for the south pole in JJA, and

almost invisible at a summer pole.

The comparison in Fig. 6 shows that the profiles from the SILAM simulations agree quite well to the observations in the

altitude range below $20 - 25\,km$, with the most diffusive "1Kz" slightly overestimating the $SF_6$ mixing ratios. In the range

above $25\,km$, the '1Kz' profiles indicate too fast decrease of $SF_6$ with altitude. The "0.03Kz" profiles give the best results up

to $\sim 40\,km$, except for south pole in JJA and North pole in DJF.

An interesting feature of the winter-pole MIPAS profiles is an increase of the $SF_6$ mixing ratio above $40\,km$. This increase

might have been caused by issues with retrievals, as the systematic errors of the retrievals increase with altitude. Note, however,

that non-monotonous profiles can occur due to the mean atmospheric dynamics (see the non-collocated 0.001Kz profile in

Fig. 6g).

None of the model setups is capable of reproducing adequately the observations above 40 km. Wintertime poles also pose

a problem to the model. The disagreement indicates a deficiency in the model representation of air flows in the upper part of

the domain caused by insufficient vertical resolution of ERA-Interim in the upper stratosphere and lower mesosphere, and lack

of pole-to-pole circulation. This discrepancy is in line with the comparisons in Fig. 5 for polar regions. The model tends to

overstate the $SF_6$ content in the lower part of a polar vortex, and understate it above $40\,km$.

As a more extensive verification of the SILAM simulations we computed statistical scores of the simulated $SF_6$ mixing ratios

for each month of the MIPAS mission. The statistics were computed separately for the altitude range of $10 - 35\,km$ (Fig. 7)

and $30 - 60\,km$ (Fig. 8). As the difference in the statistical scores between the three selected simulations shown in the two

figures is quite minor, in addition to the aforementioned selection criteria for MIPAS data, we have selected only observations

with the retrieval target noise error below $1\,pmol\,mol^{-1}$.

The root-mean square error of is mostly controlled by the bias, and does not allow for clear distinction between the simulated

cases. In order to disentangle the effect of bias, we have calculated the standard deviation of model-measurement difference

(STD), absolute bias, and normalised mean bias (NMB):

$$\text{STD(ppt)} = \left\langle (M - \langle M \rangle - O + \langle O \rangle)^2 \right\rangle^{1/2}, \tag{15}$$

$$\text{Bias(ppt)} = \langle M - O \rangle, \tag{16}$$

$$\text{NMB(\%)} = 2 \left\langle \frac{M - O}{M + O} \right\rangle \cdot 100\%, \tag{17} \quad \texttt{\{eq:NMB}$$

where $M$ and $O$ are modelled and observed values, respectively, and $\langle \cdot \rangle$ denotes averaging over the selected model-observation

pairs for the given range of times and altitudes. Along with the STD, we have plotted the RMS error of the observations due to

retrieval noise in the original MIPAS data, labeled as "MIPAS noise" in the top panels of Fig. 7 and Fig. 8.

In the altitude range of $10 - 35$ km, the STD of model-measurement difference is uniform in time with minor peaks in

August-September (Fig. 7). The level of the noise error constitutes about 85 % of the total model-measurement difference.

| Tracer/ | loss rate, | lifetime, |
|---|---|---|
| Kz scheme | $10^3$ mol/year | years |
| passive, any Kz | 0 | $\infty$ |
| $SF_6$, ECMWF Kz | 440 | 2900 |
| $SF_6$, 0.001 Kz | 480 | 2600 |
| $SF_6$, 0.01 Kz | 760 | 1700 |
| $SF_6$, 0.03 Kz | 800 | 1540 |
| $SF_6$, 0.1 Kz | 960 | 1300 |
| $SF_6$, 1 Kz | 2160 | 590 |

**Table 1.** $SF_6$ destruction rate after stopping emissions. Mid-2011 atmospheric burden of $SF_6$ of $1.27 \cdot 10^9$ moles is used as reference for the lifetime estimate

{tab:li

Application of averaging kernel to the model profiles reduces the STD. The intermediate-diffusivity case "0.03Kz" clearly shows the least STD uniformly over the whole observation period, the same case indicates the least absolute bias.

In the range of 30 – 60 km altitudes (Fig. 8) the level of retrieval noise is noticeably higher than for the lower stratosphere. Unlike in the lower stratosphere, the least biased case is "1Kz", which has the largest STD. The STDs of "0.03Kz" and "0.001Kz" are on par, however the latter has the strongest biases. Thus for this altitude range the intermediate-diffusivity case also shows the best performance.

### 4.4 Lifetime of $SF_6$ in the atmosphere

In order to estimate the atmospheric lifetime of $SF_6$ we turned off the emission of all simulated $SF_6$ tracers in July 2016 and let the model run until the end of 2018 without emissions (Fig. 9). The decrease of the simulated $SF_6$ burden after the emission stop can be used to estimate the $SF_6$ removal rate from the atmosphere.

Time series of the total burden of $SF_6$ in the atmosphere in the simulations are given in Fig. 9. For easier comparison to observed mixing ratios the burden has been normalised with $1.78 \cdot 10^{20}$ moles – the total amount of air in the atmosphere – to get the mean mixing ratio. The tabulated values for the atmospheric burden of $SF_6$ from Levin et al. (2010) and Rigby et al. (2010) are given for comparison. Since the removal of $SF_6$ from the atmosphere is mostly controlled by the transport towards the depletion layer, the vertical exchange is a key controlling factor. In all simulated cases, the removal of $SF_6$ from the atmosphere is very slow, so the relative difference between the cases is small. Similar rates could have been obtained by averaging the inverse destruction rate mass-weighted over the entire atmosphere.

The decrease of the atmospheric $SF_6$ content after the emission stop, is given at the zoom panel of Fig. 9. As expected, after July 2016 the content of passive $SF_6$ stays constant, while $SF_6$ that undergoes chemical destruction begins to decrease at a rate that depends on the transport properties of the stratosphere in the simulations, with faster removal for stronger eddy diffusivity. The removal rate is driven by the $SF_6$ content in the upper stratosphere, which is not in equilibrium with total atmospheric $SF_6$ content. A typical delay between $SF_6$ mixing ratio in the troposphere, where most of $SF_6$ resides, and the upper stratosphere,

from where $SF_6$ escapes further to the depletion layers, i.e. AoA in the topmost model layer, is about 5-6 years. Hence, to
estimate the $SF_6$ lifetimes we used the total amount of atmospheric $SF_6$ 5 years before the emission stop, i.e. $1.23 \times 10^9$ mol,
which corresponds to mean mixing ratio of about $7 \, \mathrm{pmol/mol}$. Dividing the destruction rate with the reference amount one
gets the range of corresponding simulated $SF_6$ life times in the atmosphere: 600 to 2900 years. Despite the range of assumed
diffusivities is three orders of magnitude, the loss rate of $SF_6$ varies within a factor of five ( Table 1).

The term "life time" implies a linear decay, however, due to emissions the distribution of $SF_6$ in the atmosphere is far from
equilibrium, so the decay is not proportional to the burden. A more accurate way to estimate life time would be to perform a
multi-decade simulation without sources, to get the distribution of $SF_6$ into a quasi-equilibrium with the mesospheric sink. In
such a quasi-equilibrium a model of linear decay of $SF_6$ in the whole atmosphere becomes applicable, and the life time can be
estimated as a simple ratio of burden to the loss rate. The uncertainty in the equilibrium burden corresponding to the modelled
loss rates in Table 1 can be estimated as the range of AoA in the upper stratosphere ($\sim 0.5$ years) divided by the growth rate of
the burden ($0.04 \, \mathrm{year}^{-1}$), i.e about 2%. The major uncertainty comes from the over-simplistic parametrization of the $SF_6$ loss
in the model, which is more difficult to quantify.

The best-performing in terms of $SF_6$ simulation resulted in 1540 years lifetime. Given the uncertainties above, it meets the
ranges suggested by earlier studies. It is in a good agreement with the range of $800 - 3200$ years from earlier model studies
(Ravishankara et al., 1993; Morris et al., 1995), and is close to the upper bound of the 580–1400 years range recently obtained
by (Ray et al., 2017) from the balloon profile given in Fig. 5b.

Our estimate is also slightly above the range given by Kovács et al. (2017), who obtained $1120 - 1475$ years. Note, how-
ever, that in the simulations of Kovács et al. (2017) the mixing ratios of $SF_6$ in the stratosphere and lower mesosphere were
noticeably higher than retrieved from MIPAS, which is likely to cause overstating the simulated depletion rates, and lead to
corresponding low bias of the $SF_6$ lifetime from those simulations.

**5   Simulations of AoA**

**5.1   Eddy diffusivity and simulated AoA**

The effect of the vertical eddy diffusivity on AoA in the stratosphere was evaluated with the same set of three prescribed $K_z$
profiles and one dynamic $K_z$ profile, as for $SF_6$ simulations. An example of annual-mean distributions of AoA for the same
year is given in Fig. 10. The Hunten (1975) $K_z$ profile (Fig. 10a) gives AoA in the stratosphere of about 3.5 years. It is much
shorter than available estimates of stratospheric AoA (e.g. Waugh, 2009; Engel et al., 2009) from the observations of various
tracers. Three other profiles of $K_z$ result in almost identical average distribution of AoA with typical stratospheric AoA of 5.5
years, which agrees quite well with the experimental estimates. In these cases AoA is controlled by the transport with explicitly
resolved winds. Since "0.03Kz" profiles result in most realistic distribution of $SF_6$ in our simulations, in the current section we
will use simulated distributions of tracers with "0.03Kz" eddy diffusivity.

## 5.2   AoA and apparent $SF_6$ AoA

The AoA for all tracers (except for the "ideal age") was calculated as a simple time lag between a mixing ratio in a given point of the domain and the mean near-surface mixing ratio. As it has been pointed out by (Waugh and Hall, 2002), this lag equals to AoA only in case of a fully passive tracer with linearly growing (or decreasing) near-surface mixing ratio. Corrections have been applied to the AoA derived from $SF_6$ in many studies studies (Volk et al., 1997; Stiller et al., 2008, 2012; Engel et al.,

2009) to account for non-linear growth of the near-surface $SF_6$ mixing ratio and for mesospheric sink of it. The corrections rely heavily on various assumptions that can hardly be rigorously verified for the atmospheric circulation. Therefore in this study for the sake of simplicity we do not apply any corrections to the AoA derived from time lags of tracers. The corrections and assumptions behind them are discussed in Sec. 6.

The constant-rate emission of the "passive" tracer in our simulations resulted in nearly linear growth of the near-surface

mixing ratio of the tracer after a decade of spin-up. The latter makes the age derived from the "passive" tracer equivalent to the age derived from the ideal-age tracer. The resulting distributions of "passive" and ideal-age AoA are indeed very close to each other (Fig. 11 a and b). The agreement confirms the self-consistency of the transport procedure since the tracers have opposite sensitivity to the advection errors: higher mixing ratios correspond to younger air for the accumulating tracers, while for the ideal-age tracer higher mixing ratios correspond to older air. The remaining differences of are caused by spatial

inhomogeneities of near-surface mixing ratio of "passive"due to variations in the near-surface air density.

The distribution of the AoA derived from "sf6pass" (Fig. 11c) is similar to the ideal-age one, however one can see substantial differences. The negative AoA in northern troposphere for the "sf6pass" tracer is caused by the predominant location of the sources in the northern hemisphere, so the concentrations there exceed global-mean levels. The growing rate of the $SF_6$ emissions leads to the greater-than-linear increase of near-surface mixing ratios, which leads to an old bias up to 3-5 months

of the "sf6pass" AoA. This old bias has been one of the drawbacks of the $SF_6$ AoA pointed by Garcia et al. (2011).

The ages shown in Fig. 11a – c agree well with the ages derived from *in-situ* observations of $SF_6$ and $CO_2$ at the 25 km altitude by Waugh and Hall (2002). They also agree quite well with earlier simulations with five climate models that give annual mean ages in the upper stratosphere between 4.5 and 5.5 years (Butchart et al., 2010), and with Lagrangian simulations of (Diallo et al., 2012) driven by the same ERA-Interim meteorological fields as used for the present study. A substantial

disagreement, however, exists with the ages derived from the MIPAS satellite observations (Stiller et al., 2012; Haenel et al., 2015), who calculated ages exceeding 10 years in polar areas and in the upper stratosphere. The reason for the disagreement follows from the above analysis: $SF_6$ can neither be considered as a passive tracer nor does its mixing ratio in the troposphere grows linearly with time. Denoting the AoA derived from the $SF_6$ profiles as "apparent AoA" (Waugh and Hall, 2002), we calculated it from the SILAM-predicted $SF_6$ profiles, which, as shown above, agree well with AoA derived from MIPAS. The

resulting model-based apparent AoA (Fig. 11d) is indeed much older than the "ideal-age" AoA. The distribution of apparent $SF_6$ AoA agrees to the AoA from MIPAS $SF_6$ profiles by Haenel et al. (2015): well over 5 years AoA around equator with well over 10 years AoA in polar regions.

The effect of apparent over-aging in the stratosphere due to the subsidence of the mesospheric air was estimated by Stiller et al. (2012) to be a fraction of a year in the upper stratosphere. Earlier experimental balloon studies (Strunk et al., 2000) indicated up to 3.5-year difference between $CO_2$ and $SF_6$ ages. In our simulations, the over-aging due to the $SF_6$ depletion and other factors discussed in previous sections is much stronger and affects the whole stratosphere.

## 5.3   Trends in apparent AoA

Changes in AoA have been used in many studies as an indicator of changes in the atmospheric circulation. In order to evaluate the effect of the way AoA is evaluated on trends in AoA we have calculated trends in apparent AoA at different altitudes and latitudes for 11 years 2002-2012. This period roughly covers the MIPAS mission time and allows for comparison with trends reported by Haenel et al. (2015).

The zonal-mean vertical profiles of the AoA trends during 2002-2012 are shown in Fig. 12 for five latitudinal belts. The presented variable is a slope of the linear fit of deseasonalized monthly-mean time series for each tracer, averaged over the corresponding latitudinal belt and model layer. The fit was made with the ordinary least-squares method for each tracer. The error-bars show 95-% confidence intervals, calculated as if a model of linear trend with uncorrelated Gaussian noise was applicable to the time series.

The trends of the apparent AoA for the non-passive $SF_6$ species have a clear increase with height in the upper part of the profiles. The increase is the largest at high latitudes. Such a behaviour of trends agrees well with the AoA trends of Haenel et al. (2015, Fig. 7) obtained from the MIPAS observations. The over-aging due to the mesospheric depletion of $SF_6$ has been discussed and estimated by Haenel et al. (2015); Kovács et al. (2017). However, Fig. 12 shows that the mesospheric depletion of $SF_6$ also affects its trend: the over-aging increases with time. The reason is that depletion is proportional to the $SF_6$ load, which grows with time. This effect has been pointed out earlier by Stiller et al. (2012).

The apparent AoA derived with passive $SF_6$ tracer "sf6pass" indicates a negative trend of about 0.5 years/decade. The trend is caused by the temporal variation of $SF_6$ emissions. In order to get unbiased AoA estimate from a passive tracer, one needs the mixing ratio at the surface increasing linearly with time. A steady growth of emission rate at the surface leads to faster-than-linear increase of near-surface mixing ratio and, thus, low-bias of AoA since younger (i.e. more rich with $SF_6$) air gets more weight when two volumes of different age mix. According to the inventory (Levin et al., 2010) used in this study, the $SF_6$ emission rate was growing in 1997–2000 about twice slower than after 2005. Consequently, the negative bias of apparent AoA has increased resulting in negative trend if AoA in the stratosphere.

The AoA trends derived from the "ideal age" and "passive" tracers agree through the whole range of altitudes and latitudes indicating internal consistency of our simulations. The main common feature of the profiles is the negative tendency of about $-0.5$ year/decade in the altitude range of 15-30 km with a profile that varies across altitudes. Similar-magnitude trends for the same period were reported by Plöger et al. (2015), who used the same ERA-Interim to simulate AoA. The major difference between the obtained trends is that we have consistently negative trends for both hemispheres, whereas Plöger et al. (2015) indicate a positive trend of a fraction of year per decade in the altitude range of $20 - 30\,\mathrm{km}$ in the Northern hemisphere, and

similar-magnitude negative trend in the Southern hemisphere. The reason for the discrepancy despite the same input dataset deserves further investigation.

The trends might be a feature of the non-uniformity of the ERA-Interim dataset, which was produced with assimilation of an inhomogeneous set of the observations. During 2002-2012, the amount of the assimilated data on the upper-air temperatures

was by an order of magnitude higher than before 2000 and two orders of magnitude higher than after 2010 (Dee et al., 2011). It had a clear impact on the patterns of analysis increments in ERA-Interim and, consequently, on the predicted stratospheric circulation. Due to such inhomogeneities, the quality of trends derived from reanalysis data needs to be verified for each geophysical quantity (Dee et al., 2011). Deducing reliable trends for atmospheric temperature, quantity that is measurable and extensively assimilated, took a major effort (Simmons et al., 2014). The fact that AoA is not a directly observable quantity

makes the verification of the AoA trends in ERA-Interim hardly possible.

To get more insight on the nature of the simulated long-term AoA variability at different altitudes and latitudes we have plotted the time series of monthly zonal-mean ideal-age AoA for the same latitude belts as in Fig. 12 over 1990-2018 (Fig. 13). To make the temporal variations more visible, the mean AoA profile for each latitude averaged over the same period was subtracted form the profiles. One can see a clear seasonal variation of the AoA outside of the equatorial zone. The variation

has opposite phase between the upper and the lower stratosphere. In the altitude range of $20 - 30\,\mathrm{km}$, where the trends are most pronounced, the temporal variation of AoA has a ramp structure with more-or-less steady intervals and relatively quick changes. Such structure is similar to the one shown for the ERA=Interim analysis increments (Dee et al., 2011) and is likely to be caused by temporal inhomogeneities in the assimilated dataset. Therefore we do not draw any conclusion here on the actual trends of AoA but highlight that trends of the apparent AoA are strongly influenced by selected time interval, and by the

method of trends calculation.

## 6   Discussion

{sec:di

The present study has several limitations that deserve specific attention. Forced zero air flux through the domain top at $0.1\,\mathrm{hPa}$ caused distortion of the mean transport within the domain, and left diffusive transport as the only means for the upper-boundary fluxes of $SF_6$. Moreover, we used prescribed profiles of the eddy diffusivity within the domain, which also affects the results

of the simulations. In this section we evaluate the role of these distortions.

### 6.1   Distortions of air-flows

The transport procedure used in this study is done with a "hardtop" diagnostics forcing zero mass-fluxes at the domain top and forced air-mass conservation everywhere within the domain. Since the upper boundary of the domain is at 0.1 hPa, the divergence of the air flow above that level in the meteorological driver is compensated by adjusting the divergences within the

domain. To evaluate the effect of this adjustment on the mean circulations we used the ERA-5 data set, which has the topmost level at $1 \times 10^{-3}\,\mathrm{hPa}$, as a reference. The diagnostic procedure was applied to ERA5 for two sets of vertical layers: the 61 ERA-Interim layers, same as used in $SF_6$ simulations (hereafter ERA5-cut), and a refined vertical matching 137 native ERA5

vertical layers (hereafter ERA5). The resulting vertical winds were compared to the ones used in the $SF_6$ simulations: 61 ERA-Interim layers diagnosed from ERA-Interim. The seasonal and zonal-mean vertical air-mass fluxes, expressed in units of

Pa/day for the three cases and two solstice seasons of 2017 are shown in Fig. 14 together with corresponding layer boundaries.

    The wind patterns in ERA5 (Fig. 14abde have finer features than in ERA-Interim due to the higher horizontal resolution of the former. The difference between the ERA5 and ERA5-cut vertical winds is strongest at the cut domain top (0.1 hPa, 65 km), where zero vertical air-mass flux is forced. For both seasons the disturbances introduced by the cut vertical to the ERA5 dataset below 55 km are minor, except for the summertime poles (South pole in Fig. 14ab, and North pole in Fig. 14de), where a

noticeable disturbance is visible down to 35-40 km altitude. Such systematic disturbances influence the performance of the AoA and $SF_6$ simulations in the polar stratosphere, and they are a probable reason for the failure of the model to reproduce the $SF_6$ profiles there (see Fig. 6).

    The comparison of the same-vertical mass-fluxes ( panels b vs. c, or e vs. f in Fig. 14) shows that the difference between ERA-Interim and ERA5 is noticeably larger than between cut- and full vertical of ERA5. Thus we conclude that the distortions

introduced by our diagnostic procedure are within the uncertainty of the input meteorological data.

## 6.2   Top-boundary mass fluxes and eddy diffusion profiles

The used modelling approach replaces the vertical transport through the domain top with diffusive fluxes for depleting $SF_6$ and a hard lid for other species. One can hope that the approach does not introduce any major disturbances into the AoA fields, since AoA is quite uniform close to the domain top. The uncertainty introduced with this approach into the $SF_6$ fields is not

straightforward to evaluate due to a major uncertainty in the vertical diffusivity profiles.

    As mentioned in Sec. 3.2, the eddy diffusivity profiles of the C-IFS model form the ERA5 reanalysis (Fig. 2) are clearly unrealistic within and above the stratosphere. They do not exhibit any growth of the eddy diffusivity in the mesosphere due to breaking gravity waves. According to Lindzen (1981) the mean diffusivity due to the breaking has an order of magnitude of $1 \times 10^2 \, \mathrm{m^2/s}$, whereas the eddy diffusion in ERA5 for that region is below the molecular diffusivity (Fig. 2). On the other

hand, if we assume that the mesospheric turbulence due to the breaking gravity waves results in a diffusivity profile as predicted by Lindzen (1981) (Fig. 2), then such a turbulence provides quite rapid exchange of $SF_6$ towards depletion layers making the advective vertical transport above $\sim 50 \, \mathrm{km}$ negligible. The profiles of (Lindzen, 1981), however do not allow for a simple extrapolation to below 50 km, and therefore the vertical profiles by Massie and Hunten (1981) ("1Kz") were involved as the ones that are simple to implement and smooth enough to be easily approximated and extrapolated. The scaling of the "1Kz"

profile allowed for the sensitivity tests.

    The normalized diffusive $SF_6$ mass-fluxes above the domain top for the scaled profiles of the eddy diffusivity (Fig. 3) allow for evaluation of the validity of the assumption of neglected regular vertical transport above the domain top. The equivalent vertical air-mass flux due to diffusion at the level of $0.1 \, \mathrm{hPa}$ (domain top) is $6 \times 10^{-6}$, $9 \times 10^{-7}$, and $2.5 \times 10^{-7} \, \mathrm{kg/m^2/s}$ for "1Kz", "0.03Kz", and "0.001Kz" correspondingly. These mass fluxes, divided by $g$ give the vertical velocities of $-5$, $-0.8$,

and $-0.4 \, \mathrm{Pa/day}$. Comparing these values to those shown in Fig. 2 for the level of 65 km, one can see that the diffusive limit

is valid for the "1Kz" profile except for the very vicinities of the poles. For lower values of the eddy diffusivity the regular circulation becomes comparable with the diffusion or even exceed it.

Although the "0.03Kz" profiles gave somewhat better agreement with the observations of $SF_6$, this does not indicate that "0.03Kz" profiles are more realistic. As suggested by one of the anonymous reviewers, this profile is likely to over-mix the
lower stratosphere and under-mix the upper stratosphere and the mesosphere. Thus the vertical structure of the eddy diffusivity remains a major source of uncertainty in the modelling approach. Using more realistic vertical diffusion profiles and high-top ERA5 reanalysis is planned for the future studies.

### 6.3  Notes on the observed $SF_6$-age

There are three main factors that are responsible for $SF_6$ age being different form the "ideal age": the non-linear growth of
tropospheric burden, the gravitational separation, and the mesospheric sink. Here we consider the effects of these factors and corrections to the $SF_6$ observations that can be applied to compensate for the effect of these factors on resulting AoA.

The correction for the non-linear growth rate introduced by Volk et al. (1997) and used in many subsequent studies is based on a simple analytical model of 1D diffusion with constant diffusivity and exponential distribution of air density. The model was suggested by (Hall and Plumb, 1994) as an illustration for the concept of the age spectrum. The model spectrum has two
parameters: the mean age $\Gamma$ and the width parameter $\Delta$. In order to use the spectrum for the correction one has to involve an additional constraint connecting these parameters. Based on a 3D simulation with a general circulation model Hall and Plumb (1994) suggested that a constant ratio $\Delta^2/\Gamma = 0.7\,\text{year}$ can be used throughout the stratosphere. Note that this dimensional parameter, while having proper units originally, appears without units in several subsequent papers (Engel et al., 2002; Stiller et al., 2012). Volk et al. (1997) used the value $\Delta^2/\Gamma = (1.25 \pm 0.50)\,\text{year}$ for the lower stratosphere based on the results of a
more advanced GCM than the one used by (Hall and Plumb, 1994). With this approach Volk et al. (1997) obtained the difference between the mean age and the lag time (apparent $SF_6$ age). The difference becomes significant for air older than 3-4 years and approaches $(0.50 \pm 0.25)$ years for the oldest (6 years) air measured, which agrees quite well with the difference between the ideal age and the passive $SF_6$ in our simulations shown in Fig. 11bc. The correction for this difference derived from the 1D has been used to reduce systematic biases from $SF_6$-based AoA, though "the global stratosphere is poorly represented by a 1-D
model" (Waugh and Hall, 2002). The uncertainty of the correction of up to $\pm 0.5$ years is systematic, and is not guaranteed to be uniform in space or in time, and likely to affect the trend estimates.

As shown in Sec. 4.1, the biases introduced to the $SF_6$-based AoA by gravitational separation reach a fraction of a year in the upper stratosphere. One could in principle elaborate a correction for gravitational separation; however, the correction would be well within the uncertainty of the correction for the non-linear growth rate, and thus probably not worth considering.

The mesospheric sink clearly has the largest impact on the $SF_6$-derived AoA. The effect of the mesospheric sink is clearly visible above 15-20 km at all latitudes (Fig. 11), and leads to a factor of times apparent over-aging in the upper layers, especially in polar areas. The effect of the sink alone can explain the discrepancy between AoA derived from the MIPAS observations (Haenel et al., 2015) and the AoA from modelling studies (e.g. Diallo et al., 2012; Brinkop and Jöckel, 2019). Compensating for such over-aging is hardly possible without detailed modelling of physical processes including depletion, diffusion and

mean transport, which cause the over-aging. Since the AoA is derived as a *difference* of $SF_6$ mixing ratios, whereas depletion introduces *multiplicative* change to the $SF_6$ abundance, the effect of the sink on apparent $SF_6$ AoA is unsteady in time. This effect is clearly seen in Fig. 12.

Once one has a model that is capable of reproducing the processes behind the $SF_6$ depletion, it makes a full sense to validate such a model directly against available $SF_6$ observations, rather than deriving AoA from $SF_6$ observations and comparing it

against modelled one. In any case the AoA derived from $SF_6$ tracer observations with all the needed corrections applied cannot be considered as purely observed one.

## 7    Conclusions

{sec:co

Eulerian simulations of the tropospheric and stratospheric transport of several tracers were performed with SILAM model driven by ERA-Interim reanalysis for 1980-2018. The simulations included several species representing $SF_6$ under different

assumptions, a passive tracer emitted uniformly at the surface, and an "ideal age" tracer directly comparable are comparable to other state-of-the-art CTM simulations of AoA. To our best knowledge this is the first systematic evaluation of AoA derived from several different tracers within the same simulation over several decades, combined with extensive evaluation against MIPAS and balloon $SF_6$ observations.

Due to the limited vertical coverage and resolution of ERA-Interim in the upper stratosphere, the SILAM simulation domain

had a lid at $0.1\,\mathrm{hPa}$, which is below the altitude of the $SF_6$ destruction. In order to perform realistic simulations of $SF_6$ in our setup, the eddy diffusion in the upper stratosphere and lower mesosphere had to be parameterised, along with the mesospheric sink of $SF_6$.

A set of simulations with different parameterisations for the vertical eddy diffusion showed that published profiles derived with no account for advection (see e.g. Massie and Hunten, 1981, and references threrin) overestimate the eddy diffusivity.

On other hand, the eddy-diffusivity profiles for scalars calculated from ERA-Interim fields according to the IFS procedures ECMWF (2015)), or readily available from the ERA5 reanalysis, appear to be of no relevance for the upper stratosphere, since they fall below the molecular diffusivity. Evaluation of our simulations against satellite and balloon observations indicated that the best agreement between simulated and observed $SF_6$ mixing ratios within the model domain is achieved for the tabulated eddy-diffusivity profile of Hunten (1975) scaled down with a factor of 30. Note, however, that this conclusion is likely to

be a feature the specific model setup. Thus, the question of the importance and magnitude of the eddy diffusivity in the upper stratosphere and lower mesosphere remains open, and $SF_6$ observations is a good means to validate more sophisticated parametrizations of it.

The mesospheric sink of $SF_6$ has a major impact on the mixing ratios above $20\,\mathrm{km}$. The depletion impact is especially strong in wintertime polar areas due to the downdraft within a polar vortex. A set of sensitivity tests showed that molecular diffusion

and gravitational separation of $SF_6$ are responsible for up to a few percent of further reduction in $SF_6$ mixing ratios in the upper stratosphere.

A good agreement of the simulated $SF_6$ distribution to the MIPAS observations up to the altitudes of 30-35 km and to available balloon profiles was shown. The standard deviation between MIPAS and modelled $SF_6$ mixing ratios is up to 80 % controlled by noise error of the satellite retrievals, i.e. the standard deviation between model and MIPAS is about as large as
the error on the satellite data. The results of the comparison also underline the importance of accurate collocation of model and observed data in terms of space, time and vertical averaging of observed data.

The lifetime of $SF_6$ in the atmosphere estimated from the best-performing setup is about 1500 years, which is at the high side of the range of other recent estimates. Our estimate is likely to be biased high due to underrepresented vertical exchange to at the domain top due to missing advective transport and missing effect of braking gravity waves.

Our simulations were able to reproduce both AoA obtained in other model studies, and the apparent $SF_6$ AoA derived from the MIPAS observations. This highlights the role of fast mesospheric destruction of $SF_6$ due to the electron attachment mechanism. Having all tracers within the same simulations we were able to trace the differences in the estimated AoA to the peculiarities of each tracer. A good agreement of the passive-tracer and "ideal-age" AoA indicates a consistency of the simulations, since these two methods have opposite sign of sensitivity to errors of the transport scheme.

The mesospheric sink has severe implications on the AoA derived form $SF_6$. The apparent over-aging introduced by the sink is large and variable in space and season. Moreover, the over-aging due to the sink increases as the atmospheric burden of $SF_6$ grows. All this makes $SF_6$ unsuitable to infer AoA above $\sim 20\,\mathrm{km}$. For a fully-passive $SF_6$ tracer, the variable rate of emissions causes deviations from the "ideal age", and these deviations can be compensated to some extent. However, the correcting for the deviations due to the mesospheric sink of $SF_6$ is hardly possible ithout detailed modeling. These deviations
appear as long-term trends in the apparent AoA. These trends differ from the trends in the "ideal-age AoA", and have no direct correspondence to actual trends in the atmospheric circulation.

Procedures used to derive AoA from observations of various tracers in the atmosphere are inevitably based on assumptions and idealisations that have limited and often unknown area of applicability. The resulting uncertainties in AoA are large enough to preclude the use of apparent AoA and its trends for evaluation of changes in atmospheric circulation or for validation
of atmospheric models. Observations of the tracers themselves, however, have quite well quantified uncertainties, so direct comparisons of simulated tracers to the observed ones are a very promising means for the atmospheric model evaluation. AoA in turn is a convenient means for model inter-comparison if a protocol of AoA derivation is well specified.

*Code and data availability.* The SILAM source code as well as the simulation results used for this study are available from MS or RK on request. The MIPAS observational data available from GS on request. ERA-Interim and ERA5 reanalyses data sets are available from the
European Center for Medium-range weather forecast http://www.ecmwf.int.

*Author contributions.* RK performed the simulations and data analyses, prepared text and illustrations. MS and JV inspired the study and helped with discussions on content and structure of the study, and participated in editing the text. GS provided MIPAS data and wrote sections about MIPAS observations. All authors participated in the final preparation of the text.

*Competing interests.* The authors declare no competing interests.

*Acknowledgements.* The authors acknowledge the support of projects: EU FP7-MarcoPolo (ID: 606953), ESA-ATILA (contract no. 4000105828/12/F/MO and ASTREX of Academy of Finland (grant 139126), Russian Foundation for Basic Research (project 19-05-01008).

The $SF_6$ and mean age-of-air distributions from MIPAS observations were generated within the project STI 210/5-3 of the CAWSES priority program, funded by the German Research Foundation (DFG), and the project BDCHANGE (01LG1221B), funded by the German Federal Ministry of Education and Research (BMBF) within the "ROMIC" program.

The authors are grateful to Viktoria Sofieva (Finnish Meteorological institute) for reading the manuscript and providing useful comments, to Florian Haenel and Michael Kiefer (Karlsruhe Institute of Technology) for technical assistance in handling MIPAS $SF_6$ data, and to four anonymous reviewers whose very instrumental comments helped to substantially improve the paper.

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

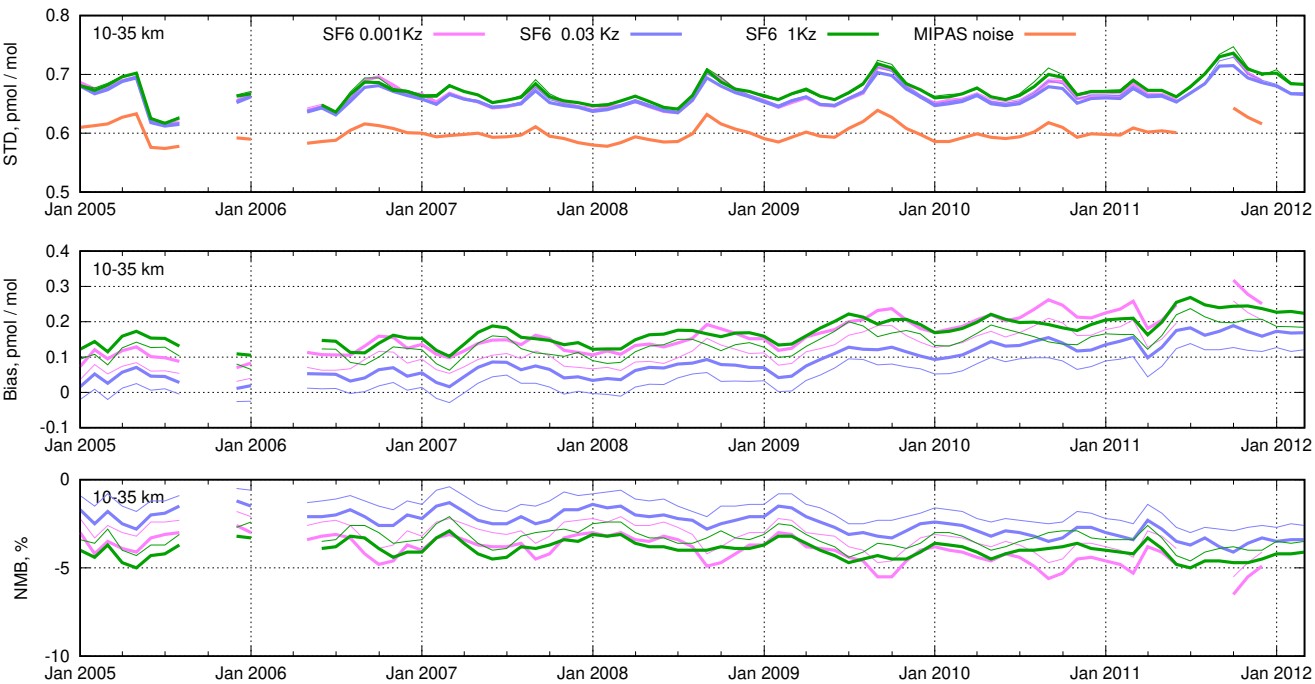

**Figure 7.** The time series of monthly scores for the SILAM-simulated $SF_6$ mixing ratios for the whole period of MIPAS observations in the altitude range of $10 - 35\,\text{km}$. The statistics are standard deviation of model-measurement difference (STD), absolute bias and normalised mean bias (NMB). The statistics of model mixing ratios extracted at nominal MIPAS altitudes are given in thin lines.

{fig:sc

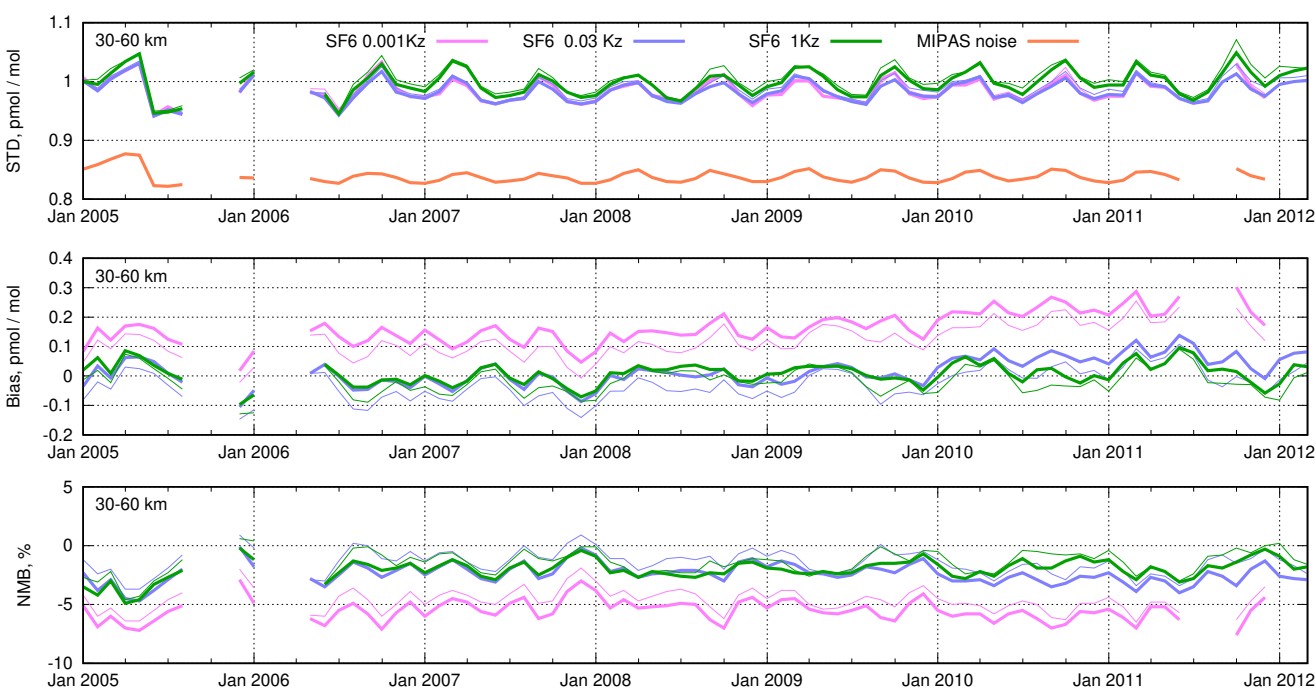

**Figure 8.** Same as in Fig. 7, but for the MIPAS altitude range of $30-60$ km.

{fig:sc

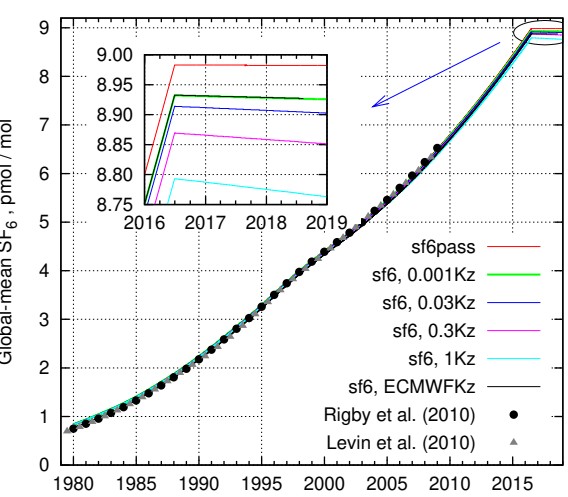

**Figure 9.** The time series of mean mixing ratio of $SF_6$ in the atmosphere simulated with emissions stopped in July 2016. The total burdens by Levin et al. (2010) and by Rigby et al. (2010) are shown for comparison.

{fig:sf

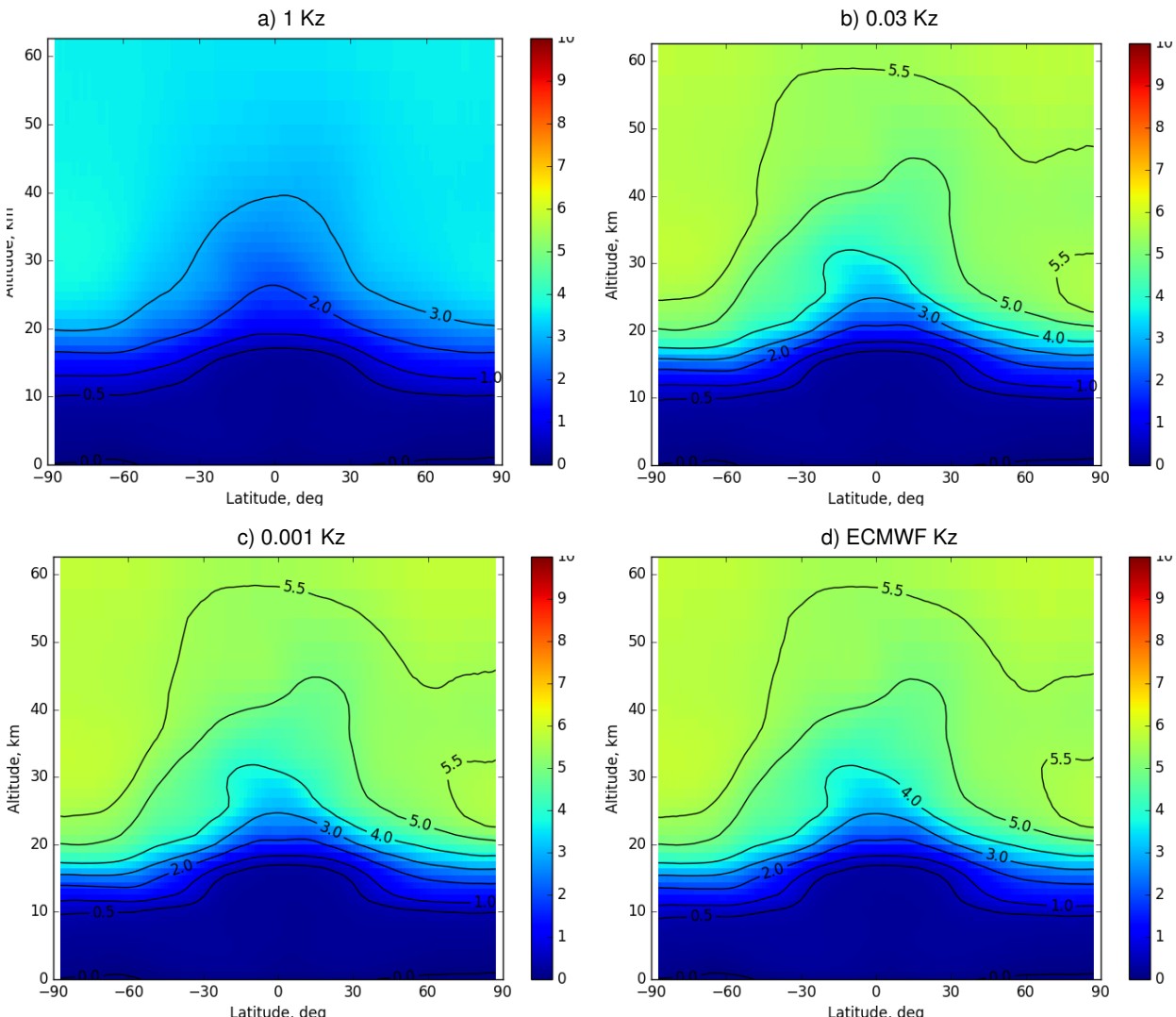

**Figure 10.** The zonal-mean spatial distribution of the ideal-age AoA for 2011 calculated for different eddy-diffusivity profiles. Roux: These pics to be replaced with runs 33-34

{fig:Ao

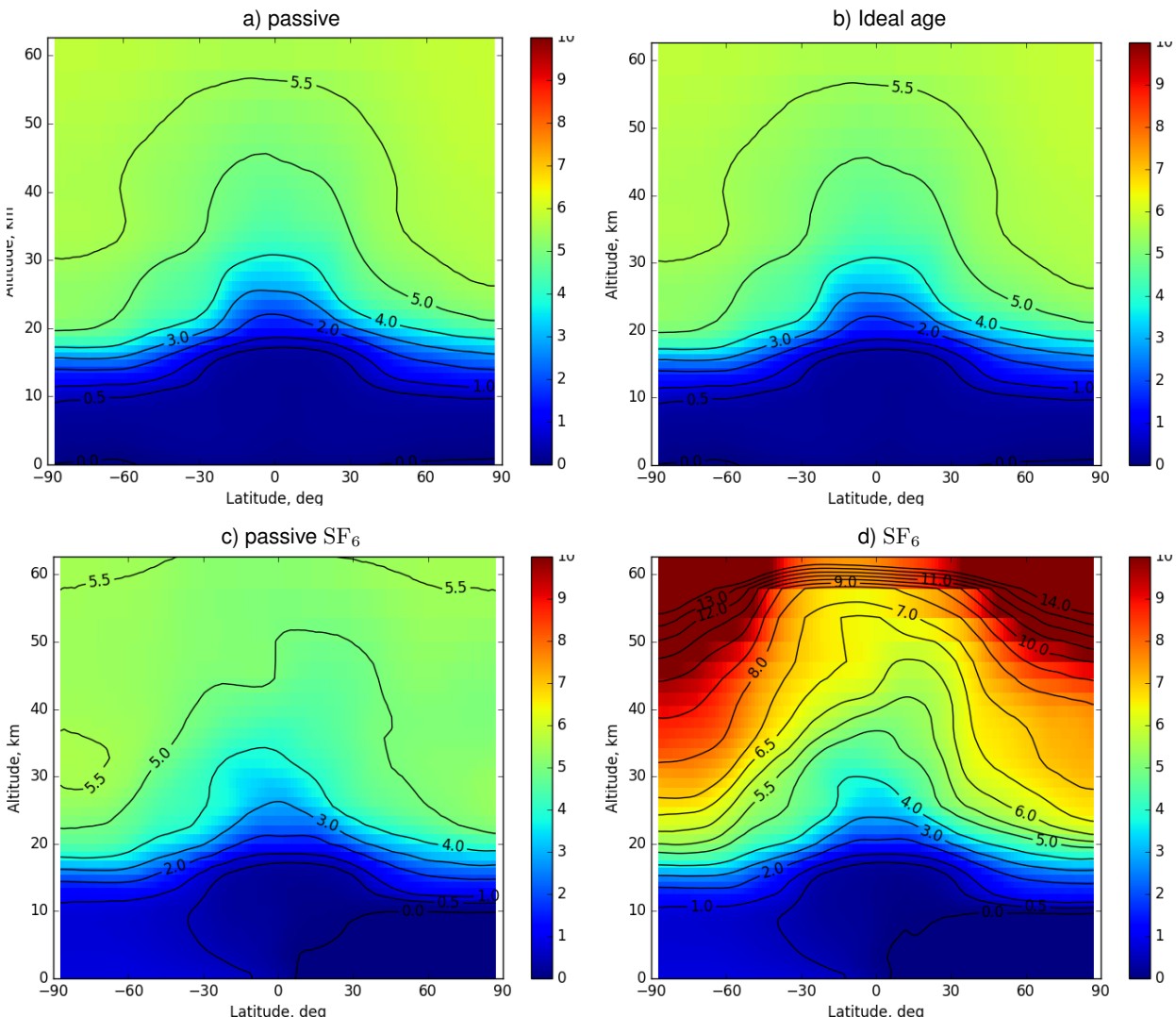

**Figure 11.** Zonal-mean distributions of atmospheric AoA simulated with "passive", ideal-age, and two $SF_6$ tracers, average for 2012. Roux:
Run32                                                                                                                    {fig:ag

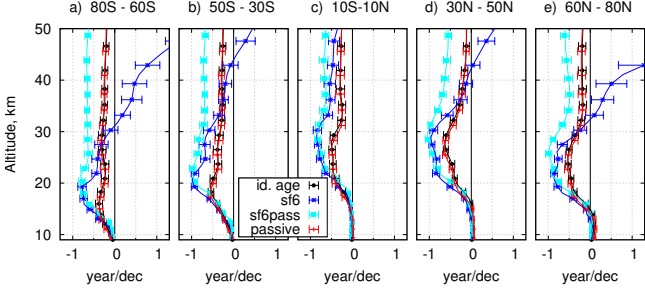

**Figure 12.** Vertical profiles of the simulated age of air linear trends over 11 years 2002-2012 for example latitude belts          {fig:tr

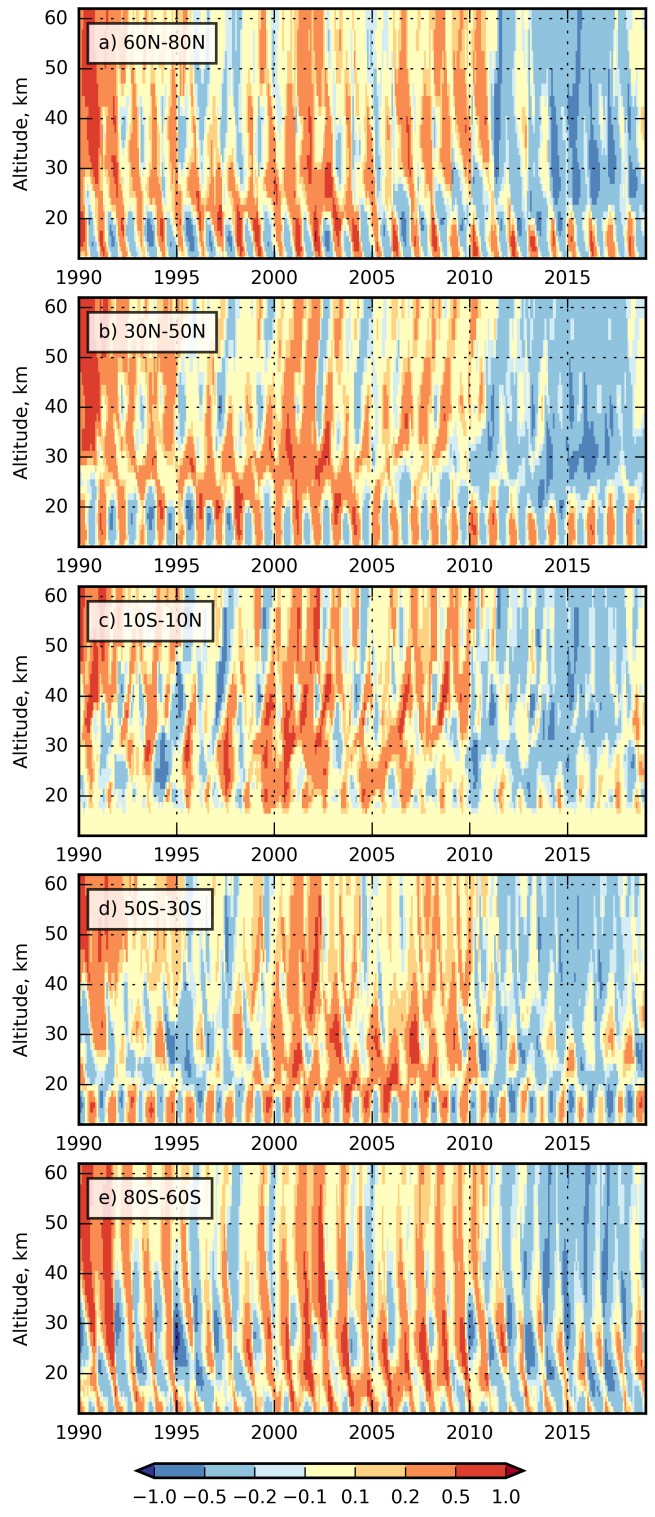

**Figure 13.** Anomaly of the ideal-age AoA (years) for the period of 1990-2018 with respect of the mean AoA over the same period {fig:an

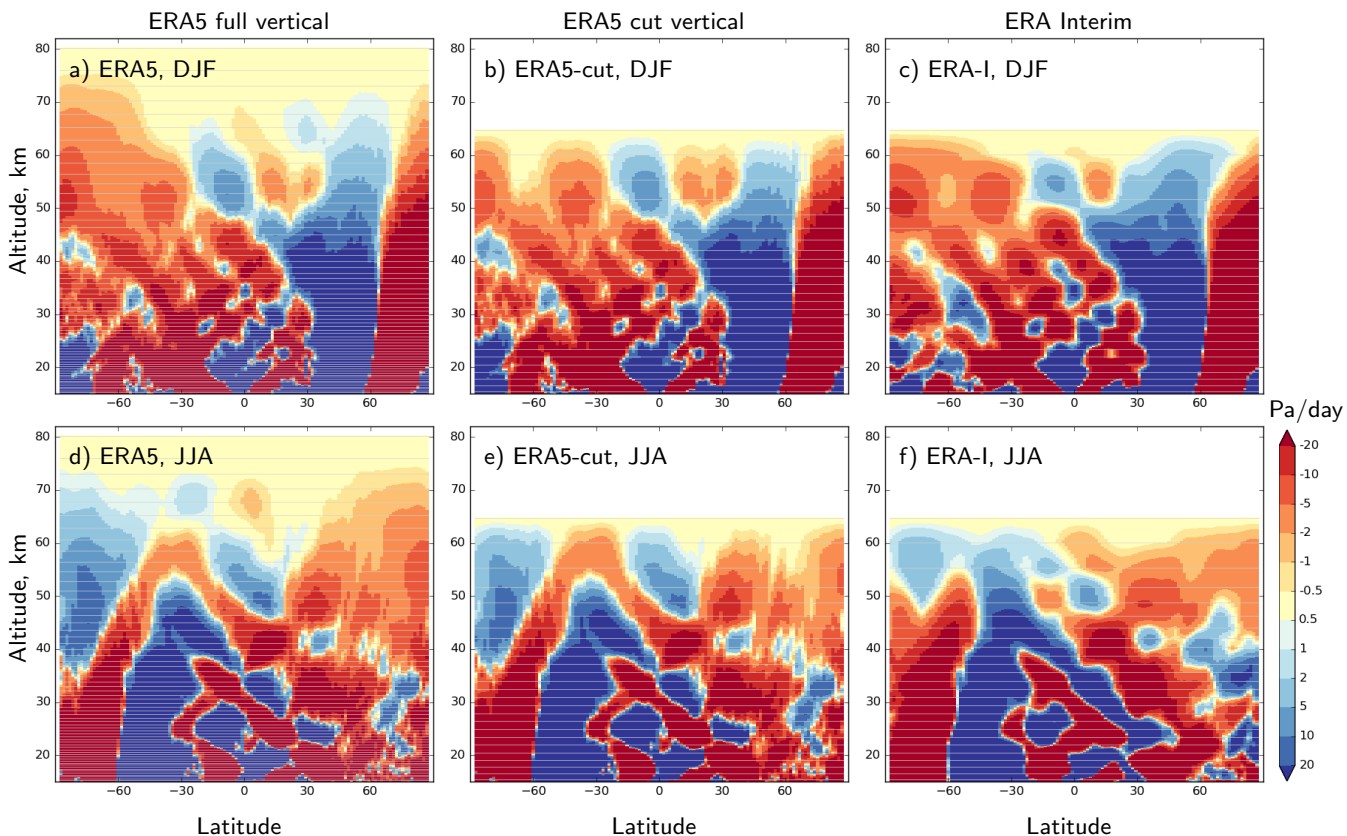

**Figure 14.** The seasonal and zonal-mean vertical air-mass fluxes diagnosed by SILAM from ERA5 and ERA-Interim fields for 2017 solstice seasons, expressed in terms of vertical velocity $\omega$. Updrafts are red. The vertical-layers boundaries are shown with grey lines.

{fig:om