# Peer review of "Simulating age of air and the distribution of $SF_6$ in the stratosphere with the SILAM model"

_Atmospheric Chemistry and Physics, 2019_

## Referee Comment (RC1) · Anonymous Referee #1 · 22 Jul 2019

Recommendation: Major Revision

This study presents simulations of SF6, and determinations of age of air from this tracer, carried out with a chemical transport model (SILAM). It contains useful material that should be suitable for publication. However, it also includes lengthy discussions of molecular diffusion effects that do not appear to be relevant, while omitting essential information on other aspects of transport that are essential for understanding the distribution of SF6.

The model's upper boundary is at 65 km (0.1 hPa). At this altitude, the effects of molecular diffusion are essentially negligible compared to the strong vertical mixing generated by breaking gravity waves, and to advection by the mean meridional circulation forced by wave breaking. Thus, the discussion of molecular diffusion, and of simula-

tions that prescribe unrealistically low values of diffusivity in the upper stratosphere and mesosphere are not very useful and should be omitted in a revised version.

On the other hand, there is little if any discussion of mean meridional advection and how the vertical flux due to the mean meridional circulation is handled in SILAM. The model uses dynamical fields from ERA-Interim, which presumably include the effects of whatever gravity wave parameterization is used in that reanalysis. Thus, vertical fluxes due to mean meridional advection should not be negligible near the upper boundary of SILAM, but the paper does not mention advective transport at all, or how mean advective fluxes are handled at the upper boundary.

Finally, the study does not emphasize enough the role of SF6 loss by the electron attachment mechanism, which becomes fast in the mesosphere (Fig, 1) and is essential for simulating the distribution of SF6, as the WACCM results shown in Fig. 6 (which do not include this loss mechanism) make clear.

I view of all of this, I do not believe the paper is suitable for publication as it stands, but could be made so if revised to (1) eliminate irrelevant material on molecular diffusion; (2) use realistic profiles of eddy diffusion that could be obtained from any high-top model that parameterizes gravity wave breaking; (3) explain explicitly how mean meridional advective fluxes are handled at the upper boundary of SILAM; (4) document how these fluxes affect the distribution of SF6; and (5) emphasize the role of SF6 loss via electron attachment, which is evidently much more important than photolysis. Specific comments on these and other issues can be found below.

Specific comments (line number)

(72) "Silam": This is an undefined acronym. If you are going to use it here you need to define it here, not in the next section. Note also that you write "Silam" here and "SILAM" elsewhere. Please pick one form and stick with it throughout the text.

(129) "10 hPa": The conventional units of pressure in the atmospheric sciences are

hPa (which are equivalent to the now deprecated mb). You might wish to consider changing references to pressure levels to units of hPa to avoid confusion (thus, 0.1 hPa in this instance).

(131) "effect of diffusion of SF6 to the upper layers": Transport through the 0.1 hPa (10 Pa) surface is not solely (or at some latitudes even mainly) due to diffusion; mean meridional advection is important, especially in the polar regions.

(167) "higher than ... accepted in models": It is not clear what this means. What models are you referring to? Global models run at practical horizontal resolution do not produce large vertical diffusivity due to the explicitly resolved motions. However, all recent such models include parameterizations of (unresolved) mesoscale gravity wave breaking. Vertical diffusion coefficients, Kzz, can be estimated from these parameterizations, and they produce values of Kzz that vary strongly with altitude, latitude and season. Thus, a single, global Kzz profile is unlikely to capture accurately the role of vertical diffusion. See also comment at line 387.

(168) "in order to cover the whole range of Kz": I think what you mean to say here is "to cover a range of vertical profiles of Kz". Is that so?

(169) "whose upper part was scaled": what do you mean by "upper part"?

(170) "The three prescribed. . . profiles": This is confusing. Fig. 2 shows profiles labeled Kz, 0.1 Kz and 0.01 Kz, but here in the text you refer to 0.03 Kz and 0.001 Kz. Which is right? The figure legend or the text?

(200) "the difference of equilibrium mixing ratio of SF6": How is this relevant in a range of pressure (0.1-0.2 hPa) where molecular diffusion is essentially negligible? The equilibrium profile defined by Eq. (5) is relevant for the upper mesosphere and above, which is beyond the upper boundary of the model used here. In fact, Eq. (5) and related discussion do not add anything useful to the problem of modeling SF6 below the lower mesosphere.

(203) "in the upper stratosphere heavy gases can no longer be considered as tracers and the molecular diffusion should be treated explicitly": I do not believe this is right. Molecular diffusion effects should be small compared to eddy mixing and mean meridional advection below the upper mesosphere (75-80 km), and certainly within the range of altitude of the present simulations (top boundary at 0.1 hPa, about 65 km).

(225) "flux decreased by several orders of magnitude . . . at the level of a few Pa": But in Fig. 3 all flux profiles increase with altitude. What is the definition of the flux shown in that figure? Does it not include a density factor?

(226) "shown in Fig. 3 with solid lines": Flux profiles in Fig. 3 are labeled Kz, 0.1Kz, 0.01Kz and 0.001Kz. Do these correspond to the Kz profiles of Fig. 2, except that 0.001Kz is not shown in that figure?

(234) "For higher eddy diffusivity . . . molecular diffusion . . . becomes negligible": This should be the situation in the middle atmosphere up to about 75-80 km. Gravity wave parametrizations yield values of Kzz of order 10 m2 s-1 in the lower mesosphere (around 65 km); and mean vertical advection is also large at these altitudes. Therefore, for all practical purposes the effects of molecular diffusion and gravitational separation should be negligible over the range of altitude considered in this study. Note also that, according to Fig. 3, molecular diffusion effects are essentially irrelevant even for unrealistically weak values of eddy diffusion near the upper boundary (0.1 Kz and 0.01 Kz).

(246) "uppermost layer": What is the upper boundary condition on the circulation? Does it force the vertical velocity to be zero at the top boundary? If so, note that the effect of mean meridional transport on SF6 distribution and lifetime will not be modeled realistically.

(265) "'ones' tracer": "unity tracer" might be better.

(304) "the southern polar region": What range of latitude does this cover?

Interactive
comment

(320) "inter-annual variability": This strongest variability seen in Fig. 4 is annual, presumably associated with the cycle of downwelling in winter and upwelling in summer. This again brings up the question what is the upper boundary condition on the dynamics (cf. comment at line 246), and how realistically mean vertical advection is modeled near the upper boundary. Note also that the mean meridional circulation in the mesosphere depends strongly on the contribution of gravity wave drag to the zonal-mean momentum budget, which would depend on how this is parameterized in the ERA-I reanalysis. Details on all of these points are needed.

(324) "simulations with 0.01 Kz": Do you mean 0.001 Kz? That is what the legends in the left column panels of Fig, 4 indicate.

(330) "molecular diffusion . . . maintains the upward flux . . . even if eddy diffusivity ceases": But in the real world, the flux at 65 km (0.1 hPa) is controlled principally by the combined effects of eddy diffusion and mean vertical advection.

(341) "vertical exchange is a key controlling factor": This is correct, but note again that flux due to mean vertical advection is also important and may or may not be modeled properly in the present study, depending on how the upper boundary condition is handled.

(357) "way and rate of SF6 destruction": What does "way of SF6 destruction" mean? You have varied the effective loss rate by changing the flux at the upper boundary, but as far as I can tell the loss mechanism was not changed.

(365) "the most diffusive case . . . overstated SF6": This is likely due to the fact that the "1 Kz" profile has too large values in the lower stratosphere (although it has more reasonable values in the upper stratosphere and lower mesosphere).

(373) "largest deviation below 20 km": See previous comment.

(375) "WACCM . . . under-representing the depletion of SF6 inside the polar vortex": The problem with the WACCM result is that the standard version of the model does not

include loss due to electron attachment, only photolysis. I would expect WACCM to simulate SF6 quite accurately if all loss terms were included. What this result demonstrates is that it is essential to include loss via electron attachment.

(386) "In all the above cases, the '1 Kz' profile is . . . too diffusive": I don't see this in all cases. The 1 Kz profile produces good agreement in the upper stratosphere in Fig. 6 b and d.

(387) "The '0.03 Kz' profiles appear to be most realistic": Actually, none of the profiles is realistic. In particular, the range of Kzz as a function of altitude obtained from gravity wave parameterizations is much larger than shown for the 1 Kz profiles of Fig. 2, where Kz varies from a little under 1 m2 s-1 at 20 km to less than 10 m2 s-1 at 65 km. In models that include a gravity wave parameterization, Kzz is estimated to vary between less than 10-3 m2 s-1 and more than 10 m2 s-1 over the same range of altitude. For a recent example, see Zhu et al. (JAS, 67, 2520, 2010).

(432) "lack of a pole-to-pole circulation": Is this a result of the way the upper boundary condition is handled in these simulations? You need to show the Transformed Eulerian mean circulation as a function of altitude and latitude, at least for the solstice seasons, so the reader can understand the role of mean meridional transport near the top boundary of the model. Explicit description of how the upper boundary flux is handled in SILAM is also necessary.

(433) "understate it above 40 km": In this instance, one could also question the observations, especially the ones that show an increase in mixing ratio with altitude. It is unclear how such profiles could be generated for a tracer that has a source in the troposphere and a sink in the mesosphere.

(435) Figure 8: Is the "de-biased RMSE" in the figure caption the same thing as "STD" in the ordinate label of the top panel? It would be desirable to keep the terminology consistent to avoid confusion.

(436) "the difference in statistical scores of the three selected simulations is quite minor": What "statistical scores" are you referring to?

(440) "standard deviation of model-measurement difference": How does this eliminate the influence of model bias, assuming that is what you mean to say here? Doesn't the model-measurement difference contain the bias? A formal definition of this quantity, similar to what is done in Eq. (11) for the NMB, would be useful.

(444) "RMS error of the observations due to retrieval noise in the original MIPAS data": Is this what you mean by the legend "MIPAS noise" in the top panel of Fig. 8?

(450) Figure 9: This needs labels for the various curves, as in Fig. 8.

(452) "for the upper troposphere": What does this refer to? This paragraph discusses results for 30-60 km. What does this have to do with the upper troposphere?

(460) "Three other profiles of Kz result in practically identical distribution of AoA": This would imply that vertical mixing is irrelevant for the small Kz cases, and raises the question what controls AoA in these simulations.

(484) "The resulting model-based apparent AoA [is] much older than the "ideal-age" AoA and pretty close to the values derived from MIPAS": This is an important result that highlights the role of fast mesospheric destruction of SF6 due to the electron attachment mechanism.

(503) "The reason is...": You should reference Stiller at al. (2012) here, who already pointed this out.

(512) "'ideal age' and "passive" tracers: Are the results for the "ideal age" tracer the set of points labeled "time lag" in Figure 12? Again, consistency in terminology would be desirable.

(537) "eddy-diffusivity profile of Hunten (1975) scaled down": The Hunten profile almost certainly overestimates diffusivity in the lower stratosphere, but reducing it by a factor

of 0.03 will not reflect the behavior of vertical mixing in the upper stratosphere and the mesosphere. Ideally, one would estimate vertical mixing (as a function of altitude, latitude and season) from a gravity wave parameterization. Since such a parameterization was not available in the context of the present study, the conclusions regarding the role of $K_z$ in determining age of air cannot be taken at face value.

Typos, etc.:

(28) "that presents an analogy of Lagrangian clock" -> "that is analogous to a Lagrangian clock"

(33) "are not possible, therefore..." -> "are not possible; therefore, ..."

(285) "Eulerian analogy" -> "Eulerian analog"

(318) "is by more than an order of magnitude stronger than one of gravity separation" -> "is stronger than diffusive separation by more than one order of magnitude"

(319) "Regardless the used Kz profiles" -> "Regardless of the Kz profiles used"

(344) "depleting SF6" -> "SF6 that undergoes chemical destruction"

(344) "start to fall down" -> "begin to decrease"

(452) "on pair" -> "on par"

(482) "nor its mixing ration" -> "nor does its mixing ratio"

---

## Referee Comment (RC2) · Anonymous Referee #2 · 8 Aug 2019

The study by Kouznetsov et al investigates the impact of the vertical diffusion and of the mesospheric sink of SF6 and the SF6 climatology and its trends using a chemistry-transport model. While the mesospheric transport is not explicitly included (due to lack of ERA-Interim data above 0.1 hPa), a parametrization of eddy diffusivity as well as molecular diffusivity is included to mimic transport to the mesosphere. The subject of the study is of high relevance, as SF6 is used frequently to estimate Age-of-Air, and the role of its sinks needs to be better understood. The study is overall well presented and the methods are overall appropriate, but some clarifications are needed (see comments below). Overall, I recommend the authors revise the paper minorly before it can be considered for publication.

General comments:

[Figure]

1. In lines 45 ff, you correctly mention that a correction has to be applied when deriving AoA from a non-linear increasing tracer, as SF6, as has been done by observational studies. However, it is not entirely clear to me how you calculated AoA from SF6 - simply as time lag, as for the linear increasing tracer? It certainly is known that just calculating the time lag leads to deviations from the true AoA values. If you choose not to include a correction method in the calculation of AoA, you certainly should stress this fact, and I suggest you to refer to the SF6-derived "AoA" as "time lag" rather than AoA. The comparison of the SF6-derived time-lag with / without chemical sink is still valid, but I caution you on the conclusions you draw from the difference of the passive sf6 tracer and the ideal age /linearly increasing tracer: as long as no correction method for the non-linearity is implied, you cannot conclude on whether the non-linear increasing tracer can be used to deduce AoA values in general.

2. While the parametrizations of eddy diffusivity, gravitational separation by molecular diffusion and of SF6 loss are described well in detail, the way they are actually implemented in the model is not entirely clear to me. According to Section 3.4, the overall budget equation of the abundance of a tracer (SF6) is solved for steady state, and this steady state solution scaled by the actual tracer concentration is used above the model top - is this correct? And how exactly is this implemented - as loss due to the lifetime given in line 238 ? Furthermore, it was not clear to me whether the diffusive parametrizations are also applied in the actual model domain, or only for the parametrization above the top level? It could be helpful if you describe the overall approach at the beginning of section 3 (i.e. parametrization of upward transport above ~10Pa by vertical diffusion, where SF6 is depleted, and thus there is no downward transport of SF6).

3. Related to the above comment, I wonder how sensitive your results are to the fact that you represent transport above the model top only as vertical diffusive process, i.e. the actual transport circulation is missing (which circulates air, and thus SF6 from pole-to-pole, as opposed to your assumption of all SF6 that is transported diffusively

upward being lost). Probably the lack of advective transport also affects the results of the evaluation of different values for Kz? Or is this more based on the layers within the model domain (if diffusion is applied there too, see comment above)? Please add discussion of those issues to your study.

Specific comments:

- line 25: you describe here the estimation of AoA with Lagrangian trajectories, but without inter-parcel mixing. The inter-parcel mixing does affect AoA, and there are studies that account for this mixing in Lagrangian frameworks (e.g. Brinkop et al., 2019, Plöger et al., 2015). Thus estimates of AoA with Eularian methods might differ from Lagrangian methods due to the way inter-parcel mixing is calulated. This methodological point should be mentioned somewhere.

(References: Brinkop and Jöckel, ATTILA 4.0: Lagrangian advective and convective transport of passive tracers within the ECHAM5/MESSy (2.53.0) chemistry–climate model, Geosci. Model Dev., 12, 1991–2008, https://doi.org/10.5194/gmd-12-1991-2019, 2019

Ploeger, F.; Riese, M.; Haenel, F.; Konopka, P.; Müller, R. & Stiller, G. Variability of stratospheric mean age of air and of the local effects of residual circulation and eddy mixing Journal of Geophysical Research: Atmospheres, 2015, 120, 716-733)

- line 27: "above-mentioned observational method": I dont see you mention the observational method above this statement?

- line 44: Garcia et al did show that the corrections improve the trend estimate, and they do not use the exact same correction method than what was applied to the observations. So I would not argue that the tracers are "ambiguous proxies" for AoA, but rather that the correction methods accounting for the non-linearity need to be investigated more deeply.

- line 95 ff: Maybe you can mention here which variables from ERA-Interim you use - I

was wondering at this point how vertical transport is calculated, and this became clear only in section 3.5.

- line 122 ff, general: How certain are the SF6 destruction rates, i.e. how do the results by Totterdill et al compare to other studies? Please add a short statement.

- line 156: its not clear to me what the "limiting value" is, and why Kz is "practically always" set to it? Please be more specific here.

- line 159: Kz does not fall below the molecular diffusivity is the lower stratosphere, below ∼40 km, according to Fig. 2, so please refine the statement.

- line 196: do you mean mixing ratio differences between the two layers? Why two layers, and not at one layer? Or do you mean the mean mixing ration in the layer bounded by an upper and lower pressure? it might be easier to put down the equation rather than describing it.

- line 212: I assume you use the US standard atmosphere because at the levels where it matters, ERA-Interim is not available any more? Again, it is not entirely clear if / how you apply this parametrization only at the "top layer", or also throughout the model domain. If the latter is true, the actual ERA-Interim temperatures could be used in the model domain (even tough you could argue that it does not make much of a difference there, as molecular diffusion does not play a role).

- line 224: please be more specific and describe how you obtain the flux F(p) from the steady-state solution of the mixing ratios.

- line 236: see general comment: please be more specific on how exactly the different parametrizations are used in the different areas, and how the upper boundary parametrization is implemented (via the lifetimes?)

- line 250: which "other parameters" do you use?

- line 267: were the other tracers corrected using the "ones" tracer, or just the error

"evaluated"?

- line 343 ff: is this the best way to estimate lifetimes, or couldn't you just average the inverse destruction rate mass-weighted over the entire atmosphere? Also, at line 348, you write that the delay of SF6 between troposphere and upper layers is about 5-6 years, and then use the value 5 years previous to the emission stop to evaluate the lifetime - is this quantitative, or just a rough estimate?

- line 361: "we have found in literature"-> be more specific, e.g. observations that were published by ...

- line 365: "strong exchange through the troposphere"? do you mean too strong upward transport by the diffusion?

- line 384: what is the dynamical reason for the minimum in SF6, and why do you think it is weaker in the model?

- line 464: "practically"? please be more specific

- line 471 ff: as mentioned in the general comment, you should clarify how AoA was calculated from the SF6 tracers, and possibly change the naming to "time-lag".

- line 486: you mentioned earlier that you use a new version of the MIPAS SF6 data, but do not show its AoA, but instead refer to the older published AoA figures. Why don't you add the new MIPAS AoA to Figs. 11 and 12?

- line 515: "non-uniformity" of ERA-Interim, what do you mean? Couldn't this just be the trend in AoA over the period, or why do you think it is an artefact? Further, in line 519, you state that ERA-Interim was not recommended for climatological studies. I'm surprised by this statement, given that ERA-Interim is the basis for a lot of studies of climatologies and trends in various variables. Can you specify which source you quote here, and what exactly should not be done?

- line 521: The trends over the MIPAs period could be compared to other CTM results,

e.g. by Plöger et al, 2015b, who showed that their CTM was capable to reproduce the MIPAS AoA trend rather well.

(Ploeger, F.; Abalos, M.; Birner, T.; Konopka, P.; Legras, B.; Müller, R. & Riese, M. Quantifying the effects of mixing and residual circulation on trends of stratospheric mean age of air Geophysical Research Letters, 2015, 42, 2047-2054)

- line 525: why comparable with Lagrangian simulations? As pointed out before, one difference is the accounting for inter-parcel mixing. I'd rather argue that your results are comparable to other state-of-the-art CTM simulations of AoA.

- line 542: Are those "best" estimates in the upper stratosphere based on the "upper layer", where advective transport is not accounted for? Or dou you refer to the results in the model domain?

-line 549: I dont understand the sentence on the standard deviation controlled by noise. Do you mean to say that the standard deviation between model and MIPAS is about as large as the error on the satellite data?

- line 551: you might want to add the range of lifetimes you obtain.

- line 560: as stated in the general comments, as long as you do not apply corrections for the non-linear growth, you can not conclude on the suitability of the non-linear tracers in general. You can conclude here that without correcting for the non-linear growth, the apparent AoA and its trends deviate strongly, and that this motivates the investigation of correction methods.

Typos/ Language / Technical:

- Abstract, line 11: ".. does not exceed 6-6.5 years": it is not clear to me what this statement refers to - is this the "true" (ideal age) maximum value for AoA?

- line 18: what do you mean by "polar circulation" ?

- line 37, and general: check the parenthesis around references, they are incorrect at

several places, for example here it should read (Waugh, 2009 ; Stiller et al., 2012)

- line 86: "transformation procedure" - do you mean the chemical "transformation"? -> change to "chemical sinks" (?)

- line 122: over 60 km -> above 60 km; "that fall..." -> "i.e. within and above the top most..."

- line 159: please avoid using the word "practically", as it is not very specific

- line 168: "than ones accepted.."- Do you mean "than the ones usually used in models"?

- line 176: "the mesosphere" (add the)

- line 196: "in the vertical, one obtains that the ..."

- line 202: the overwhelming" (add "the")

- line 247: remove "been"

- line 267: "rations" -> "ratios";

- line 300: "downdraught" -> "downwelling"

- line 344: fall down -> decrease

- line 383: "the one in Fig.." (add the)

- line 386: "furthermost" -> furthest

- line 419: to the polar (replace "a" by "the")

- line 426: overstating -> overestimating

- line 452: do you mean upper stratosphere?

- line 482: "nor its ..."-> "nor does its mixing ratio" (remove "n")

- line 484: replace second "well" with "with"

- line 490: pointed out (add "out")

---

## Referee Comment (RC3) · Anonymous Referee #3 · 8 Aug 2019

This is an interesting manuscript exploring, in a model environment, the effects of chemistry, gravitational separation and diffusivity on SF6 mixing ratios in the stratosphere and the mean age of air derived from it. Clearly a lot of work has gone into devising the various model setups and I would in general support the publication of this work. However, some questions need to be answered and some potential issues resolved beforehand. One example is e.g. that even though it is driven by ERA-Interim there is no guarantee that this model will accurately reproduce stratospheric transport patterns including the overturning circulation, transport barriers, the QBO, etc., all of which can influence AoA. Perhaps this was demonstrated in Sofiev et al., 2015? If so, it would be good to give a short summary, if not, some further details are required. Some further points can be found in the below.

[Figure]

Title: Consider adding "the" before "distribution" and "SILAM". There are various other places in the manuscript with small language deficiencies like this.

Abstract: An introductory sentence to alert the reader to the fact that this paper is on the stratospheric overturning circulation (and perhaps its importance) would be helpful.

Line 11: This should be "adds" and I would also recommend adding "up to".

Line 32-33: Age is not the correct term here as oceanic water has been around for some time. I suggest replacing it with e.g. "transport times".

Line 109-113: I was quite surprised to find a new satellite product hidden in this modelling-focused manuscript. Given that there are "considerable" differences to previously published SF6 data sets I urge the authors to provide more details and make their statements more quantitative (e.g. defining "considerably higher" and "closer"; where does the "new" CFC-11 band come from?; does the correction influence trends in the 2002-2012 period?), perhaps even by adding a figure to support their claims.

Line 118-119: Figure 1 is bad quality and Figure 2 needs some further explanation in the caption.

Line 119: Is it Silam or SILAM?

Line 170-171: Looking at Figure 2 none of the three profiles seem to capture the vertical gradient from the ERA-5 data. Why is that?

Line 219-220: Please quantify: How negligible does vertical advection need to be? And how does that compare to actual vertical advection in the stratosphere and mesosphere?

Line 259: The details of the simulation setup are beyond my expertise. However, this statement seems somewhat vague. What does "normally" mean here? And how large is the precision of the input wind fields? Does it e.g. vary over time?

Line 273-275: This is a major problem. A linear extrapolation can introduce biases,

especially since 4 years of the extrapolated period overlap with the MIPAS period. Why do the authors not use more up-to-date publically available data, e.g. from the AGAGE and NOAA ESRL networks. Looking at Figure 1-21 in the recent Scientific Assessment of Ozone Depletion (2018), global emissions of SF6 appear to have been much lower between 2008 and 2016, closer to 0.21 Gg/yr. The implications for the derived AoA and its trend could be quite severe and should at least be assessed.

Line 287-294: This paragraph raises a few questions. What does "0.001Kz eddy diffusivity" mean in detail and why was the initialisation performed that way? When was the initialisation started? Which emissions were used for SF6 species from 1980-1989 and which meteorological fields for 1970-1979? What about the pre-1970 time period?

Line 306: This should be "from".

Line 313-314, 353-355: Why are the lifetimes in Table 1 so long? It looks like the model is not able to reconcile realistic diffusion rates with recently published lifetime estimates for SF6 (e.g. Kovacs et al. and Ray et al.). This needs to be discussed. I also do not agree that there is good agreement with lifetime estimates from other studies (Line 353-355) as all of the higher lifetime estimates (>∼1500 years) come from outdated studies.

Line 399-400: Please improve Figure 7. It is currently very hard to decipher the legend and text inside the graph area and the two lines for each colour are undistinguishable. Also, please add the uncertainties of the MIPAS data points (one could at least add standard deviations of the observed values as in Kovacs et al.) and why are SF6 mixing ratios increasing at the high altitude end for some profiles?

Line 471-475: Plotting the residual between 11a,b and c might help visualising the differences. Also, please quantify "slight old bias".

Line 547-548: Looking at Figure 7 I cannot agree with this statement, at least not until some realistic uncertainty estimates have been added to the observations.

Line 551-552: This is right at the upper end of recent estimates, so not too good agreement. Given that the authors state themselves earlier in the manuscript that "insufficient vertical resolution of ERA-Interim in the upper stratosphere and lower mesosphere, and lack of pole-to-pole circulation" limit model performance (resulting also no conclusion being drawn on AoA trends) I find that statement too strong.

Page 28-29: Figure 8 and 9 are currently not mentioned anywhere in the manuscript.

---

## Referee Comment (RC4) · Anonymous Referee #4 · 28 Aug 2019

This paper simulates the impact of the mesospheric destruction and gravitational separation on stratospheric SF6 distribution using a chemical transport model driven by ERA-Interim meteorology. In the model, mesospheric depletion and gravitational separation of SF6 are parameterized as upper boundary conditions. Sensitivity simulation were conducted and the roles of mesospheric destruction, gravitational separation, and vertical turbulent diffusion in the distribution of stratospheric SF6 are determined. The effects of these processes on the derived mean age of air and its trend are also discussed.

This paper clearly demonstrate that the apparent mean age of air derived from SF6 measurements is not suited for studying the trend of stratospheric mean age of air. The results have important implications in understanding the differences in the observed

and modeled Brewer-Dobson circulation trends. I recommend publication of the paper after my comments are addressed.

Comments:

My major concern is that the SILAM model doesn't capture the SF6 distribution in the upper stratosphere. The authors attribute this deficiency to the low top of ERA-interim that can't accurately represent the circulation in the upper stratosphere and mesosphere. However, the mesosphere circulation, particularly the downwelling branch of the summer-to-winter pole circulation, is essential to understand how the mesospheric sink affects SF6 distribution in the stratosphere. This issue needs to be discussed in more detail. I wonder if it is possible to drive SILAM with a model of higher top, e.g., WACCM, to see if SF6 in the upper stratosphere can be improved.

Section 5: Describe how the mean age of air is derived using SF6.

Lines 484-486: Figure 7 shows that the simulated SF6 distribution doesn't agree with MIPAS measurement about 40 km (above 30 km in the winter pole). How can the derived AoA agrees with each other?

---

## Author Comment (AC1) · 20 Dec 2019

**Responses to the Interactive comments on "Simulating age of air and distribution of $SF_6$ in the stratosphere with SILAM model"**

Rostislav Kouznetsov[1,2], Mikhail Sofiev[1], Julius Vira[1,3], and Gabriele Stiller[4]

[1]Finnish Meteorological Institute, Helsinki, Finland
[2]Obukhov Institute for Atmospheric Physics, Moscow, Russia
[3]Currently at Cornell University, Ithaca, NY, USA
[4]Karlsruhe Institute of Technology, Karlsruhe, Germany

**Correspondence:** Rostislav Kouznetsov (Rostislav.Kouznetsov@fmi.fi)

**1    Response to the Interactive comment RC1**

*This study presents simulations of SF6, and determinations of age of air from this tracer, carried out with a chemical transport model (SILAM). It contains useful material that should be suitable for publication. However, it also includes lengthy discussions of molecular diffusion effects that do not appear to be relevant, while omitting essential information on other aspects of*
5    *transport that are essential for understanding the distribution of SF6.*

*The model's upper boundary is at 65 km (0.1 hPa). At this altitude, the effects of molecular diffusion are essentially negligible compared to the strong vertical mixing generated by breaking gravity waves, and to advection by the mean meridional circulation forced by wave breaking. Thus, the discussion of molecular diffusion, and of simulations that prescribe unrealistically low values of diffusivity in the upper stratosphere and mesosphere are not very useful and should be omitted in a revised*
10    *version.*

*On the other hand, there is little if any discussion of mean meridional advection and how the vertical flux due to the mean meridional circulation is handled in SILAM. The model uses dynamical fields from ERA-Interim, which presumably include the effects of whatever gravity wave parameterization is used in that reanalysis. Thus, vertical fluxes due to mean meridional advection should not be negligible near the upper boundary of SILAM, but the paper does not mention advective transport at*
15    *all, or how mean advective fluxes are handled at the upper boundary.*

*Finally, the study does not emphasize enough the role of SF6 loss by the electron attachment mechanism, which becomes fast in the mesosphere (Fig, 1) and is essential for simulating the distribution of SF6, as the WACCM results shown in Fig. 6 (which do not include this loss mechanism) make clear.*

*I view of all of this, I do not believe the paper is suitable for publication as it stands, but could be made so if revised to*
20    *(1) eliminate irrelevant material on molecular diffusion; (2) use realistic profiles of eddy diffusion that could be obtained from any high-top model that parameterizes gravity wave breaking; (3) explain explicitly how mean meridional advective fluxes are handled at the upper boundary of SILAM; (4) document how these fluxes affect the distribution of SF6; and (5) emphasize the*

*role of SF6 loss via electron attachment, which is evidently much more important than photolysis. Specific comments on these and other issues can be found below.*

25 **Response:** Thank you very much for your valuable comments and nice and concise summary of the major points. Here are the responses for them.

(1) eliminate irrelevant material on molecular diffusion;

We would not agree that molecular diffusion is irrelevant. The range of magnitudes for turbulent diffusion mentioned below in the comment for line. 387 ($10^{-3} - 10^{2}\,\mathrm{m^2s^{-1}}$) overlaps with the range of molecular diffusivities in the upper stratosphere and

30 mesosphere ($10^{-3} - 10^{-1}\,\mathrm{m^2s^{-1}}$, see Fig. 1 of the manuscript). Thus molecular diffusivity cannot be considered as negligible in general.

The study deals with turbulent diffusion. Molecular diffusion poses quite well defined lower limit for diffusive transport. Consideration of molecular diffusion helps to interpret results of parametrizations for turbulent diffusion as irrelevant when they fall below the molecular diffusion. In particular, without molecular diffusion it would be impossible to draw a conclusion

35 that regardless the eddy diffusivity or advective transport the lifetime of SF6 in the upper model layer of our simulations is at most 60 days. This conclusion is at least interesting, to our view.

Moreover, to our best knowledge, the present study is the only one to date that explicitly quantifies the role of gravitational separation on the SF6 distribution in the atmosphere. The molecular diffusion is the mechanism responsible for gravitational separation.

40 Thus we decided to keep corresponding parts of the paper.

(2) use realistic profiles of eddy diffusion that could be obtained from any high-top model that parameterizes gravity wave breaking

We agree that some of the turbulent profiles tested in the paper look unrealistic - and that was the very reason for testing several options. As we saw in the literature and pointed out in the paper, there are several parameterizations and estimates of

45 the turbulence in the upper troposphere and the stratosphere - and they do not agree with each other. Therefore, in the paper we evaluated the sensitivity of the AoA and SF6 to a variety of assumptions about the absolute levels of turbulent diffusion but preferred not to select the one out of many.

We agree that use of more sophisticated schemes could bring extra information but it would also require resolving the vertical and horizontal air motions near and above the model lid $0.1\,\mathrm{hPa}$ both in vertical and in horizontal dimensions, consistent

50 matching these motions to the air-flux fields we derive from ERA-Interim winds, involvement of ERA5 with higher vertical coverage, etc. All these deserve a separate study and can hardly be fit into the current paper.

As we have specified in Introduction, the aim of the study is to provide consistent simulations simultaneously reproducing the spatio-temporal distribution of AoA and the $\mathrm{SF_6}$ mixing ratio in the troposphere and stratosphere.

The way the simulations were made resulted in a distributions of $\mathrm{SF_6}$ in troposphere and lower stratosphere that agree quite

55 well to both balloon measurements and MIPAS retrievals.

(3) explain explicitly how mean meridional advective fluxes are handled at the upper boundary of SILAM;

We are grateful for pointing out this omission. The model uses a "hardtop" wind diagnostic procedure, forcing zero vertical wind at the domain top ($0.1\,\mathrm{hPa}$), thus precluding any advective fluxes through the lid. This is an immediate consequence of the topmost level of the ERA-Interim reanalysis at $0.1\,\mathrm{hPa}$.

Therefore, the hard-top assumption is the only that allows for the global air-mass conservation. The statement somehow slipped out of the "Simulation setup" section. Added now at the beginning of Sec. 3.

The diffusive fluxes through the domain top are however allowed.

(4) document how these fluxes affect the distribution of SF6

The ways the fluxes through the domain top affect distribution of $SF_6$ are analyzed in Section "Sensitivity and validation of SF6 simulations".

(5) emphasize the role of SF6 loss via electron attachment, which is evidently much more important than photolysis

We fully agree that electron attachment is much more important than photolysis, as it has been shown by Totterdill et al. (2015). A brief overview of relative role of electron attachment and photolysis is added to the "SF6 destruction" section.

Explicit mention of electron attachment as the mechanism for mesospheric loss is added to the Conclusions.

*Specific comments (line number)*

*(72) "Silam": This is an undefined acronym. If you are going to use it here you need to define it here, not in the next section. Note also that you write "Silam" here and "SILAM" elsewhere. Please pick one form and stick with it throughout the text.*

**Response:** The acronym has been explained, "Silam" has been changed to "SILAM" throughout the paper.

*(129) "10 hPa": The conventional units of pressure in the atmospheric sciences are hPa (which are equivalent to the now deprecated mb). You might wish to consider changing references to pressure levels to units of hPa to avoid confusion (thus, 0.1 hPa in this instance).*

**Response:** Pressure units were to hPa throughout the paper.

*(131) "effect of diffusion of SF6 to the upper layers": Transport through the 0.1 hPa (10 Pa) surface is not solely (or at some latitudes even mainly) due to diffusion; mean meridional advection is important, especially in the polar regions.*

**Response:** As we had a lid at 0.1hPa imposed by ERA-Interim, no regular mass fluxes are possible through the domain top in our simulations. The paragraph has been rephrased to avoid the word "diffusion" here. The role of a regular transport is discussed in the "Discussion section".

*(167) "higher than . . . accepted in models": It is not clear what this means. What models are you referring to? Global models run at practical horizontal resolution do not produce large vertical diffusivity due to the explicitly resolved motions. However, all recent such models include parameterizations of (unresolved) mesoscale gravity wave breaking. Vertical diffusion coefficients, Kzz, can be estimated from these parameterizations, and they produce values of Kzz that vary strongly with altitude, latitude and season. Thus, a single, global Kzz profile is unlikely to capture accurately the role of vertical diffusion. See also comment at line 387.*

**Response:** The expression has been replaced with a more precise reference to the ERA5 dataset. Applying these parameterizations in SILAM would be certainly interesting to do. However, these parameterizations tend to disagree with each other

and with (few) observations, which also implies additional research for the reasonable choice of a particular parametrization. In the current study we chose rather to evaluate the sensitivity to Kz levels than to attempt to find the best-fitting formulations. This topic has been included in our plans for the next studies that will be driven with ERA-5.

*(168) "in order to cover the whole range of Kz": I think what you mean to say here is "to cover a range of vertical profiles of Kz". Is that so?*

**Response:** Corrected.

*(169) "whose upper part was scaled": what do you mean by "upper part"?*

**Response:** Rephrased.

*(170) "The three prescribed. . . profiles": This is confusing. Fig. 2 shows profiles labeled Kz, 0.1 Kz and 0.01 Kz, but here in the text you refer to 0.03 Kz and 0.001 Kz. Which is right? The figure legend or the text?*

**Response:** Initially we used 1Kz, 0.1 Kz and 0.01 Kz, but then it turned out that 0.001 Kz is also interesting, so finally 1Kz, 0.03 Kz and 0.001Kz were used for the paper. The figure has been replaced now. Also line colors changed to agree with the rest of the paper.

*(200) "the difference of equilibrium mixing ratio of SF6": How is this relevant in a range of pressure (0.1-0.2 hPa) where molecular diffusion is essentially negligible? The equilibrium profile defined by Eq. (5) is relevant for the upper mesosphere and above, which is beyond the upper boundary of the model used here. In fact, Eq. (5) and related discussion do not add anything useful to the problem of modeling SF6 below the lower mesosphere.*

**Response:** We would not agree with the statement that molecular diffusion is always negligible below the mesosphere. The molecular diffusion is the key mechanism for gravitational separation which has been observed in stratosphere Ishidoya et al. (2008, 2013); Sugawara et al. (2018). As shown in Sec. 4.1, the effect of the gravitational separation on the AoA reaches a fraction of a year, which is comparable to the magnitude of corrections to the SF6-AoA considered e.g. by Stiller et al. (2012). The sensitivity studies indicate small but noticeable effect of molecular diffusion even for "1Kz". The molecular diffusion has little effect on the SF6 distribution due to the overwhelming impact of mesospheric depletion rather than due to the eddy diffusion. The latter result is worth including into the paper, in our opinion.

*(203) "in the upper stratosphere heavy gases can no longer be considered as tracers and the molecular diffusion should be treated explicitly": I do not believe this is right. Molecular diffusion effects should be small compared to eddy mixing and mean meridional advection below the upper mesosphere (75-80 km), and certainly within the range of altitude of the present simulations (top boundary at 0.1 hPa, about 65 km).*

**Response:** Since the gravitational separation occurs in the stratosphere, molecular diffusion should be accounted for in order to reproduce the separation.

*(225) "flux decreased by several orders of magnitude . . . at the level of a few Pa": But in Fig. 3 all flux profiles increase with altitude. What is the definition of the flux shown in that figure? Does it not include a density factor?*

**Response:** The plotted quantity is $\tilde{F}(p)$, which is defined as upward flux $F(p)$ [kg/m2] normalized with mass mixing ratio $\xi(p)$ [kg/kg] at each level. The statement is about the flux $F(p)$, which indeed vanishes as the destruction rate gets higher. In the revised version we use $F(p)/\xi(p)$ everywhere instead of $\tilde{F}(p)$.

*(226) "shown in Fig. 3 with solid lines": Flux profiles in Fig. 3 are labeled Kz, 0.1Kz, 0.01Kz and 0.001Kz. Do these correspond to the Kz profiles of Fig. 2, except that 0.001Kz is not shown in that figure?*

**Response:** Yes. Corresponding note added. The figures have been changed to same set of profiles.

*(234) "For higher eddy diffusivity . . . molecular diffusion . . . becomes negligible": This should be the situation in the middle atmosphere up to about 75-80 km. Gravity wave parametrizations yield values of Kzz of order 10 m2 s-1 in the lower mesosphere (around 65 km); and mean vertical advection is also large at these altitudes. Therefore, for all practical purposes the effects of molecular diffusion and gravitational separation should be negligible over the range of altitude considered in this study. Note also that, according to Fig. 3, molecular diffusion effects are essentially irrelevant even for unrealistically weak values of eddy diffusion near the upper boundary (0.1 Kz and 0.01 Kz).*

**Response:** We have to respectfully disagree. Molecular diffusion is the mechanism behind the gravitational separation. The effects of molecular diffusion on SF6 are negligible only because of the depletion. We are not aware of any earlier studies that explicitly quantify the effect of molecular diffusion on SF6 and apparent AoA, this, we believe that molecular diffusion is worth considering.

*(246) "uppermost layer": What is the upper boundary condition on the circulation? Does it force the vertical velocity to be zero at the top boundary? If so, note that the effect of mean meridional transport on SF6 distribution and lifetime will not be modeled realistically.*

**Response:** Yes, we force zero vertical velocity at 0.1 hPa. We put it more explicitly in Sec. 3. A note on mean meridional transport added to Discussion.

*(265) "'ones' tracer": "unity tracer" might be better.*

**Response:** The term replaced.

*(304) "the southern polar region": What range of latitude does this cover?*

**Response:** The corresponding note has been in the figure caption. Same duplicated in the text now.

*(320) "inter-annual variability": This strongest variability seen in Fig. 4 is annual, presumably associated with the cycle of downwelling in winter and upwelling in summer. This again brings up the question what is the upper boundary condition on the dynamics (cf. comment at line 246), and how realistically mean vertical advection is modeled near the upper boundary. Note also that the mean meridional circulation in the mesosphere depends strongly on the contribution of gravity wave drag to the zonal-mean momentum budget, which would depend on how this is parameterized in the ERA-I reanalysis. Details on all of these points are needed.*

**Response:** Note on annual variability added. Specification of the upper boundary condition added to "Model setup" section, and discussion of the resulting artifacts added to new "Discussion" section.

*(324) "simulations with 0.01 Kz": Do you mean 0.001 Kz? That is what the legends in the left column panels of Fig, 4 indicate.*

**Response:** The description was from the previous version of initialisation run, which was indeed made with 0.01 Kz. The new description is made more consistent.

*(330) "molecular diffusion . . . maintains the upward flux . . . even if eddy diffusivity ceases": But in the real world, the flux at 65 km (0.1 hPa) is controlled principally by the combined effects of eddy diffusion and mean vertical advection.*

**Response:** We put more clearly that the statement is about the model. The relation of the model to the real world is addressed in "Discussion" section.

*(341) "vertical exchange is a key controlling factor": This is correct, but note again that flux due to mean vertical advection is also important and may or may not be modeled properly in the present study, depending on how the upper boundary condition is handled.*

**Response:** The description of the upper boundary condition was clarified.

*(357) "way and rate of SF6 destruction": What does "way of SF6 destruction" mean? You have varied the effective loss rate by changing the flux at the upper boundary, but as far as I can tell the loss mechanism was not changed.*

**Response:** We meant that in the troposphere even difference in passive vs. non-passive SF6 is small. The statement rephrased.

*(365) "the most diffusive case . . . overstated SF6": This is likely due to the fact that the "1 Kz" profile has too large values in the lower stratosphere (although it has more reasonable values in the upper stratosphere and lower mesosphere).*

**Response:** Agree. The next sentence points exactly that. The discussion on how reasonable the profiles are has been added to "Discussion".

*(373) "largest deviation below 20 km": See previous comment.*

**Response:** Rephrased.

*(375) "WACCM . . . under-representing the depletion of SF6 inside the polar vortex": The problem with the WACCM result is that the standard version of the model does not include loss due to electron attachment, only photolysis. I would expect WACCM to simulate SF6 quite accurately if all loss terms were included. What this result demonstrates is that it is essential to include loss via electron attachment.*

**Response:** Thank you! Corresponding note has been added. We agree that "it is essential to include loss via electron attachment". However, comparison of our Fig. 7 to Fig. 3 from the next WACCM paper by Kovács et al. (2017) shows that just including loss via electron attachment into WACCM is not sufficient to reproduce the SF6 profiles in polar regions.

*(386) "In all the above cases, the '1 Kz' profile is . . . too diffusive": I don't see this in all cases. The 1 Kz profile produces good agreement in the upper stratosphere in Fig. 6 b and d.*

**Response:** Rephrased. Despite the modelled SF6 profiles for '1 Kz' are the closest to observed points above 25 km within the polar vortex, they fail to reproduce the shape of the profiles there.

*(387) "The '0.03 Kz' profiles appear to be most realistic": Actually, none of the profiles is realistic. In particular, the range*
*of Kzz as a function of altitude obtained from gravity wave parameterizations is much larger than shown for the 1 Kz profiles*
*of Fig. 2, where Kz varies from a little under 1 m2 s-1 at 20 km to less than 10 m2 s-1 at 65 km. In models that include a gravity*
*wave parameterization, Kzz is estimated to vary between less than 10-3 m2 s-1 and more than 10 m2 s-1 over the same range*
*of altitude. For a recent example, see Zhu et al. (JAS, 67, 2520, 2010).*

**Response:** The word "realistic" was meant for SF6 profiles rather than for Kz profiles. The statement rephrased. The Discussion on how (un-) realistic the Kz profiles has been added.

*(432) "lack of a pole-to-pole circulation": Is this a result of the way the upper boundary condition is handled in these*
*simulations? You need to show the Transformed Eulerian mean circulation as a function of altitude and latitude, at least for*
*the solstice seasons, so the reader can understand the role of mean meridional transport near the top boundary of the model.*
*Explicit description of how the upper boundary flux is handled in SILAM is also necessary.*

**Response:** "lack of a pole-to-pole circulation" is a known feature of ERA-Interim reanalysis. The analysis of the mean circulation and distortions introduced by the "hardtop" diagnostics has been added to the paper. We compare the seasonal-mean diagnosed vertical velocity fields used for the run to the ones obtained from ERA5 meteorological dataset. For ERA5 "hardtop" was implemented at 10 Pa, to match one in our simulations and at 0.1 Pa to have a reference case.

*(433) "understate it above 40 km": In this instance, one could also question the observations, especially the ones that show*
*an increase in mixing ratio with altitude. It is unclear how such profiles could be generated for a tracer that has a source in*
*the troposphere and a sink in the mesosphere.*

**Response:** We agree, such behaviour is counter-intuitive and is probably related to the observational uncertainty. However, an intermittent increase of the mixing ratio with altitude is possible and could be noticed also in the model results, e.g. the SF6 shape of the non-collocated 0.001Kz profile at Fig. 6g of the revised manuscript.

*(435) Figure 8: Is the "de-biased RMSE" in the figure caption the same thing as "STD" in the ordinate label of the top*
*panel? It would be desirable to keep the terminology consistent to avoid confusion.*

**Response:** Thank you! Replaced with "standard deviation of model-measurement difference".

*(436) "the difference in statistical scores of the three selected simulations is quite minor": What "statistical scores" are you*
*referring to?*

**Response:** The scores (RMSE, bias and NMB) shown in the figures. The sentence has been rephrased.

*(440) "standard deviation of model-measurement difference": How does this eliminate the influence of model bias, assuming*
*that is what you mean to say here? Doesn't the model-measurement difference contain the bias? A formal definition of this*
*quantity, similar to what is done in Eq. (11) for the NMB, would be useful.*

**Response:** Definitions for standard deviation of model-measurement difference (STD) and absolute bias added, along with normalised mean bias (NMB).

*(444) "RMS error of the observations due to retrieval noise in the original MIPAS data": Is this what you mean by the legend "MIPAS noise" in the top panel of Fig. 8?*

**Response:** Yes. Note added.

*(450) Figure 9: This needs labels for the various curves, as in Fig. 8.*

**Response:** Corrected.

*(452) "for the upper troposphere": What does this refer to? This paragraph discusses results for 30-60 km. What does this have to do with the upper troposphere?*

**Response:** We meant the upper stratosphere. Misprint corrected.

*(460) "Three other profiles of Kz result in practically identical distribution of AoA": This would imply that vertical mixing is irrelevant for the small Kz cases, and raises the question what controls AoA in these simulations.*

**Response:** The AoA is controlled by the transport with explicitly resolved winds, which have a dominant effect unless the eddy diffusivity is too high. Corresponding note added.

*(484) "The resulting model-based apparent AoA [is] much older than the "ideal-age" AoA and pretty close to the values derived from MIPAS": This is an important result that highlights the role of fast mesospheric destruction of SF6 due to the electron attachment mechanism.*

**Response:** Thank you! The statement has been added to the Conclusions.

*(503) "The reason is. . .": You should reference Stiller at al. (2012) here, who already pointed this out.*

**Response:** The reference added.

*(512) "'ideal age' and "passive" tracers: Are the results for the "ideal age" tracer the set of points labeled "time lag" in Figure 12? Again, consistency in terminology would be desirable.*

**Response:** Corrected

*(537) "eddy-diffusivity profile of Hunten (1975) scaled down": The Hunten profile almost certainly overestimates diffusivity in the lower stratosphere, but reducing it by a factor of 0.03 will not reflect the behavior of vertical mixing in the upper stratosphere and the mesosphere. Ideally, one would estimate vertical mixing (as a function of altitude, latitude and season) from a gravity wave parameterization. Since such a parameterization was not available in the context of the present study, the conclusions regarding the role of Kz in determining age of air cannot be taken at face value.*

**Response:** We agree that proper parametrisation of eddy diffusivity would be more appropriate. The next sentence explicitly states that the conclusion is specific for our setup, and in the revised version we have tried to make it more clear. In the future we would be happy to implement some more realistic Kz scheme, that would use some physical parameters that govern the turbulence rather than just altitude, latitude and season.

*Typos, etc.:*

*(28) "that presents an analogy of Lagrangian clock" -> "that is analogous to a Lagrangian clock"*

*(33) "are not possible, therefore. . ." -> "are not possible; therefore, . . ."*

255 *(318) "is by more than an order of magnitude stronger than one of gravity separation"*

*-> "is stronger than diffusive separation by more than one order of magnitude"*

*(319) "Regardless the used Kz profiles" -> "Regardless of the Kz profiles used"*

*(344) "depleting SF6" -> "SF6 that undergoes chemical destruction"*

*(344) "start to fall down" -> "begin to decrease"*

260 *(452) "on pair" -> "on par"*

*(482) "nor its mixing ration" -> "nor does its mixing ratio"*

**Response:** Thank you! Corrected accordingly.

**2 Response to the Interactive comment RC2**

*The study by Kouznetsov et al investigates the impact of the vertical diffusion and of the mesospheric sink of SF6 and the SF6*
265 *climatology and its trends using a chemistry transport model. While the mesospheric transport is not explicitly included (due to lack of ERA-Interim data above 0.1 hPa), a parametrization of eddy diffusivity as well as molecular diffusivity is included to mimic transport to the mesosphere. The subject of the study is of high relevance, as SF6 is used frequently to estimate Age-of-Air, and the role of its sinks needs to be better understood. The study is overall well presented and the methods are overall appropriate, but some clarifications are needed (see comments below). Overall, I recommend the authors revise the*
270 *paper minorly before it can be considered for publication.*

*General comments:*

*1. In lines 45 ff, you correctly mention that a correction has to be applied when deriving AoA from a non-linear increasing tracer, as SF6, as has been done by observational studies. However, it is not entirely clear to me how you calculated AoA from SF6 simply as time lag, as for the linear increasing tracer? It certainly is known that just calculating the time lag leads to*
275 *deviations from the true AoA values. If you choose not to include a correction method in the calculation of AoA, you certainly should stress this fact, and I suggest you to refer to the SF6-derived "AoA" as "time lag" rather than AoA. The comparison of the SF6-derived time-lag with / without chemical sink is still valid, but I caution you on the conclusions you draw from the difference of the passive sf6 tracer and the ideal age /linearly increasing tracer: as long as no correction method for the non-linearity is implied, you cannot conclude on whether the non-linear increasing tracer can be used to deduce AoA values*
280 *in general.*

**Response:** We agree that the difference between "time lag" and AoA is influenced by the source variation and non-linear growth of concentrations. However, as shown by Waugh and Hall (2002), the "time lag" is a function of both the variation of surface concentration and the transient time distribution (TTD, also known as "age spectrum"). While the surface concentrations of SF6 are relatively well known, TTD is quite uncertain and can be only partially constrained with multi-tracer
285 observations. Therefore, we believe that no fully-consistent correction to the "time-lag" AoA can be designed solely on SF6 distribution and non-linearity of its growth. Without this correction, the time lag is somewhat different from the mean age. The

"apparent age" however refers to the much larger problem of the strong $SF_6$ loss in the mesosphere and describes the fact that the age derived from $SF_6$ subsided from the mesosphere is much older than the realistic age. The corrections applied this-far in the literature do not address it. Throughout the paper, we refer to the $SF_6$-derived "AoA" as "apparent AoA", which is derived without any corrections. A paragraph has been added to the "AoA and apparent $SF_6$ AoA" section explicitly pointing it out and noting that our conclusions refer to this very quantity.

*2. While the parametrizations of eddy diffusivity, gravitational separation by molecular diffusion and of SF6 loss are described well in detail, the way they are actually implemented in the model is not entirely clear to me. According to Section 3.4, the overall budget equation of the abundance of a tracer (SF6) is solved for steady state, and this steady state solution scaled by the actual tracer concentration is used above the model top - is this correct? And how exactly is this implemented - as loss due to the lifetime given in line 238 ? Furthermore, it was not clear to me whether the diffusive parametrizations are also applied in the actual model domain, or only for the parametrization above the top level? It could be helpful if you describe the overall approach at the beginning of section 3 (i.e. parametrization of upward transport above ∼10Pa by vertical diffusion, where SF6 is depleted, and thus there is no downward transport of SF6).*

**Response:** The way the molecular diffusion is implemented has been described in the last paragraph of the "Molecular diffusivity and gravitational separation" section. The loss of SF6 through the domain top was implemented as a linear decay of $SF_6$ in the topmost model layer, at a rate derived from the $K_z(p)$ profile used in each simulation. This is now expressed more explicitly at the end of the "Parametrization for destruction of $SF_6$ in the mesosphere" section.

As it was stated in ll. 287-289 of the original submission, the runs were made with a set of eddy-diffusivity profiles and corresponding SF6 destruction rates in the topmost layer. The Kz was adjusted inside the model domain (ll 170-174 of the original submission). The statement in the last paragraph of the "model setup" section has been reformulated to make it more explicit that Kz was adjusted inside the model domain accordingly.

*3. Related to the above comment, I wonder how sensitive your results are to the fact that you represent transport above the model top only as vertical diffusive process, i.e. the actual transport circulation is missing (which circulates air, and thus SF6 from pole-to-pole, as opposed to your assumption of all SF6 that is transported diffusively upward being lost). Probably the lack of advective transport also affects the results of the evaluation of different values for Kz? Or is this more based on the layers within the model domain (if diffusion is applied there too, see comment above)? Please add discussion of those issues to your study.*

**Response:** The discussion of the effect of a hard "lid" for regular transport has been added. The current simulations indeed could not include the pole-to-pole circulation due to the limitations of ERA-Interim. However, as we showed, lifetime of SF6 quickly reduces starting practically from the top of our domain. Therefore, the impact of the missing topmost layers is bound to be limited, mainly reducing the SF6 concentrations in the downdraft regions: there is little SF6 above 60 km.

*Specific comments: - line 25: you describe here the estimation of AoA with Lagrangian trajectories, but without inter-parcel mixing. The inter-parcel mixing does affect AoA, and there are studies that account for this mixing in Lagrangian frameworks (e.g. Brinkop and Jöckel, 2019; Plöger et al., 2015b) Thus estimates of AoA with Eularian methods might differ*

*from Lagrangian methods due to the way inter-parcel mixing is calulated. This methodological point should be mentioned somewhere.*

**Response:** References added, the role of mixing is emphasised.

*- line 27: "above-mentioned observational method": I dont see you mention the observational method above this statement?*

**Response:** Corrected

*- line 44: Garcia et al did show that the corrections improve the trend estimate, and they do not use the exact same correction method than what was applied to the observations. So I would not argue that the tracers are "ambiguous proxies" for AoA, but rather that the correction methods accounting for the non-linearity need to be investigated more deeply.*

**Response:** Garcia et al. argue that corrections would need a knowledge of age spectra in order to estimate a mean age. The corrections we are aware of (e.g. Stiller et al., 2012) are based on the assumed shape of age distribution and validity of a world constant of $w = 0.7\,\text{year}$ that describes the broadening of the spectra. One of the goals of the paper is to show that there are more direct and more accurate ways of computing AoA without involving such corrections. We added discussion of the corrections to "discussion" section.

*- line 95 ff: Maybe you can mention here which variables from ERA-Interim you use - I was wondering at this point how vertical transport is calculated, and this became clear only in section 3.5.*

**Response:** List of variables and ref to Sec. 3.5 added.

*- line 122 ff, general: How certain are the SF6 destruction rates, i.e. how do the results by Totterdill et al compare to other studies? Please add a short statement.a*

**Response:** Added. Intriguingly enough, IPCC (2013, Sec 8.2.3.5) states that photolysis is the main destruction mechanism without references. Totterdill et al. (2015) says:"Photolysis is currently recognized as the major sink of SF6 (Ravishankara et al., 1993), though with a significant contribution from electron attachment in the upper mesosphere and lower thermosphere (Reddmann et al., 2001)." However (Reddmann et al., 2001) clearly shows the dominant role of the electron attachment below 80 km.

*- line 156: its not clear to me what the "limiting value" is, and why Kz is "practically always" set to it? Please be more specific here.*

**Response:** The sentence discarded as redundant. It was supposed to stress the idea that ECMWF scheme is equivalent to zero-Kz in the stratosphere.

*- line 159: Kz does not fall below the molecular diffusivity in the lower stratosphere, below ~40 km, according to Fig. 2, so please refine the statement.*

**Response:** Rephrased.

*- line 196: do you mean mixing ratio differences between the two layers? Why two layers, and not at one layer? Or do you mean the mean mixing ration in the layer bounded by an upper and lower pressure? it might be easier to put down the equation rather than describing it.*

**Response:** Thank you! The equation is indeed more clear.

*- line 212: I assume you use the US standard atmosphere because at the levels where it matters, ERA-Interim is not available any more? Again, it is not entirely clear if / how you apply this parametrization only at the "top layer", or also throughout the model domain. If the latter is true, the actual ERA-Interim temperatures could be used in the model domain (even tough you could argue that it does not make much of a difference there, as molecular diffusion does not play a role).*

**Response:** We use standard atmosphere because it allows for pre-calculating a single set of exchange coefficients for a given species and vertical discretisation. The coefficients applied throughout the domain with a simple explicit scheme. The paragraph rephrased to emphasise this.

*- line 224: please be more specific and describe how you obtain the flux F(p) from the steady-state solution of the mixing ratios.*

**Response:** Rewritten. The requested details added.

*- line 236: see general comment: please be more specific on how exactly the different parametrizations are used in the different areas, and how the upper boundary parametrization is implemented (via the lifetimes?)*

**Response:** Yes. Note added at the end of the "model setup" section.

*- line 250: which "other parameters" do you use?*

**Response:** Surface pressure, temperature and humidity. Note added.

*- line 267: were the other tracers corrected using the "ones" tracer, or just the error "evaluated"?*

**Response:** Yes. Note added.

*- line 343 ff: is this the best way to estimate lifetimes, or couldn't you just average the inverse destruction rate mass-weighted over the entire atmosphere?*

**Response:** Yes. Corresponding note has been added.

*- Also, at line 348, you write that the delay of SF6 between troposphere and upper layers is about 5-6 years, and then use the value 5 years previous to the emission stop to evaluate the lifetime - is this quantitative, or just a rough estimate?*

**Response:** 5-6 years is an estimate of AoA in the topmost model layer. Corresponding note added.

*- line 361: "we have found in literature"-> be more specific, e.g. observations that were published by ...*

**Response:** The references duplicated from the figure caption.

*- line 365: "strong exchange through the troposphere"? do you mean too strong upward transport by the diffusion?*

**Response:** Yes. Thank you!

*- line 384: what is the dynamical reason for the minimum in SF6, and why do you think it is weaker in the model?*

**Response:** The minimum is a result of the spring breakdown of the polar vortex, when a regular down draught ceases, and atmospheric layers decouple from each other. The reduced depth of the modelled minimum is probably caused by insufficient

385   decoupling of the layers in the driving meteorology. Since we make an offline modelling, driving meteorology is a usual suspect. Corresponding note added.

   *- line 464: "practically"? please be more specific*

   **Response:** The sentence was about SF6 profiles rather than Kz profiles. Rephrased.

   *- line 471 ff: as mentioned in the general comment, you should clarify how AoA was calculated from the SF6 tracers, and*
390   *possibly change the naming to "time-lag".*

   **Response:** A note on the method added. As AoA derived from passive Eulerian tracers is a time lag by definition, adding "time-lag" would be probably redundant in this case.

   *- line 486: you mentioned earlier that you use a new version of the MIPAS SF6 data, but do not show its AoA, but instead refer to the older published AoA figures. Why don't you add the new MIPAS AoA to Figs. 11 and 12?*

395   **Response:** Figs. 11 and 12 show average model fields. As it is shown in Fig. 7, average of sampled fields differs noticeably from the "true" average. Adding observational data would require model data to be sampled according to observation timings and averaging kernels, which would change the message of the figures. Their purpose is rather to show, how sensitive AoA and estimated trends can be to the choice of particluar method to infer it. For that, we need full model output without down-sampling to the satellite overpasses.

400   *- line 515: "non-uniformity" of ERA-Interim, what do you mean? Couldn't this just be the trend in AoA over the period, or why do you think it is an artefact? Further, in line 519, you state that ERA-Interim was not recommended for climatological studies. I'm surprised by this statement, given that ERA-Interim is the basis for a lot of studies of climatologies and trends in various variables. Can you specify which source you quote here, and what exactly should not be done?*

   **Response:** Figures 21 and 22 of Dee et al. (2011) indicate a clear impact of the inhomogeneous assimilated data set on
405   analysis increments. We have found similar features in the simulated ideal-age AoA. The main reason for the inhomogeneity in the ERA-Interim data is varying amount of observations from year to year, as shown by ?????Dee et al????????. Excluding artefacts caused by changes in the amount of the observational information would require extensive effort and an independent homogeneous dataset, which does not exist for AoA. Deducing reliable trends even for atmospheric temperature, quantity that is directly and indirectly measurable and has been extensively assimilated throughout the whole ERA period, was a major effort
410   (Simmons et al., 2014). Therefore, we are sceptical on mere possibility of deducing reliable trends for AoA using ERA-Interim alone.

   *- line 521: The trends over the MIPAS period could be compared to other CTM results, e.g. by Plöger et al. (2015a), who showed that their CTM was capable to reproduce the MIPAS AoA trend rather well.*

   **Response:** The reference and discussion of the differences added.

415   *- line 525: why comparable with Lagrangian simulations? As pointed out before, one difference is the accounting for inter-parcel mixing. I'd rather argue that your results are comparable to other state-of-the-art CTM simulations of AoA.*

   **Response:** Thank you! Corrected.

*- line 542: Are those "best" estimates in the upper stratosphere based on the "upper layer", where advective transport is not accounted for? Or do you refer to the results in the model domain?*

420 **Response:** We refer to the results in the model domain. Remark added.

*-line 549: I dont understand the sentence on the standard deviation controlled by noise. Do you mean to say that the standard deviation between model and MIPAS is about as large as the error on the satellite data?*

**Response:** Yes. The remark added.

*- line 551: you might want to add the range of lifetimes you obtain.*

425 **Response:** Added.

*- line 560: as stated in the general comments, as long as you do not apply corrections for the non-linear growth, you can not conclude on the suitability of the non-linear tracers in general. You can conclude here that without correcting for the non-linear growth, the apparent AoA and its trends deviate strongly, and that this motivates the investigation of correction methods.*

**Response:**

430 We agree, that one could more-or-less compensate for the non-linear growth. The main issue with SF6 age is the mesospheric sink, that can hardly be compensated. The statement has been rephrased to make it more explicit.

*Typos/ Language / Technical:*

*- Abstract, line 11: ".. does not exceed 6-6.5 years": it is not clear to me what this statement refers to - is this the "true" (ideal age) maximum value for AoA?*

435 *- line 18: what do you mean by "polar circulation" ?*

*- line 37, and general: check the parenthesis around references, they are incorrect at several places, for example here it should read (Waugh, 2009 ; Stiller et al., 2012)*

**Response:** Corrected

*- line 86: "transformation procedure" - do you mean the chemical "transformation"? -> change to "chemical sinks" (?)*

440 **Response:** We mean both sink of SF6 and the increment of "ideal age", and also molecualr diffusion for SF6. "corresponding transformation and transport routines" is hopefully less ambiguous.

*- line 122: over 60 km -> above 60 km; "that fall..." -> "i.e. within and above the top most..."*

**Response:** Nominal top of the ERA-Interim (topmost half-level) is at the top of the atmosphere.

*- line 159: please avoid using the word "practically", as it is not very specific*

445 **Response:** Removed/replaced.

*- line 168: "than ones accepted.."- Do you mean "than the ones usually used in models"?*

*- line 176: "the mesosphere" (add the)*

*- line 196: "in the vertical, one obtains that the ..."*

*- line 202: the overwhelming" (add "the")*

450 *- line 247: remove "been"*

*- line 267: "rations" -> "ratios";*

*- line 300: "downdraught" -> "downwelling"*

*- line 344: fall down -> decrease*

*- line 383: "the one in Fig.." (add the)*

455    *- line 386: "furthermost" -> furthest*

*- line 419: to the polar (replace "a" by "the")*

*- line 426: overstating -> overestimating*

*- line 452: do you mean upper stratosphere?*

*- line 482: "nor its ..."-> "nor does its mixing ratio" (remove "n")*

460    *- line 484: replace second "well" with "with"*

*- line 490: pointed out (add "out")*

**Response:** Thank you! Corrected.

**3   Response to the Interactive comment RC3**

465    *This is an interesting manuscript exploring, in a model environment, the effects of chemistry, gravitational separation and diffusivity on SF6 mixing ratios in the stratosphere and the mean age of air derived from it. Clearly a lot of work has gone into devising the various model setups and I would in general support the publication of this work. However, some questions need to be answered and some potential issues resolved beforehand. One example is e.g. that even though it is driven by ERA-Interim there is no guarantee that this model will accurately reproduce stratospheric transport patterns including the overturning*

470    *circulation, transport barriers, the QBO, etc., all of which can influence AoA. Perhaps this was demonstrated in Sofiev et al., 2015? If so, it would be good to give a short summary, if not, some further details are required. Some further points can be found in the below.*

     **Response:** Sofiev et al., 2015 describes the transport procedure used in SILAM without touching any specific meteorological driver. ERA-Interim until recently had been a State-of-the-Art reanalysis which has been evaluated in many studies. The

475    purpose of the present study was not to analyse in details how well the particular phenomena are reproduced by ERA-Interim, but rather to simulate SF6 evolution and distribution simultaneously with other AoA-related quantities, and see how well SF6 and AoA can be reproduced within a single model run and whether their errors are correlated. Moreover, quite a few findings of the study are valid regardless the quality of the meteorological driver. Particular features of ERA-Interim and its interfacing to SILAM are considered only as long as it is necessary for the main topic of the paper. .

480    *Title: Consider adding "the" before "distribution" and "SILAM". There are various other places in the manuscript with small language deficiencies like this.*

     **Response:** Corrected

*Abstract: An introductory sentence to alert the reader to the fact that this paper is on the stratospheric overturning circulation (and perhaps its importance) would be helpful.*

485 **Response:** We have tried to focus more on physical processes controlling SF6 concentrations in a given velocity field, and a way to simulate it, without getting into too much details about stratospheric circulation in general.

*Line 11: This should be "adds" and I would also recommend adding "up to".*
**Response:** Rephrased.

*Line 32-33: Age is not the correct term here as oceanic water has been around for some time. I suggest replacing it with e.g.*
490 *"transport times".*
**Response:** Corrected

*Line 109-113: I was quite surprised to find a new satellite product hidden in this modelling-focused manuscript. Given that there are "considerable" differences to previously published SF6 data sets I urge the authors to provide more details and make their statements more quantitative (e.g. defining "considerably higher" and "closer"; where does the "new" CFC-11 band*
495 *come from?; does the correction influence trends in the 2002-2012 period?), perhaps even by adding a figure to support their claims.*

**Response:** The $SF_6$ data used in this paper are retrieved following the procedure described by Haenel et al. (2015). The only difference to the latter dataset is the use of newly provided spectroscopic information. $SF_6$ mixing ratios are up to 0.6 pptv higher in the upper stratosphere above 35 km, with main differences in the tropics and the polar regions. This brings AoA
500 above 25 km in close agreement with reference balloon data as shown in Waugh and Hall (2002). The AoA trends change on detail level, however the general pattern with increasing AoA in the NH and decreasing AoA in the tropics an SH remains. A paper on the differences between the new and the Haenel et al. version is in preparation (Stiller, G.P., J.J. Harrison, F.J. Haenel, N. Glatthor, S. Kellmann, N.N., Improved global distributions of $SF_6$ and mean age of stratospheric air by use of new spectroscopic data, to be submitted to Atmos. Chem. Phys.). Similarly, a paper on the laboratory measurements of the $SF_6$
505 absorption cross sections is in preparation by J.J. Harrison. The "new" CFC-11 band in the spectral vicinity of the $SF_6$ spectral signature is described in Harrison (2018)

*Line 118-119: Figure 1 is bad quality and Figure 2 needs some further explanation in the caption.*
**Response:** Figure 1 is an attempt to plot the parametrisation over the original graphics from Totterdill et al. (2015, Fig. 9). Caption of Figure 2 extended.

510 *Line 119: Is it Silam or SILAM?*
**Response:** SILAM is used through the manuscript now. Backronym introduced.

*Line 170-171: Looking at Figure 2 none of the three profiles seem to capture the vertical gradient from the ERA-5 data. Why is that?*

**Response:** The ERA5 Kz profile is below the molecular diffusivity. Thus we have more questions to the order of magnitude than to the gradient. Investigating the physical reasoning behind the ERA5 profiles is definitely worth the effort, we plan it for the next study.

*Line 219-220: Please quantify: How negligible does vertical advection need to be? And how does that compare to actual vertical advection in the stratosphere and mesosphere?*

**Response:** Brief note on the magnitude of vertical advection added. The comparison to the magnitude of the vertical advection in ERA5 added to the discussion section.

*Line 259: The details of the simulation setup are beyond my expertise. However, this statement seems somewhat vague. What does "normally" mean here? And how large is the precision of the input wind fields? Does it e.g. vary over time?*

**Response:** The precision is a feature of the way the fields are encoded. More specific figure added.

*Line 273-275: This is a major problem. A linear extrapolation can introduce biases, especially since 4 years of the extrapolated period overlap with the MIPAS period. Why do the authors not use more up-to-date publically available data, e.g. from the AGAGE and NOAA ESRL networks. Looking at Figure 1-21 in the recent Scientific Assessment of Ozone Depletion (2018), global emissions of SF6 appear to have been much lower between 2008 and 2016, closer to 0.21 Gg/yr. The implications for the derived AoA and its trend could be quite severe and should at least be assessed.*

**Response:** Thank you for the reference. Corresponding note and citation added. The difference is about 7% of annual total emission by 2016, which is comparable to the uncertainty range shown in Fig 1-21 of Engel et al. (2018). The difference is certainly worth accounting in follow-up studies, but we do not see how it could noticeably affect the results of the present study.

*Line 287-294: This paragraph raises a few questions. What does "0.001Kz eddy diffusivity" mean in detail and why was the initialisation performed that way? When was the initialisation started? Which emissions were used for SF6 species from 1980-1989 and which meteorological fields for 1970-1979? What about the pre-1970 time period?*

**Response:** The description has been corrected with more details provided.

*Line 306: This should be "from".*

**Response:** Corrected

*Line 313-314, 353-355: Why are the lifetimes in Table 1 so long? It looks like the model is not able to reconcile realistic diffusion rates with recently published lifetime estimates for SF6 (e.g. Kovacs et al. and Ray et al.). This needs to be discussed. I also do not agree that there is good agreement with lifetime estimates from other studies (Line 353-355) as all of the higher lifetime estimates ($\gtrsim$ 1500 years) come from outdated studies.*

**Response:** We are not so certain about the over-estimation in the older studies and whether the shorter lifetime is a evidence-based consensus now. For instance, the model of Kovács et al. (2017) overstates the SF6 content in the altitude range of 25 – 50 km (compare Fig. 3 there to the Fig. 7 of the manuscript). It should lead to enhanced transport of SF6 towards the depletion layers and thus overstate the destruction rate and underestimate the lifetime. The estimate of Ray et al. (2017) is

based essentially on a single observed profile of SF6. The subsection is moved after the SF6 evaluations, and corresponding discussion added.

*Line 399-400: Please improve Figure 7. It is currently very hard to decipher the legend and text inside the graph area and the two lines for each colour are undistinguishable.*

**Response:** The legend has been moved to the upper-right panel. The thin lines made also dashed and the grid made light-grey. Hopefully, the figure is more readable now.

*Also, please add the uncertainties of the MIPAS data points (one could at least add standard deviations of the observed values as in Kovacs et al.) and why are SF6 mixing ratios increasing at the high altitude end for some profiles?*

**Response:** The uncertainties added. The note on non-monotonous profiles added.

*Line 471-475: Plotting the residual between 11a,b and c might help visualising the differences. Also, please quantify "slight old bias".*

**Response:** Colors are indeed difficult to compare. The shape of the isolines, however seems to visualize it quite well. "slight old bias" replaced with more specific "old bias up to 3-5 months".

*Line 547-548: Looking at Figure 7 I cannot agree with this statement, at least not until some realistic uncertainty estimates have been added to the observations.*

**Response:** All but one balloon data sets come without uncertainties. Mipas uncertainties added. The satement rephrased.

*Line 551-552: This is right at the upper end of recent estimates, so not too good agreement. Given that the authors state themselves earlier in the manuscript that "insufficient vertical resolution of ERA-Interim in the upper stratosphere and lower mesosphere, and lack of pole-to-pole circulation" limit model performance (resulting also no conclusion being drawn on AoA trends) I find that statement too strong.*

**Response:** The word "good" removed. Also see the notes at the end of the "Lifetime…" section.

*Page 28-29: Figure 8 and 9 are currently not mentioned anywhere in the manuscript.*

**Response:** They were discussed and referenced in lines 434 – 453 of the original submission.

**4   Response to the Interactive comment RC4**

*This paper simulates the impact of the mesospheric destruction and gravitational separation on stratospheric SF6 distribution using a chemical transport model driven by ERA-Interim meteorology. In the model, mesospheric depletion and gravitational separation of SF6 are parameterized as upper boundary conditions. Sensitivity simulation were conducted and the roles of mesospheric destruction, gravitational separation, and vertical turbulent diffusion in the distribution of stratospheric SF6 are determined. The effects of these processes on the derived mean age of air and its trend are also discussed. This paper clearly demonstrate that the apparent mean age of air derived from SF6 measurements is not suited for studying the trend of*

*stratospheric mean age of air. The results have important implications in understanding the differences in the observed and modeled Brewer-Dobson circulation trends. I recommend publication of the paper after my comments are addressed.*

*Comments: My major concern is that the SILAM model doesn't capture the SF6 distribution in the upper stratosphere. The authors attribute this deficiency to the low top of ERA-interim that can't accurately represent the circulation in the upper stratosphere and mesosphere. However, the mesosphere circulation, particularly the downwelling branch of the summer-to-winter pole circulation, is essential to understand how the mesospheric sink affects SF6 distribution in the stratosphere. This issue needs to be discussed in more detail. I wonder if it is possible to drive SILAM with a model of higher top, e.g., WACCM, to see if SF6 in the upper stratosphere can be improved.*

**Response:** Yes. It is possible to drive SILAM with other models. It is in our plans to repeat the exercise with ERA5 once this publication has been complete. Such simulation would be possible to do with more rigorous handling of depletion, since ERA5 covers the depletion layers quite well. The processes are still to be implemented, however.

As an illustration on how the low-top affects the circulation we have added a comparison of the mean vertical transport in SILAM driven with ERA-Interim to the one driven with ERA5.

*Section 5: Describe how the mean age of air is derived using SF6.*

**Response:** The procedure described in the beginning of the section 5.2, together with a brief rationale behind the choice of the method.

*Lines 484-486: Figure 7 shows that the simulated SF6 distribution doesn't agree with MIPAS measurement about 40 km (above 30 km in the winter pole). How can the derived AoA agrees with each other?*

**Response:** The agreement is rather qualitative: both SILAM and MIPAS indicate over 10-years-old air in the upper polar stratosphere, which is way above the "ideal-age" estimates. The sentence has been reformulated.

**References**

[revised manuscript text omitted]

---

## Author Response (AR2)

**Responses to the reviewers comments on "Simulating age of air and distribution of $SF_6$ in the stratosphere with SILAM model"**

Rostislav Kouznetsov[1,2], Mikhail Sofiev[1], Julius Vira[1,3], and Gabriele Stiller[4]

[1]Finnish Meteorological Institute, Helsinki, Finland
[2]Obukhov Institute for Atmospheric Physics, Moscow, Russia
[3]Currently at Cornell University, Ithaca, NY, USA
[4]Karlsruhe Institute of Technology, Karlsruhe, Germany

**Correspondence:** Rostislav Kouznetsov (Rostislav.Kouznetsov@fmi.fi)

**1    Response to Anonymous Referee #2**

*Overall the authors addressed the reviewer comments in detail, and they added thorough discussion to the paper. I have no major concerns about the paper and recommend it for publication after a few minor additional comments are addressed (see below).*

5 *Minor comments:*

*- page 22, line 552: Did you actually check the lifetimes obtained via loss rates, and are they indeed similar? I'm wondering whether differences in lifetime estimates in different models might also be due to methodological differences?*

**Response:**   No, we did not. Our period of the stopped emission (2.5 years) is much shorter than the transport time from the surface to the stratosphere. Then these approaches should be equivalent. We fully agree that the differences in the lifetime
10 estimates in different models can be caused by methodological differences. The issue starts from the definition of the lifetime in non-equilibrium system (see forth paragraph of the "Lifetime..." subsection).

*- page 25, line 644: The theory for why sinks introduce apparent trends on AoA was also already mentioned by Schoeberl et al. (2000)*

**Response:**   Thank you for the reference! The reference is added, the sentence referring to Stiller et al. (2012) rewritten more
15 accurately.

*- page 26, line 659: The reason for the difference to the study by Plöger et al. (2015) likely is that they used diabatic heating rates as vertical velocity, and it is known that the diabatic and kinematic vertical transport is inconsistent in the reanalysis (Abalos et al., 2015).*

**Response:**   This note and the corresponding reference added.

20 *- page 27, line 710: Did I understand it correctly that the diffusivities in ERA data are refer to parameterized vertical diffusion? In particular in ERA5, likely a considerable amount of vertical mixing by the resolved gravity waves is included, so you might want to restate the judgement of diffusivities in ERA?*

**Response:** The diffusivities in the ERA5 dataset are given with "mtdh" (mean turbulent diffusion for heat) parameter. It is not clear to us how resolved gravity waves can transport scalars, unless they brake and induce turbulence. Since ERA-interim clearly does not resolve any turbulence, one would expect mtdh to be a measure of mixing due to turbulence and other subgrid processes. Then our judgement holds.

*- page 28, line 730: In general, I wonder what the role of "numerical" diffusion by the advection scheme is?*

**Response:** The scheme used in SILAM has no diffusion, i.e. it does not create down-gradient transport. It has however a hyper-diffusion, which can be controlled by having thin enough vertical layers.

*- page 28, line 749: For a discussion on the correction for the non-linear increase in SF6, you might want to take a look at a new article in ACPD (Fritsch et al, Sensitivity of Age of Air Trends on the derivation method for non-linear increasing tracers). It is important to point out that in principle it is possible to derive "correct" AoA also from non-linear increasing tracers if the parameters are chosen correctly (which is of course very challenging for observational data).*

**Response:** Thank you! We agree that correct choice of the parameters is important. Besides that, the approach of **?** also heavily relies on the assumption that the quadratic fit is a valid model, and that the spectra are steady in time. Essentially, that means that one has to know quite a lot about AoA spectra in order to get AoA properly.

*- page 28, line 723: Did you mean to say "Fig. 14" for a comparison to the vertical velocities?*

**Response:** Yes. Corrected.

*Technical:*

*In general, please recheck the language in the new sections and paragraphs*

*- page 5, line 149: "transport with of SF6" ? (delete with)? - page 6, line 165: "transport" duplicated*

**Response:** Corrected.

*- page 23, line 578: "likely to cause overstating..." -> "which is likely due to higher depletion rates... " (?)*

**Response:** The phrase rewritten.

*- page 24, line 604: "differences of are" - something missing? - page 25, line 651: if AoA -> of AoA - page 26, line 686: "in the meteorological driver" -> e.g. "...in the meteorological data used to drive the model.." - page 27, line 694: missing ")" after Fig. 14 - page 27, line 701: "same-vertical mass-flux" - not sure what the "same" refers to? - page 27, line 706: "One can hope": please rephrase, hope is great in life, but not in a scientific paper - page 28. line 727: explicit referencing to anonymous reviewers is maybe not appropriate, just remove the first part of the sentence. - page 28, line 754: "factor of times" - dont understand this phrasing - page 29, line 761: "it makes a full sense": please rephrase, e.g. "it would be useful to..." - page 29, line 768/769: "comparable" duplicated - page 30, line 810: "ithout" (w missing)*

**Response:** Corrected.

**2 Response to Anonymous Referee #3**

*The authors have improved many parts of the manuscript considerably. However, some issues remain as is detailed in the following. Firstly, it is very hard to find out how and where some of the changes have been implemented. As an example, in response to one of my comments the authors say that "Brief note on the magnitude of vertical advection added. The comparison to the magnitude of the vertical advection in ERA5 added to the discussion section." However, it is not obvious what has been changed and exactly where, since a) there have been various changes to different parts Section 3.4 and b) the entire Discussion section appears in blue in the version of the manuscript included with the references. Similar problems are encountered with many other responses as the authors do not give the lines corresponding to the changes in the manuscript. I would encourage them to consider this in future responses.*

**Response:** We apologise for lack of clarity in the responses and will be more precise in the future. However, the "Discussion" section has been added during the revision, thus it appears as blue in whole.

*Secondly, I would highly recommend proof-reading of the revised document by a native speaker as it still contains numerous language deficiencies (old and new).*

**Response:** We have made another effort to correct language deficiencies in the manuscript. Hopefully, it will become even better after Copernicus language editing.

*Specific points (referenced to the pages and lines in the manuscript version added to the responses):*

*On the response to my first comment of the initial review (which referenced Sofiev et al., 2015): Perhaps the authors have misunderstood this point, so I will rephrase it here: Can you demonstrate that the SILAM model is able to accurately reproduce stratospheric transport patterns including the overturning circulation, transport barriers, the QBO, etc.? Without sufficient evidence towards such a demonstration the frequency distribution underlying the both the SF6 mixing ratios and the mean AoA could be very unrealistic, which would in turn invalidate the SF6-AoA comparison i.e. a main point of this paper.*

**Response:** Apologies for not being sufficiently clear. But SILAM does not compute atmospheric dynamics. Being an offline model, SILAM is only able to reproduce features that are reproduced by a meteorological model that is used to drive it. The Sofiev et al paper showed that SILAM is good at it - this is as much as one say about the model if its class. The ability of ERA-Interim (and SILAM driven with it) to reproduce stratospheric transport patterns including the overturning circulation, transport barriers, the QBO, etc. is certainly important but it is a separate study partly already done by other groups, e.g. Abalos et al. (2015) for Brewer-Dobson circulation.

Imperfection in reproducing stratospheric transport patterns by the model is certainly something to consider when interpreting the comparison results, but it can by no meas invalidate the comparison.

*On the response to my comment on line 273-275 of the initial review: Given that the absolute bias between MIPAS and SILAM appears to be constant until about 2008 but is increasing subsequently (Figure 7). Such an increasing bias would also affect the SILAM-based AoA (and also AoA trends). I therefore ask the authors to provide actual evidence for their claim that it could not "noticeably affect the results of the present study".*

**Response:** The overstating of the emissions is a likely reason for the bias. We added corresponding paragraph to the end of the "Evaluation of $SF_6$ against MIPAS data" subsection. This evaluation is probably the only result affected by the overstating.

*On the response to my comment on line 547-548 of the initial review: As these balloon profiles were published with some uncertainties (and I am sure about three of the four), it should be possible to obtain these from the respective manuscripts, or if necessary through a FOIA request.*

**Response:** The uncertainties are given in Fig. 5c. For Fig. 5a they are within 2% ($1\sigma$)(Patra et al., 1997), and for Fig. 5b and Fig. 5d they are "smaller than the size of the symbol" (Ray et al., 2017). Corresponding note added to the figure caption.

*Line 116-142: I appreciate the response to my comment on Line 119-113 of the initial manuscript. However, the information given in the response should be added to the revised version. Whether it is acceptable to cite two unpublished papers I leave for the editor to decide.*

**Response:** The part of that response about MIPAS $SF_6$ retrievals has been added to the "MIPAS observations..." section. We would like to abstain from reviewing unpublished papers.

*Line 266-67: Please explain what is meant by "The shooting method was used together with bisection".*

**Response:** Rephrased to "The shooting method with bisection was used..." The term seems to be quite common. It is explained e.g. here https://en.wikipedia.org/wiki/Shooting_method in details.

*Line 348: Should this be "0.001 Kz"?*

**Response:** No. The stated in the paper is correct.

*Line 544-579: My count came to 27 mentions of "SF6" in this section, which is a bit excessive given that its title already indicates that it focuses on SF6.*

**Response:** Reduced to 16 (incl. section caption).

*Line 576-579: Higher mixing ratios in the stratosphere and lower mesosphere could indeed cause an "overstating" of simulated depletion rates. However, are the mixing ratios between 25 and 50 km from Kovacs et al. actually significantly different (i.e. outside the uncertainties) from the new MIPAS results presented here? This would especially be worth looking into since the new MIPAS SF6 mixing ratios seem to be higher than those in the old data product.*

**Response:** As one can see from comparing Fig. 6 from the manuscript to Fig. 3 from (Kovács et al., 2017) (corresponding panels from the figures are comparable), that with new data the MIPAS values got indeed slightly higher, but the gap between model and measurements is still quite large.

*Line 691: "ERA-Interim layers diagnosed from ERA-Interim."*

**Response:** Corrected.

**3   Response to Anonymous Referee #4**

*accepted as is*

**Response:**  Thank you!

[revised manuscript text omitted]

---

## Author Response (AR3)

**Responses to the editors comments on "Simulating age of air and distribution of $SF_6$ in the stratosphere with SILAM model"**

Rostislav Kouznetsov[1,2], Mikhail Sofiev[1], Julius Vira[1,3], and Gabriele Stiller[4]

[1]Finnish Meteorological Institute, Helsinki, Finland
[2]Obukhov Institute for Atmospheric Physics, Moscow, Russia
[3]Currently at Cornell University, Ithaca, NY, USA
[4]Karlsruhe Institute of Technology, Karlsruhe, Germany

**Correspondence:** Rostislav Kouznetsov (Rostislav.Kouznetsov@fmi.fi)

*Your further changes and responses seem generally adequate to me and I think that the paper makes some interesting points. I understand that you do not wish to address all of Referee 3's comments by making changes to the paper – and, in the end, you as authors, have to take responsibility for what is in the paper. On the other hand I do feel that some of Referee 3's comments remain valid and that they can be addressed by simple further clarifications.*

5 *Please provided a revised version of the paper which makes the following minor changes. I regard these as straightforward, but if you really do not want to make one or two of them then provide clear arguments why this is the case. After these changes have been made I will accept the paper.*

*[The next comment is quite long – but the summary is that I am requesting a clearer specification of the model and a clearer argument as to why it is fit for purpose.]*

10 *l98-117: Referee 3 requested evidence that the SILAM model was fit for purpose. You have cited the Sofiev et al (2015) paper. That demonstrates that the SILAM model performs well on a number of basic tests, but it does not demonstrate that, when driven by e.g. ERA-I velocity fields, it provides a good simulation of stratospheric tracers. Please take a little more time to explain the logic here.*

*Sofiev et al (2015) show that the SILAM numerical model performs well in a range of problems in simulating tracer advection*
15 *by a specified wind field.*

*Others, e.g. Diallo et al (2012) have shown that using other numerical approaches that ERA-I winds give a good (to the extent that can be assessed) simulation of stratospheric tracers. Therefore on this basis you consider that the SILAM model driven by ERA-I winds will also give a good simulation of stratospheric tracers.*

*But this logic ignores the point that the vertical winds are calculated in SILAM using a certain method – can you be sure*
20 *that this method works for ERA-I wind fields? Have you for example, done any tests of prediction of age-of-air or some other artificial tracer vs the prediction of Lagrangian methods such as those used by Diallo et al?*

*Furthermore the procedure for calculating vertical transport in given in 3.5 is quite complicated and I am a bit confused whether this is the procedure described in Sofiev et al (2015), or whether it is a new procedure – please clarify.*

*You should also make it clear that the model is 3-dimensional and give details of the resolution. You have said that the*
25 *velocities are retrieved on a 500x250 lat-lon grid – I think actually you mean lon-lat grid – but you haven't said explicitly that*

*this is the resolution of the numerical model. Again re vertical levels you have given details of the levels on which the velocities are provided, but have not confirmed explicitly that these are the levels in the SILAM model. (The velocities could have been interpolated to some other grid.) Actually I now see that some of this information is in Section 3.5 – and the information on the resolution of the velocity data is provided again here – that is a bit confusing. At the very least the term 'setup' used in Section 2.1 needs to be more explicit and mention grid resolution etc.*

*This then returns to Referee 3's overall point – can you provide some concrete evidence that the SILAM model gives similar results to other approaches for calculating stratospheric tracers from ERA-I winds or similar. If you cannot then you should say explicitly that whilst the SILAM model has been carefully constructed it is not yet know how it performs against, say, Lagrangian calculations using diabatic vertical velocities and any results are therefore subject to this uncertainty.*

**Response:**

Thank you for the detailed suggestion! We followed the recommendation and expanded the "SILAM. . ." section (Sec. 2.1) by collecting the setup information from other sections and adding the missing data. Details of the meteorological setup are now all collected to the section . Finally, the chemical setup of the run is in the unified sections 3.1 and 3.4, which is now the section 3.3. We did not put it into the section 2 because the list of tracers was dictated by the physical processes considered in the study - and they are described in sections 3.x.

The validity of the meteorological data processing restoring the global mass conservation of the fields is an important question, which however has been addressed by the authors of the methodology. We highlighted it in the updated section 2.2. The second confirmation indeed comes from the comparison with the Lagrangian experiments of Diallo et al. (2012), the current study includes this very comparison in the section 5.2 - the corresponding sentence is now extended. We have also added a reference to that comparison in the section 2.2.

*l449: Give the difference from Kovacs et al in the same units as are used in the plots (and elsewhere), i.e. pmol/mol rather than pptv. Also include a following sentence 'Note that whilst we regard this newer version of MIPAS SF6 data as an improvement, it has not yet been reported in a publication, and on that basis is subject to uncertainty.'*

*Figure 6: include information on meaning of dashed lines in the caption.*

*l471: 'non-monotonous' > 'non-monotonic'*

*l500: 'overstating' – not the right word – I suggest 'overestimating' is better:*

*l540: as above 'over-stated/overstating' should be 'over-estimated/over-estimating'*

*l755: 'and the SF6 observations are potentially a good means to evaluate it'*

*l758: 'downdraft' > 'descent'*

*l769: 'braking' > 'breaking'*

**Response:** Done. Thank you!

**References**

Diallo, M., Legras, B., and Chédin, A.: Age of stratospheric air in the ERA-Interim, Atmos. Chem. Phys., 12, 12 133–12 154, https://doi.org/10.5194/acp-12-12133-2012, 2012.